# KernelBench: Can LLMs Write Efficient GPU Kernels?

Anne Ouyang [* 1]   Simon Guo [* 1]   Simran Arora [1]   Alex L Zhang [2]   William Hu [1]   Christopher Ré [1]
Azalia Mirhoseini [1]

## Abstract

Efficient GPU kernels are crucial for building performant machine learning architectures, but writing them is a time-consuming challenge that requires significant expertise; therefore, we explore using language models (LMs) to automate kernel generation. We introduce **KernelBench**, an open-source framework for evaluating LMs' ability to write fast and correct kernels on a suite of 250 carefully selected PyTorch ML workloads. KernelBench represents a real-world engineering environment and making progress on the introduced benchmark directly translates to faster practical kernels. We introduce a new evaluation metric $\text{fast}_p$, which measures the percentage of generated kernels that are functionally correct and offer a speedup greater than an adjustable threshold $p$ over baseline. Our experiments across various state-of-the-art models and test-time methods show that frontier reasoning models perform the best out of the box but still fall short overall, matching the PyTorch baseline in less than 20% of the cases. While we show that results can improve by leveraging execution and profiling feedback during iterative refinement, KernelBench remains a challenging benchmark, with its difficulty increasing as we raise speedup threshold $p$.

## 1. Introduction

AI relies on efficient GPU kernels to achieve high performance and cost and energy savings; however, developing kernels remains challenging. There has been a Cambrian explosion of ML architectures (Tay et al., 2022; Peng et al., 2023; Dao & Gu, 2024a), but their available implementations routinely underperform their peak potential. We are seeing a proliferation of AI hardware (NVIDIA, 2017b; 2020; 2022; Jouppi et al., 2023; Groq; Cerebras; Graphcore), each with different specs and instruction sets, and porting algorithms across platforms is a pain point. A key example is the FlashAttention kernel (Dao et al., 2022), which is crucial for running modern Transformer models — the initial kernel released in 2022, five years after the Transformer was proposed; it took two more years from the release of NVIDIA Hopper GPUs to transfer the algorithm to the new hardware platform. We explore the question: *Can language models help write correct and optimized kernels?*

AI engineers use a rich set of information when developing kernels and it is not clear whether language models (LMs) can mimic the workflow. They use compiler feedback, profiling metrics, hardware-specific specs and instruction sets, and knowledge of hardware-efficiency techniques (e.g., tiling, fusion, recompute). They can use programming tools ranging from assembly (e.g., PTX as in DeepSeek-AI (2025)) to higher-level libraries (ThunderKittens (Spector et al., 2024), Triton (Tillet et al., 2019)). Compared to existing LM code generation workloads (Yang et al., 2024a), kernel writing requires a massive *amount* and *diversity* of information.

We first design an environment that reflects the typical AI engineer's workflow and supports providing LMs with this rich information. The environment should:

- **Automate** the AI engineer's workflow. The model should have full flexibility to decide which operators to optimize and how to optimize them.

- Support a **diverse** set of AI algorithms, programming languages, and hardware platforms.

- Make it **easy to evaluate** both performance and functional correctness of LM generations, ideally in a programmatic way. It should also capture profiling and execution information from generated kernels.

We introduce **KernelBench** to generate and evaluate kernels, which addresses the above considerations. KernelBench tests LM optimizations on three levels of AI workloads, each posing a different set of challenges:

---

[*]Equal contribution  [1]Department of Computer Science, Stanford University, Stanford, California, USA [2]Department of Computer Science, Princeton University, Princeton, New Jersey, USA. Correspondence to: Anne Ouyang <aco@stanford.edu>, Simon Guo <simonguo@stanford.edu>.

*Proceedings of the 42nd International Conference on Machine Learning*, Vancouver, Canada. PMLR 267, 2025. Copyright 2025 by the author(s).

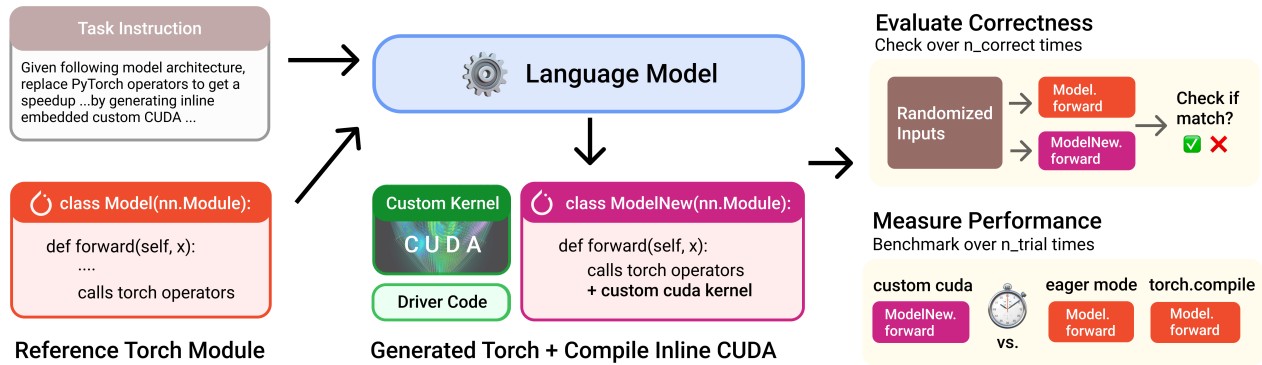

Figure 1. **KernelBench evaluates LMs' ability to generate performant GPU Kernels**. Overview of tasks in KernelBench: KernelBench tasks LMs with generating optimized CUDA kernels for a given target PyTorch model architecture and conducts automated evaluation.

1. **Individual operations:** We include various AI operators, including matrix multiplies, convolutions, activations, norms, and losses. While PyTorch already uses expert-optimized closed-source kernels, making this a potentially challenging baseline, it is valuable if LMs can generate open-source kernels for the operations.

2. **Sequence of operations:** We provide problems that contain 3-6 individual operations together (e.g. a mainloop operator like matmul followed by pointwise operators like ReLU and Bias). This enables evaluating the models' ability to fuse multiple operators.

3. **End-to-end architectures:** We select architectures from popular AI repositories on Github including `pytorch`, `huggingface/transformers`, and `huggingface/pytorch-image-models`. These architectures contain many operations.

Mimicking an AI researcher's workflow, the LM takes PyTorch reference code as input and outputs an optimized version of the code. Similar to the human kernel development process, our environment enables the LM to iterate with compiler and profiler feedback to refine performance. The LM is free to use any programming language and decide both *which parts* of the PyTorch code to optimize, and *how* to optimize them. Our pipeline allows us to feed diverse information to the LMs, including hardware-specific information, example kernels, and compiler/profiler feedback.

We observe that frontier and open-source models perform poorly out-of-the-box on KernelBench, with OpenAI-o1 and DeepSeek-R1 matching the PyTorch Eager baseline on $< 20\%$ of the tasks. These model-generated kernels greatly suffer from execution errors, functional correctness issues, and are unable to perform platform-specific optimizations.

To identify areas for improvement, we conduct a series of experiments and analysis, and find that:

1. *Writing functionally **correct** kernels remains challenging for models:* while models are able to fix execution failures through either reasoning or multiple attempts, they struggle to produce functionally correct code. Furthermore, we observe a trade-off between LMs attempting more complex optimizations / niche hardware instructions (e.g., tensor core `wmma`) and producing error-free kernels. We hypothesize this is due to CUDA being a low-resource language in open-source training data, only $0.073\%$ of popular code corpus The Stack v1.2 (Li et al., 2023; Kocetkov et al., 2022).

2. *Models demonstrate potential to produce **performant** kernels via optimizations:* We observe a few instances where LMs make algorithmic improvements – e.g., exploiting sparsity, operator fusion, and utilizing hardware features. We notice more of such instances when we explicitly condition the LM on hardware information (e.g., bandwidth and TFLOP specs) and demonstrations of hardware optimization techniques (e.g., tiling, fusion). While these capabilities remain nascent, LMs do **demonstrate potential** for generating performant kernels.

3. *Leveraging **feedback** is important for reducing execution errors and discovering faster solutions:* By providing execution results and profiler feedback to the LM in context, the kernel quality significantly improves after multiple refinements from $12\%$, $36\%$, and $12\%$ in fast$_1$ to $43\%$, $72\%$, and $18\%$ respectively.

Our findings highlight the technical challenges we need to solve in order to adopt LMs for kernel writing. These include but are not limited to: how to improve LM performance in a low-resource data regime, and how to select from the rich set of information we can provide to models. To address these challenges, we contribute (1) **an open-source framework** to study LM kernel generation with a comprehensive suite of evaluation problems and (2) **analysis of where current LMs stand** and how to realize a future of efficient kernels generated by models.

## 2. Related Works

**Kernel libraries and compilers.** We evaluate existing approaches for kernel programming along the dimensions of automation, breadth, and performance. Mainstream kernel programming libraries like cuDNN (NVIDIA, 2014), CUT-LASS (NVIDIA, 2017a), and Apple MLX (Apple, 2020) are hardware-specific and demand substantial engineering effort from human experts. Other libraries, like ThunderKittens (Spector et al., 2024) and Triton (Tillet et al., 2019), successfully help AI researchers write a breadth of fast and correct kernels (Arora et al., 2024; Yang & Zhang, 2024), but still require human programming effort. Compiler-based tools, like torch.compile (Paszke et al., 2019) and FlexAttention (Team PyTorch et al., 2024), automatically provide a narrow slice of optimizations. These methods have powerful guarantees in terms of generating provably-correct and robust kernels but are based on fixed operator-fusion or graph-transformation policies (Zheng et al., 2022; Shi et al., 2023; Zhu et al., 2022). Previous work on ML-based kernel generation like TVM (Chen et al., 2018) incorporate ML algorithms within a single, often sand boxed, component such as schedule generation. In contrast to these efforts, we ask if LMs can automatically generate performant kernels for a breadth of AI workloads.

**LLMs for performance-optimized code generation.** In the past year, there have been several efforts to build LMs that can automate algorithmic coding (Chen et al., 2021; Shi et al., 2024; Li et al., 2022), resolving GitHub issues (Yang et al., 2024a;b), and domain-specific coding (Yin et al., 2022; Lai et al., 2022). Other works have explored using LMs (Chen et al., 2023; Xia & Zhang, 2024) for program repair and debugging, through models leveraging feedback. While these works focus on producing correct and functional code, subsequent works have explored LMs' ability to produce solutions with better *algorithmic and asymptotic efficiency* (Nichols et al., 2024; Waghjale et al., 2024).

KernelBench focuses on *wall-clock efficiency*. For LMs to generate high-performance computing (HPC) code, which requires an understanding of the underlying hardware features and device instruction set, and common performance characteristics of parallel processors. Existing works in the space of HPC code generation have evaluated LM performance on translating arbitrary code samples from C++ to CUDA (TehraniJamsaz et al., 2024; Wen et al., 2022) or generating well-known, low-level kernels such as GEMMs (Valero-Lara et al., 2023; Wijk et al., 2024). KernelBench instead curates a set of 250 diverse kernels from real-world, modern deep learning workloads, many of which do not have existing human-written implementations — in other words, solving KernelBench tasks are immediately beneficial for real deep learning workloads. See Appendix J for a comparison with popular coding benchmarks.

## 3. KernelBench: A Framework for AI Kernel Generation

KernelBench is a new framework for evaluating the ability of language models to generate performant kernels for a breadth of AI workloads. In this section, we describe the task format, contents, and evaluation metric.

### 3.1. KernelBench Task Format

KernelBench contains 250 tasks representing a range of AI workloads, and is easily extensible to new workloads. The end-to-end specification for a task is illustrated in Figure 1 and described below.

**Task input:** Given an AI workload, the input to the task is a reference implementation written in PyTorch. Mimicking an AI researcher's workflow, the PyTorch code contains a class named `Model` derived from `torch.nn.Module()`, where the standard `__init__` and `forward()` functions (and any helper functions) are populated with the AI workload's PyTorch operations. We explore alternative forms of input specifications in Appendix M.

AI algorithms generally operate on large tensors of data. The optimal kernel for a workload depends on the size and data type (e.g., BF16, FP8) of the tensor. Therefore, each task additionally contains functions `get_inputs()` and `get_init_inputs()`, which specify the exact input tensors that the kernel needs to handle.

**Task output:** Given the input, the LM needs to output a new class named `ModelNew` derived from `torch.nn.Module()`, which contains custom optimizations. For example, the LM can incorporate in-line kernel calls during the `forward()` function using the CUDA-C extension in PyTorch.

In order to succeed, the LM needs to identify (1) which operations in the `Model` class would benefit most from optimizations, and (2) how to optimize those operations. The LM can use any hardware-efficiency techniques such as fusion and tiling or specialized instructions (e.g., tensor cores) and any programming library (e.g., PTX, CUDA, CUTLASS, Triton, ThunderKittens). We focus on CUDA in this paper, and explore programming libraries like Triton in Appendix O.

### 3.2. Task Selection

The 250 tasks in KernelBench are partitioned into three levels, based on the number of primitive operations, or PyTorch library functions, they contain:

- **Level 1 (100 tasks): Single primitive operation.** This level includes the foundational building blocks of AI (e.g. convolutions, matrix-vector and matrix-matrix multiplica-

tions, losses, activations, and layer normalizations).

Since PyTorch makes calls to several well-optimized and often closed-source kernels under-the-hood, it can be challenging for LMs to outperform the baseline for these primitive operations. However, if an LM succeeds, the open-source kernels could be an impactful alternative to the closed-source (e.g., CuBLAS (NVIDIA, 2023)) kernels.

- **Level 2 (100 tasks): Operator sequences.** This level includes AI workloads containing multiple primitive operations, which can be fused into a single kernel for improved performance (e.g., a combination of a convolution, ReLU, and bias).

  Since compiler-based tools such as the PyTorch compiler are effective at fusion, it can be challenging for LMs to outperform them. However, LMs may propose more complex algorithms compared to compiler rules.

- **Level 3 (50 tasks): Full ML architectures.** This level includes architectures that power popular AI models, such as AlexNet and MiniGPT, collected from popular PyTorch repositories on GitHub.

  Given the scale of modern models, it is critical to use kernels when running training and inference. Unfortunately, it has been difficult for the AI community to generate performant kernels. For instance, it took 5 years from the release of the Transformer architecture (Vaswani et al., 2017) to obtain performant kernels (Dao et al., 2022), let alone today's many new architectures. Peak performance kernels for these architectures require algorithmic modifications that are often beyond the scope of a compiler.

We reiterate that each task contains a meaningful set of AI primitive operations or architectures, such that LM success on the task can directly lead to real world impact. We provide further details on task definitions and breakdowns in Appendix K.

### 3.3. Metric Design

We describe the evaluation approach for KernelBench and how we compare the success of different LMs.

**Evaluation approach** KernelBench is an evaluation-only benchmark. We do not provide ground truth kernels for the tasks since we imagine users benchmarking on a variety of hardware platforms (including new platforms), input types, and workloads. However, by design, KernelBench is *automatically verifiable*. Given a task, we randomly generate input tensors of the prescribed shape and precision and collect the PyTorch `Model` output. We can evaluate whether LM generations are correct and fast as follows:

1. **Correctness** We compare the `Model` output to the LM-generated `ModelNew` output. We evaluate on 5 random inputs per problem (detailed in Appendix B).

2. **Performance** We compare the wall-clock execution time of `Model` against `ModelNew` using repeated trials to account for timing variations.

**Comparing LMs on KernelBench** Some LMs may generate a small number of correct kernels that are very fast, while other LMs generate a large number of correct kernels that are quite slow. Here, we explain our proposed unified metric for ranking LM quality on KernelBench. We elaborate on our process of score design in Appendix I.

To capture both axes of correctness and performance, we introduce a new metric called $\text{fast}_p$, which is defined as the fraction of tasks that are both correct and have a speedup (computed as the ratio of PyTorch wall-clock time to generated kernel time) greater than threshold $p$. Formally:

$$\text{fast}_p = \frac{1}{N} \sum_{i=1}^{N} \mathbb{1}(\text{correct}_i \wedge \{\text{speedup}_i > p\}),$$

where $\text{fast}_0$ is equivalent to the LM's correctness rate, as it measures the fraction of tasks for which the LM code is functionally correct regardless of its speed.

By adjusting the threshold parameter $p$, we enable evaluation of kernel performance at different speedup thresholds and capture the speedup distributions. For our evaluations, we focus on $p = 1$ as a starting point, with the possibility of increasing $p$ as future methods for kernel generation improve. Additionally, using $p < 1$ for training is valuable, since PyTorch relies on complex optimized kernels, and matching even a fraction of their performance is still considered beneficial.

## 4. KernelBench Baseline Evaluation

| **fast$_1$** over: | PyTorch Eager | | | torch.compile | | |
|---|---|---|---|---|---|---|
| KernelBench Level | 1 | 2 | 3 | 1 | 2 | 3 |
| GPT-4o | 4% | 5% | 0% | 18% | 4% | **4%** |
| OpenAI o1 | 10% | 24% | **12%** | 28% | 19% | **4%** |
| DeepSeek V3 | 6% | 4% | 8% | 20% | 2% | 2% |
| DeepSeek R1 | **12%** | **36%** | 2% | **38%** | **37%** | 2% |
| Claude 3.5 Sonnet | 10% | 7% | 2% | 29% | 2% | 2% |
| Llama 3.1-70B Inst. | 3% | 0% | 0% | 11% | 0% | 0% |
| Llama 3.1-405B Inst. | 3% | 0% | 2% | 16% | 0% | 0% |

*Table 1.* **KernelBench is a challenging benchmark for current LMs**. Here we present fast$_1$, i.e. the percentage of problems where the model-generated kernel is faster than the PyTorch Eager and `torch.compile` baseline (default config) on NVIDIA L40S. The `torch.compile` baseline runtime is sometimes slower than Torch Eager – this is due to reproducible runtime overhead (*not compile time*) that could be significant for small kernels in Level 1. We focus on PyTorch Eager for the rest of our analysis, but we elaborate on other baselines in Appendix B.

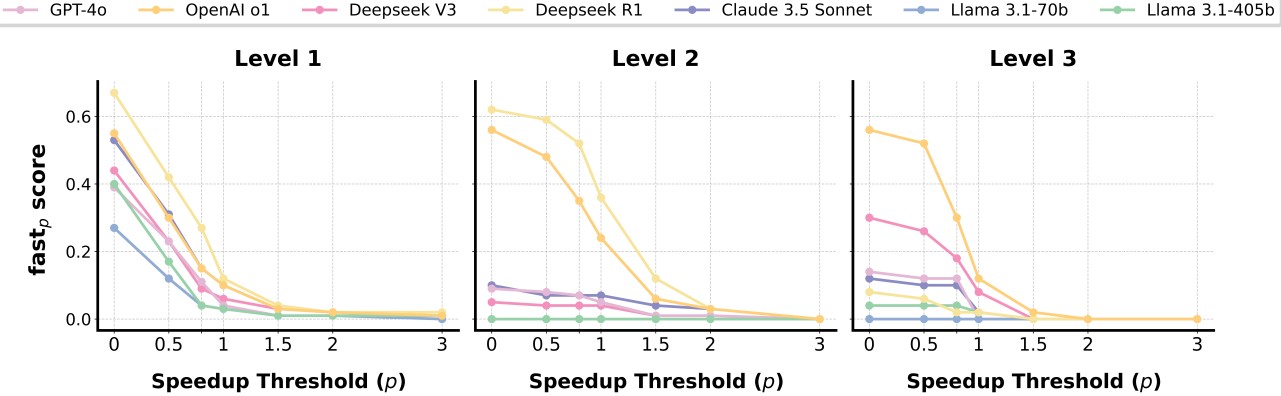

Figure 2. **Most LM-generated kernels are slow.** This figure shows the distribution of the $fast_p$ metric as the speedup threshold $p$ increases. $fast_0$ represents the number of *correct* kernels regardless of speed, and $fast_1$ those achieving at least $> 1\times$ speedup over PyTorch. Increasing $p$ increases difficulty.

In this section, we investigate how a range of LMs perform when evaluated off-the-shelf on KernelBench and explore their capabilities and failure modes.

### 4.1. One-shot Baseline

We evaluate LMs using a prompt that contains one example of a PyTorch `Model` input and `ModelNew` output, highlighting the task format. The example is simple, containing only an `add` operator (See Appendix C.1). Given this in-context example and the PyTorch task `Model` to optimize, the LM generates `ModelNew` via greedy decoding. We profile the generated code on an NVIDIA L40S GPU, and measure the $fast_p$ metric across all problems. Table 1 shows that the LM-generated kernels achieves a speedup over PyTorch Eager in fewer than 20% of tasks on average.

### 4.2. Correctness: Error Analysis

In Figure 3, we analyze the failure modes of LMs across problems. It can be seen that a large proportion of model-generated kernels are incorrect. To better understand where model-generated kernels fail, we break down their correctness issues into execution failures (CUDA/`nvcc` / Python compile-time errors, CUDA memory violations, and runtime errors) and correctness errors (output tensor shape and value mismatches). We observe that the reasoning LMs (o1, R1) produce fewer incorrect solutions ($< 55\%$) than other models ($> 70\%$) due to fewer execution failures. All LMs struggle with functional correctness to a similar degree.

### 4.3. Performance: Speedup Distribution

A key point of interest is whether the functionally correct LM-generated kernels outperform the PyTorch baseline. Figure 2 shows the distribution of $fast_p$ as $p$ varies, indicating the percentage of kernels that are $p$-times faster than the PyTorch Eager baseline (the top right of the plot is better). At $p = 1$, fewer than 15% of LM-generated

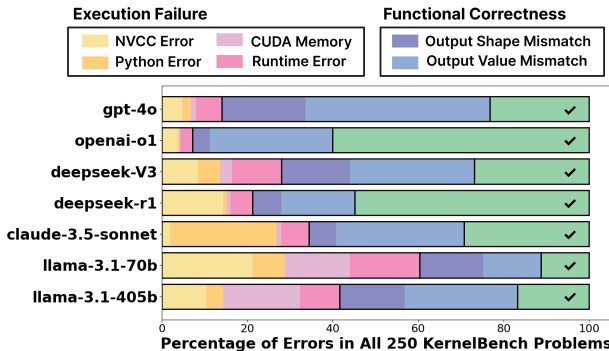

Figure 3. **We categorize failure modes of kernel code into execution failure and functional correctness**. For the one-shot baseline, reasoning models generate fewer kernels with execution failures, but all models struggle with functional correctness.

kernels outperform PyTorch across all KernelBench levels. Reasoning-based LMs generally outperform the other LMs in providing speedups. We dive into causes of performance degradation in Appendix N.

### 4.4. Performance Variations across Hardware

Our one-shot baseline makes no assumptions about the underlying hardware, so a natural question is how our analysis of the LM-generated kernels generalizes across various GPU types. Table 14 and Figure 8 show that kernels outperforming PyTorch Eager on NVIDIA L40S in Level 1 achieve similar speedups versus the baselines on other GPUs. However, on problems in Level 2, LMs exhibit larger variations in speedups across GPUs (Figure 9): DeepSeek R1-generated kernels achieve a $fast_1$ of 36% on NVIDIA L40S but 47% on NVIDIA A10G for Level 2. This suggests that one-shot LM-generated kernels may not generalize well across hardware. To generate more target-specific kernels, we explore further in Section 5.2 whether providing hardware-specific details in-context can improve LM performance.

Our analysis reveals that the best models available today struggle to generate correct kernels that outperform the baseline PyTorch speeds. LM-generated kernels frequently fail due to simple compiler and run-time errors. Furthermore, it is difficult for LMs to write kernels that perform well across hardware platforms given simple instructions.

# 5. Analysis of Model Capabilities

In the last section, we found that KernelBench is a challenging benchmark for today's models. In this section, we conduct case studies to explore opportunities for improvement in future models and AI systems.

## 5.1. Case Study: Leveraging the KernelBench Environment Feedback at Test-Time

As observed in Section 4.2, execution failures are the most frequent failure mode in LM-generated kernels. The environment provided by KernelBench allows us to collect rich signals, including compiler errors, correctness checks, and runtime profiling metrics, all of which can be fed back in to the LM to help it resolve kernel failures. To explore how well LMs can use this feedback, we evaluate and compare two baselines: (1) generating multiple parallel samples from the LM per KernelBench task and (2) sequentially generating kernels per KernelBench task by allowing the LM to iteratively refine using the execution feedback.

### 5.1.1. REPEATED SAMPLING

The KernelBench environment enables programmatic verification of LM-generated kernels, allowing us to collect and evaluate multiple LM generations *per task* (Brown et al., 2024; Li et al., 2022; Grubisic et al., 2024). We evaluate this *repeated sampling* approach using $\text{fast}_p@k$, which measures the percentage of tasks where the model generated **at least one functionally correct kernel that is $p$ times faster than PyTorch Eager when drawing $k$ samples**.

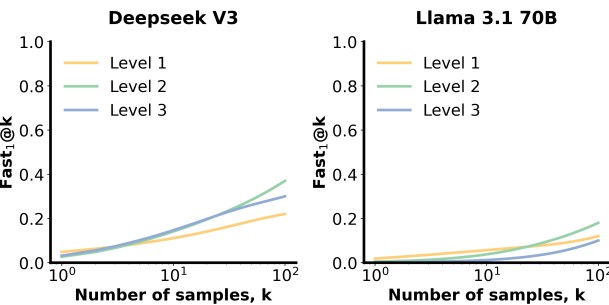

*Figure 4.* **Repeated sampling helps discover more correct and performant kernels**. As the number of parallel samples $k$ increases (up to 100), $\text{fast}_1@k$ improves for both DeepSeek-V3 and Llama 3.1-70B Instruct across all 3 KernelBench levels.

**Repeated sampling helps LMs discover more fast and correct solutions.** Figure 4 shows that repeated sampling with high temperature improves $\text{fast}_1$ as $k$ increases across all three levels with both DeepSeek-V3 and Llama 3.1 70B. Notably, on Level 2, DeepSeek-V3 reaches a $\text{fast}_1$ of 37% with $k = 100$ samples, compared to just 4% in the one-shot baseline.

Examining the samples, we find that high-temperature sampling helps explore the solution space, increasing the chances of generating error-free kernels with better optimizations. However, if a model has a very low inherent probability of solving a task, simply increasing the sampling budget has limited impact. For example, DeepSeek-V3 was never able to generate any correct solution for a group of 34 convolution variants in Level 1, even when attempting with 100 samples.

### 5.1.2. ITERATIVE REFINEMENT OF GENERATIONS

The KernelBench environment is well-suited for collecting compiler feedback, execution errors, and timing analysis using tools like the PyTorch profiler as ground-truth signals. We investigate whether leveraging this feedback can help LMs to iteratively refine their generations.

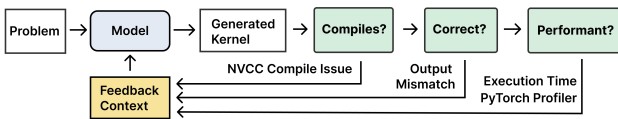

*Figure 5.* **The KernelBench framework enables models to receive and leverage feedback during iterative refinement.** These ground-truth signals include NVCC compiler error messages, execution statistics (correctness and wall clock time), and the PyTorch profiler (operator timing breakdown).

We provide feedback to the model after each generation in a multi-turn process: after the initial generation, we provide the model with its previous generation $G$, as well as compiler/execution feedback $E$ and/or profiler output $P$ over its current generation. We define each generation and subsequent feedback as a *turn*, and run this **Iterative Refinement** process over $N$ turns. For each turn, we measure $\text{fast}_p@N$, which is the percentage of tasks where the model generated **at least one** functionally correct kernel that is $p$ times faster than PyTorch Eager by turn $N$.

**Leveraging execution feedback helps reduce errors and improves overall speedups over time.** We examine the $\text{fast}_1$ behavior at turn $N = 10$ in Table 2 and find that iterative refinement consistently improves performance across models and levels of KernelBench. DeepSeek-R1 on Level 2 results in the most notable improvement, where the combination of execution feedback $E$ and profiler feedback $P$

| Method | Level 1 | | | Level 2 | | | Level 3 | | |
|---|---|---|---|---|---|---|---|---|---|
| | Llama 3.1 70B | DeepSeek V3 | DeepSeek R1 | Llama 3.1 70B | DeepSeek V3 | DeepSeek R1 | Llama 3.1 70B | DeepSeek V3 | DeepSeek R1 |
| Single Attempt (Baseline) | 3% | 6% | 12% | 0% | 4% | 36% | 0% | 8% | 2% |
| Repeated Sampling (@10) | 5% | 11% | N/A | 3% | **14%** | N/A | 1% | 14% | N/A |
| Iterative Refinement w G | **9%** | 9% | 18% | 0% | 7% | 44% | 0% | 14% | 4% |
| Iterative Refinement w G+E | 5% | 13% | 41% | **5%** | 5% | 62% | **8%** | **22%** | 12% |
| Iterative Refinement w G+E+P | 7% | **19%** | **43%** | 4% | 6% | **72%** | 2% | 14% | **18%** |

*Table 2.* **Both repeated sampling and iterative improvement enable models to generate more correct and fast kernels compared to baseline:** Here we present the percentage of problems where the LM-generated kernel is correct and faster than baseline Torch Eager (Fast$_1$ in %) for the two test-time methods, both with the same sample budget of 10 calls. We further compare performance within iterative refinement achieved when leveraging previous Generation $G$, Execution Result $E$, and Timing Profiles $P$. Note we do not repeatedly sample DeepSeek R1, as its API endpoint does not provide a temperature parameter.

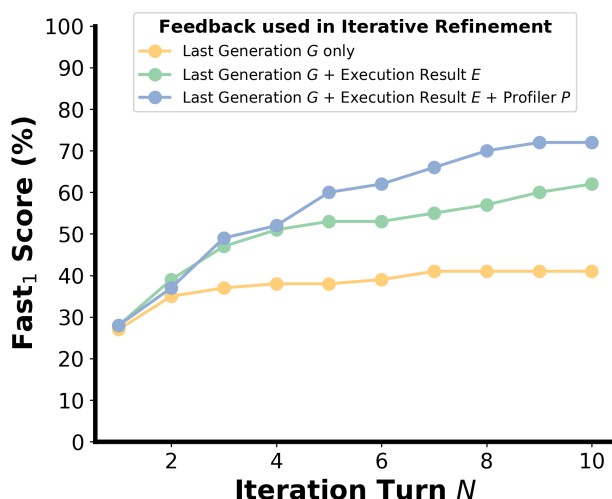

*Figure 6.* **Iterative refinement with execution feedback $E$ and profiling information $P$ enable models to improve kernel generations over turns,** as shown in the fast$_1$@$N$ trajectory of DeepSeek-R1 on Level 2. The percentage of problems where the best generated kernel up to turn $N$ is correct and faster than PyTorch Eager consistently increases as we increase number of turns.

boosts fast$_1$ from 36% to 72% (shown in Figure 6).

Furthermore, by examining iterative refinement trajectories, we find that models self-correct more effectively with execution feedback $E$, fixing issues especially related to execution errors. DeepSeek-R1 on Level 1 and 2 can generate a functional kernel on >90% of the tasks within 10 turns of refinement (Table 9). However, the remaining incorrect kernels almost always fail due to functional incorrectness, likely because correctness feedback is less granular than execution failure messages. We include successful and failed examples of iterative refinement trajectories in Appendix D.4.

### 5.1.3. COMPARING REPEATED SAMPLING AND ITERATIVE REFINEMENT

In Table 2, we compare repeated sampling and iterative refinement given a fixed budget of 10 inference calls. Both methods provide meaningful improvements over the one-shot baseline, with iterative refinement being more effective in 5 of the 6 cases. However, ultimately we find that the effectiveness of the test-time methods is inherently dependent on the quality of the base model. For instance, with repeated sampling, DeepSeek-V3 consistently outperforms Llama-3.1 70B across all three levels. Similarly, with iterative refinement, DeepSeek-R1 consistently improves using feedback $E$ and $P$, while DeepSeek-V3 and Llama-3.1 70B does not always benefit from having such information.

### 5.2. Case Study: Generating Hardware-Efficient Kernels via Hardware Knowledge

It is clear that LMs demonstrate limited success at generating hardware-efficient kernels. This is likely due to the scarcity of kernel code in the training data and the fact that the optimal kernel may need to change depending on the hardware platform-specific properties, as discussed in Section 4.4. In this case study, we explore providing 1) in-context examples of best-practices for kernel engineering and 2) in-context hardware specification details.

### 5.2.1. HARDWARE-AWARE IN-CONTEXT EXAMPLES

Well-written kernels often use techniques such as fusion, tiling, recompute, and asynchrony to maximize performance. We find that most of the one-shot generated kernels evaluated in Section 4 often do not use these techniques. Here, we explore whether providing explicit in-context examples that use these techniques can help the LMs improve their performance on KernelBench. Specifically, we include three in-context examples: GeLU (Hendrycks & Gimpel, 2023) using operator fusion, matrix multiplication using tiling (Mills, 2024), and a minimal Flash-Attention (Dao et al.,

2022; Kim, 2024) kernel that demonstrates shared memory I/O management.

**In-context examples degrade the LM's *overall* fast$_1$ score since LMs attempt more aggressive optimization strategies, but result in more execution failures.** OpenAI o1's generations are 25% longer on average using the few-shot examples, compared to the generations produced by Section 4 baseline. However, among the correct solutions, the LMs apply interesting optimizations: we find that on 77% of GEMM variants in KernelBench Level 1, o1 applies tiling and improves speed over the one-shot baseline (although remains slower than PyTorch Eager due to the lack of tensor core utilization). On Level 2, o1 applies aggressive shared memory I/O management on 11 problems, and is able to outperform PyTorch Eager (See Appendix F).

### 5.2.2. SPECIFYING HARDWARE INFORMATION

As discussed in Section 4.4, kernel performance varies depending on the hardware platform. For instance, FlashAttention-2 (Dao, 2024) degrades 47% in hardware utilization going from the NVIDIA A100 to H100 GPU. FlashAttention-3 (Shah et al., 2024), an entirely different algorithm, was written for the H100. In this study, we explore whether LMs can use (1) hardware specifications such as the GPU type (H100, A100, etc.), memory sizes, bandwidths, TFLOPS and (2) hardware knowledge (e.g. definitions of threads, warps, thread-blocks, streaming multiprocessors) in-context to generate improved kernels.

**Models rarely generate kernels that are optimized for the underlying hardware, highlighting room for improvement for future models.** Certain generations of GPUs (e.g. H100) feature a variety of new hardware units and instructions from their predecessors. Providing hardware information does not significantly impact the outputs of Llama 3.1 70B or DeepSeek-V3. Interestingly, we find that a subset of OpenAI o1 and DeepSeek-R1 generated kernels use hardware-specific instructions and optimizations. R1 attempts to generate warp matrix multiply-accumulate (`wmma`) instructions (Figure 10) for approximately 50% of the Level 1 matrix multiplication problems, although most fail to compile. Among the functionally correct generations, R1 and o1 produce 1-3 outliers per level that are $\geq 2\times$ faster than the Section 4 baselines. Overall, we find that modern LMs are better at adjusting their approaches when provided with few-shot examples in Section 5.2.1 than when conditioned on hardware information. Even when explicitly guiding R1 to use architecture-specific instructions (e.g. `wmma` and `memcpy_async`) through in-context examples, the model struggled to apply these instructions correctly on simple Level 1 matrix multiplication problems. See Appendix G for more details.

## 6. Discussion

In this section, we discuss qualitative examples of LM generations, and discuss opportunities for improvement.

### 6.1. Deep Dive Into Interesting Kernels

Here, we discuss a few surprising LM-generated kernels that demonstrate significant speedups over the PyTorch baseline. See detailed examples in Appendix D.

**Operator fusion** GPUs have small amounts of fast-access memory and large amounts of slow-access memory. Fusion can help reduce slow-access I/O costs by performing multiple operations on data that has been loaded into fast-access memory. We find that LMs optimize the GELU (2.9x) and Softsign (1.3x) operators by fusing their computations into a single kernel. LMs generated a kernel that fuses multiple foundational operators – matrix multiplication with division, summation, and scaling – giving a 2.6x speedup. Overall, LMs leave many fusion opportunities on the table, we provide additional analysis on kernel fusion behavior in Appendix L.

**Memory hierarchy** Effective kernels explicitly manage how the limited amounts of fast-access memory (e.g., shared and register memory) gets utilized. In the generated kernels, we found kernels that uses GPU shared memory – cosine similarity (2.8x) and triplet margin loss (2.0x) – to achieve speedups. We did not find successful usages of tensor core instructions, which are crucial for AI performance.

**Algorithmic optimizations** Kernels can require algorithmic modifications to better utilize the hardware features. We found one interesting generation for the problem of performing a multiplication between a dense and diagonal matrix, where the kernel scales each row (or column), rather than loading the zero-entries of the diagonal matrix, yielding a 13x speedup over PyTorch Eager.

### 6.2. Opportunities for Future Work

We show that there is significant room for improvement on KernelBench given the currently available models. First, future work can explore the development of advanced fine-tuning and reasoning techniques, including agentic workflows. Since CUDA is a low-resource language, it would be valuable for future work to open-source more high quality data. Second, LMs generate raw CUDA code in our experiments. However, future work can explore whether generating code using alternative programming abstractions (e.g., provided in ThunderKittens, CUTLASS, Triton, and others) can simplify the generation problem, for instance by making it easier for LMs to leverage tensor core instructions. Third, our evaluation has also been limited to GPUs so far and future work can expand to other hardware accelerators.

### 6.3. Conclusion

Our contributions are: (1) We present KernelBench, a framework that lays the groundwork for LM-driven kernel optimization, and (2) We evaluate a diverse set of models and approaches, analyzing their strengths and limitations, and providing insights for opportunities to enhance kernel generation using AI models.

Overall, while most benchmarks eventually saturate, KernelBench is designed to dynamically evolve as new AI workloads arise. Our $fast_p$ metric can be adapted over time to measure the speedup threshold ($p$) over increasingly advanced baselines (i.e., beyond the PyTorch baseline used in our work). Since PyTorch is cross-hardware platform compatible, the PyTorch-based tasks in KernelBench tasks can be evaluated on every *new hardware platform* release.

Finally, unlike many benchmarks, success on KernelBench directly maps to production value and real-world impacts (lowering costs and reducing energy consumption at scale). These properties ensure that KernelBench will remain valuable in the ever-evolving AI landscape.

### Impact Statement

Optimized GPU kernels can lead to significant energy savings in large-scale machine learning workloads, reducing both computational costs and environmental impact. By providing a framework for AI-assisted performance tuning, KernelBench contributes to more energy-efficient AI systems, aligning with global efforts to reduce the carbon footprint of computing infrastructure.

KernelBench does not involve human studies or collect user data, eliminating privacy concerns. It also avoids proprietary or private code, relying solely on publicly available Github repositories.

### Acknowledgements

We thank Aaryan Singhal, AJ Root, Allen Nie, Anjiang Wei, Benjamin Spector, Bilal Khan, Bradley Brown, Daniel Y. Fu, Dylan Patel, Fredrik Kjolstad, Genghan Zhang, Hieu Pham, Hugh Leather, John Yang, Jon Saad-Falcon, Jordan Juravsky, Marcel Rød, Mark Saroufim, Michael Zhang, Minkai Xu, Ofir Press, Ryan Ehrlich, Sahan Paliskara, Sahil Jain, Shicheng (George) Liu, Suhas Kotha, Tatsunori Hashimoto, Vikram Sharma Mailthody, and Yangjun Ruan for insightful discussions and constructive feedback in shaping this work. We are also grateful to PyTorch, Prime Intellect, and Modal for supporting this work.

We gratefully acknowledge the support of Google DeepMind, Google Research, Stanford HAI, and members of the Stanford SEAMS project: IBM and Felicis; NIH under No. U54EB020405 (Mobilize), NSF under Nos. CCF2247015 (Hardware-Aware), CCF1763315 (Beyond Sparsity), CCF1563078 (Volume to Velocity), and 1937301 (RTML); US DEVCOM ARL under Nos. W911NF-23-2-0184 (Long-context) and W911NF-21-2-0251 (Interactive Human-AI Teaming); ONR under Nos. N000142312633 (Deep Signal Processing); Stanford HAI under No. 247183; NXP, Xilinx, LETI-CEA, Intel, IBM, Microsoft, NEC, Toshiba, TSMC, ARM, Hitachi, BASF, Accenture, Ericsson, Qualcomm, Analog Devices, Google Cloud, Salesforce, Total, the HAI-GCP Cloud Credits for Research program, the Stanford Data Science Initiative (SDSI), and members of the Stanford DAWN project: Meta, Google, and VMWare. The U.S. Government is authorized to reproduce and distribute reprints for Governmental purposes notwithstanding any copyright notation thereon. Any opinions, findings, and conclusions or recommendations expressed in this material are those of the authors and do not necessarily reflect the views, policies, or endorsements, either expressed or implied, of NIH, ONR, or the U.S. Government.

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

# A. KernelBench Task Example

Here we provide an example task from KernelBench. Each task is wrapped in a class named `Model`. A task contains two key functions in the `Model` class, `__init__` and `forward`; helper functions are included if necessary. We fix the shape of inputs and vary the numerical values through randomly generated tensors. We provide two functions, `get_inputs` and `get_init_inputs`, for generating random parameters for initializing the model and running a forward pass, respectively.

**PyTorch Reference Architecture:**

```
1  import torch
2  import torch.nn as nn
3
4  class Model(nn.Module):
5      """
6      Simple model that performs a single matrix multiplication (C = A * B) with a large K
           ↪ dimension
7      """
8      def __init__(self):
9          super(Model, self).__init__()
10
11     def forward(self, A: torch.Tensor, B: torch.Tensor) -> torch.Tensor:
12         """
13         Performs matrix multiplication of A and B.
14
15         Args:
16             A: Input tensor of shape (M, K)
17             B: Input tensor of shape (K, N)
18
19         Returns:
20             Output tensor of shape (M, N)
21         """
22         return torch.matmul(A, B)
23
24 M = 256
25 N = 256
26 K = 131072
27
28 def get_inputs():
29     A = torch.randn(M, K)
30     B = torch.randn(K, N)
31     return [A, B]
32
33 def get_init_inputs():
34     return []  # No special initialization inputs needed
```

**We include an example output from a model below that attempts to optimize the task specification above.** The model is not only responsible for generating the kernel code, but is also responsible for generating the peripheral code to integrate the kernel into the PyTorch framework. The evaluation framework will run the forward pass of the model as if it is a PyTorch operator, so a common approach is to inline the CUDA code.

```
1   import torch
2   import torch.nn as nn
3   from torch.utils.cpp_extension import load_inline
4
5   # Define the custom CUDA kernel for matrix multiplication
6   matmul_source = """
7   #include <torch/extension.h>
8   #include <cuda_runtime.h>
9
10  __global__ void matmul_kernel(const float* A, const float* B, float* C, int M, int N, int
        ↪ K) {
11      int row = blockIdx.y * blockDim.y + threadIdx.y;
12      int col = blockIdx.x * blockDim.x + threadIdx.x;
13
14      if (row < M && col < N) {
15          float sum = 0.0f;
16          for (int k = 0; k < K; ++k) {
17              sum += A[row * K + k] * B[k * N + col];
18          }
19          C[row * N + col] = sum;
20      }
21  }
22
23  torch::Tensor matmul_cuda(torch::Tensor A, torch::Tensor B) {
24      int M = A.size(0);
25      int N = B.size(1);
26      int K = A.size(1);
27
28      auto C = torch::zeros({M, N}, A.options());
29
30      dim3 threadsPerBlock(16, 16);
31      dim3 numBlocks((N + threadsPerBlock.x - 1) / threadsPerBlock.x, (M + threadsPerBlock.y
          ↪  - 1) / threadsPerBlock.y);
32
33      matmul_kernel<<<numBlocks, threadsPerBlock>>>(A.data_ptr<float>(), B.data_ptr<float>()
          ↪ , C.data_ptr<float>(), M, N, K);
34
35      return C;
36  }
37  """
38
39  matmul_cpp_source = (
40      "torch::Tensor matmul_cuda(torch::Tensor A, torch::Tensor B);"
41  )
42
43  # Compile the inline CUDA code for matrix multiplication
44  matmul = load_inline(
45      name="matmul",
46      cpp_sources=matmul_cpp_source,
47      cuda_sources=matmul_source,
48      functions=["matmul_cuda"],
49      verbose=True,
50      extra_cflags=[""],
51      extra_ldflags=[""],
52  )
53
54
55  class ModelNew(nn.Module):
56      def __init__(self):
57          super(ModelNew, self).__init__()
58          self.matmul = matmul
59
60      def forward(self, A: torch.Tensor, B: torch.Tensor) -> torch.Tensor:
61          return self.matmul.matmul_cuda(A, B)
```

# B. Evaluation Methodology and Baselines

All evaluations are conducted on a bare-metal NVIDIA L40S GPU with Ada Lovelace architecture unless otherwise stated (such as the device generalization experiments in Section 4.4 and the hardware case study in 5.2). The NVIDIA L40S has 48 GB of HBM memory and operates at 300W. Our environment uses Python `3.10`, PyTorch `2.5.0+cu124`, and CUDA `12.4`, which is also where our PyTorch Eager and `torch.compile` baselines are derived from.

## B.1. Kernel Evaluation Setup

Recall the KernelBench task entails a PyTorch reference module `Model` as baseline, and model-generated PyTorch architecture `ModelNew` with custom inline CUDA kernel.

For **correctness**, we set *num_correctness* to 5, where we check equivalence of output between reference architecture `Model` and generated architecture with custom kernel `ModelNew` with 5 randomized inputs. We elaborate on our choice in B.2.

For **performance**, we measure the wall-clock execution time of `nn.module.forward` for both `Model` and `ModelNew`. We ensure only one kernel is being evaluated (no other CUDA process) on current GPU. We warm up for 3 iterations and then set *num_profile* to 100 times which measures the elapsed execution time signaled between CUDA events `torch.cuda.Event`. We take the mean of the 100 trials, and also note its max, min, and standard deviation. While the wall clock time might vary for every trial, we note our coefficient of variation (CV): std/mean is consistently $< 3\%$, we use the mean of both measured wall clock time for comparisons.

To compute the speedup of generated architecture over baseline architecture for individual problems, we use the mean for both $speedup = T_{Model}/T_{ModelNew}$. For example, if $T_{Model} = 2$ ms and $T_{ModelNew} = 1$ ms, we have a 2x speedup with the newly generated kernel. We compare this speedup with our speedup threshold parameter $p$ (as explained in section 3.3) to compute $fast_p$ scores.

## B.2. Correctness Analysis Varying Number of Randomly Generated Inputs

Checking equivalence of programs in a formal sense is undecidable. "The Halting Problem" (Turing, 1936) states that it is impossible to decide, in general, whether a given program will terminate for every possible input. This problem naturally extends to checking equivalence because in order to check whether two programs are equivalent, it is necessary to check their behavior for all inputs, including cases where one or both programs may not terminate. Since determining whether a program halts on a given input is undecidable (the Halting Problem), checking equivalence also becomes undecidable.

Approximate or heuristic methods are often used in practice for checking program equivalence. Random testing is the most common practical approach, where the program is run with sets of randomly chosen inputs, and their outputs are compared. Random testing is particularly effective for AI kernels, where control flow is simpler and the focus is primarily on numerical correctness. By using diverse inputs, it can uncover errors in computations or memory handling with high probability.

We use five sets of random inputs for correctness, which is a good tradeoff between the ability to catch errors and efficiency. In an experiment with 100 generated kernels, the results were as follows: 50 kernels were correct (all 5/5 and 100/100), 19 had output value mismatches (19 0/5 and 0/100), 4 had output shape mismatches, 10 encountered runtime errors, and 17 had compilation errors. Notably, the 0/5 and 0/100 failures indicate that no partial correctness was observed.

## B.3. Distribution of Model Performance for One-Shot Baseline

Here we examine the quality of (functionally correct) kernel generations across a wide variety of models. Figure 7 shows the distribution of speedups for various kernels across different levels and models. The median speedup for both Level 1 and Level 3 are less than 1, and the median speedup for Level 2 is only slightly above one. Level 1 has the most significant outliers, in one case showing a speedup greater than 10. We explored some of these outlier cases in greater detail in Section 6.

**Reasoning-optimized models (OpenAI-o1 and DeepSeek-R1) perform the best of out-of-the-box across all levels.** These models demonstrate superior kernel generation capabilities, particularly excelling at Level 2 tasks (which mainly involve kernel fusion). In contrast, Llama 3.1 models (both 405B and 70B) perform poorly regardless of model size, suggesting that larger models do not necessarily guarantee better results for this task. DeepSeek-R1, while strong at Level 1 and 2, suffers significantly at Level 3, often generating incorrect kernels.

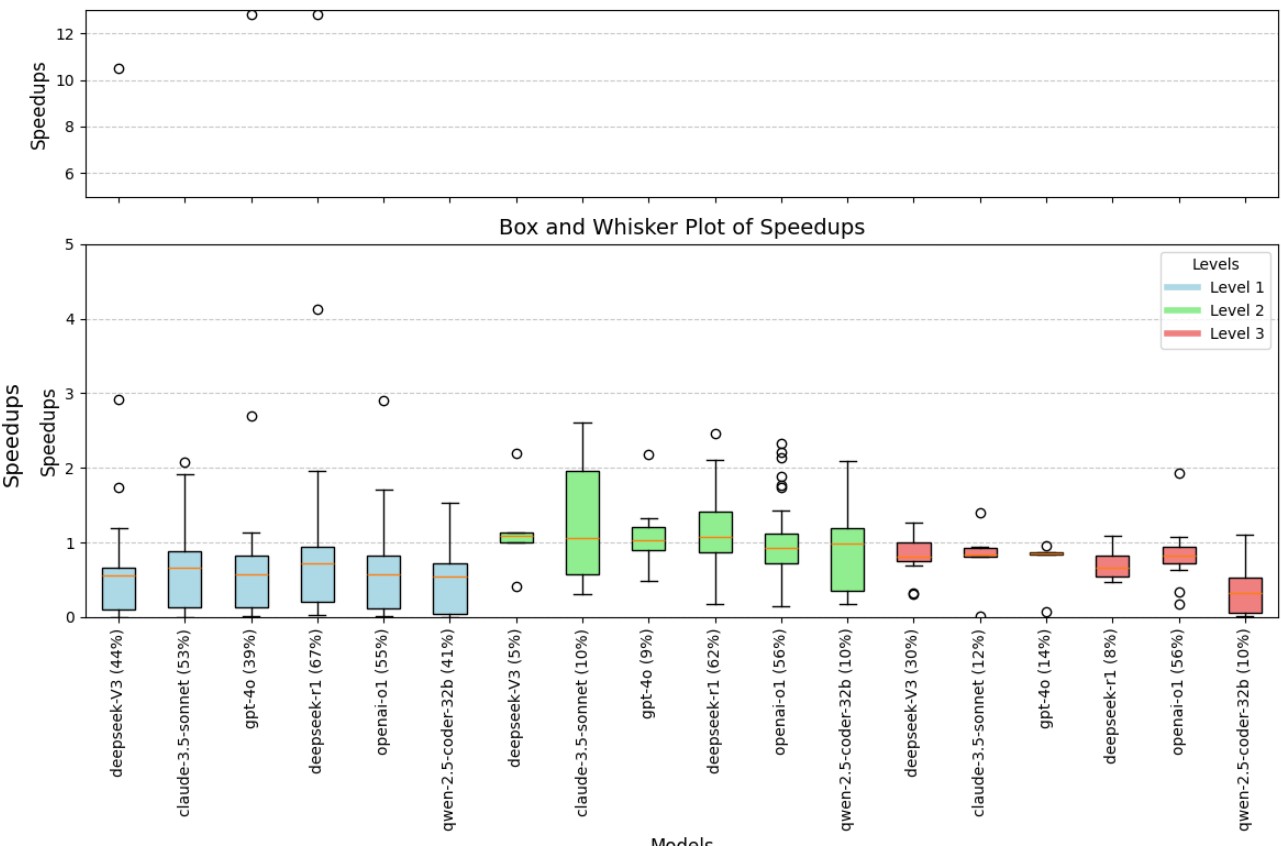

*Figure 7.* A box and whisker plot of the speedup relative to Torch Eager of (correct) kernels generated by various models in the one-shot baseline setting. We also write the percentage of correctly generated kernels next to the model name. We observe that among most models, the median speedup for correctly generated kernels is below 1.

## B.4. PyTorch Baselines

PyTorch offers two common execution modes: Eager and `torch.compile`. Aside from the results shown in Table 1, all performance analysis is evaluated against PyTorch Eager.

**PyTorch Eager** is the default execution mode of PyTorch, which dynamically executes computation by invoking calls to highly optimized closed-source kernels.

**PyTorch Compile** or `torch.compile` uses rule-based heuristics over the underlying computation graph during an initial compilation phase and invokes various backends to perform optimizations like kernel fusion and graph transformations. In Table 1, our performance baseline for `torch.compile` assumes the default configuration using PyTorch Inductor in default mode. Furthermore, we **exclude** the `torch.compile` compile time in our timing analysis, as we are only interested in the raw runtime behavior. `torch.compile` features multiple other backends and configurations, which we describe in Table 3.

We observe that the `torch.compile` baseline runtime is generally faster on Level 2 and 3 of KernelBench reference problems compared to PyTorch Eager, mostly due to the availability of graph-level optimizations like operator fusion. However, on Level 1 problems, `torch.compile` can exhibit higher runtimes than PyTorch Eager, which can be attribute to empirically-reproducible runtime overhead for `torch.compile` (*not compile time*) that is significant for small kernels.

**Other `torch.compile` backends**. In Table 4, we show more one-shot baseline results for fast$_1$ against some of the other `torch.compile` baselines. We note on some other configurations fast$_1$ drops especially for Level 2, as the `torch.compile` backends apply more aggressive optimization (at the cost of extra compile-time overhead, which we do not measure). Due to the variability of `torch.compile` across configurations, we focus our analysis on PyTorch Eager.

| Configuration | Backend | Mode | Description |
|---|---|---|---|
| PyTorch (Eager) | - | - | Standard PyTorch eager execution |
| Torch Compile | inductor | default | Default `torch.compile` behavior |
| Torch Compile | inductor | reduce-overhead | Optimized for reduced overhead |
| Torch Compile | inductor | max-autotune | Maximum autotuning enabled |
| Torch Compile | inductor | max-autotune-no-cudagraphs | Maximum autotuning without CUDA graphs |
| Torch Compile | cudagraphs | - | CUDA graphs with AOT Autograd |

*Table 3.* Configurations and modes for PyTorch execution and optimization backends.

| $fast_1$ over: | torch.compile default | | | cudagraphs | | | max-autotune | | | max-autotune no-cudagraphs | | | reduce-overhead | | |
|---|---|---|---|---|---|---|---|---|---|---|---|---|---|---|---|
| KernelBench Level | 1 | 2 | 3 | 1 | 2 | 3 | 1 | 2 | 3 | 1 | 2 | 3 | 1 | 2 | 3 |
| Claude 3.5 Sonnet | 29% | 2% | 2% | 31% | 7% | 2% | 31% | 2% | 0% | 29% | 2% | 2% | 31% | 2% | 0% |
| DeepSeek V3 | 20% | 2% | 2% | 21% | 4% | 20% | 21% | 2% | 2% | 20% | 2% | 2% | 21% | 2% | 0% |
| DeepSeek R1 | **38%** | **37%** | 2% | **42%** | **52%** | 0% | **42%** | 29% | 0% | **38%** | **32%** | 4% | **42%** | 28% | 0% |
| GPT-4o | 18% | 4% | **4%** | 22% | 6% | 6% | 21% | 4% | 2% | 18% | 3% | 4% | 21% | 4% | 0% |
| Llama 3.1-70B Inst. | 11% | 0% | 0% | 12% | 0% | 0% | 12% | 0% | 0% | 11% | 0% | 0% | 12% | 0% | 0% |
| Llama 3.1-405B Inst. | 16% | 0% | 0% | 16% | 0% | 4% | 16% | 0% | 0% | 16% | 0% | 0% | 16% | 0% | 0% |
| OpenAI O1 | 28% | 19% | **4%** | 33% | 37% | **26%** | 34% | 8% | **4%** | 30% | 19% | **6%** | 34% | 8% | **2%** |

*Table 4.* We compare KernelBench `torch.compile` baseline runtime across various configurations, all measured on NVIDIA L40S, in addition to what is showed in Table 1.

## C. Experiment Prompting Details

We provide details for the prompting strategies and associated sampling strategies used in Section 4 and Section 5.

### C.1. One-shot Baseline Prompt

For the one-shot baseline as shown in Section 4.1, we want to examine each model's out-of-the-box ability to generate kernels by providing the minimum set of information while ensuring the instructions and output format are clear. We query each model with the following prompt and a pair of in-context `add` examples (the PyTorch reference `add` and its CUDA kernel counterpart using inline compilation) to provide the output format. We sample the model with greedy decoding to ensure deterministic output, which is setting temperature $= 0$.

```
1  You write custom CUDA kernels to replace the pytorch operators in the given architecture
2  to get speedups.
3
4  You have complete freedom to choose the set of operators you want to replace. You may
5  make the decision to replace some operators with custom CUDA kernels and leave others
6  unchanged. You may replace multiple operators with custom implementations, consider
7  operator fusion opportunities (combining multiple operators into a single kernel, for
8  example, combining matmul+relu), or algorithmic changes (such as online softmax). You are
9  only limited by your imagination.
10
11 Here\'s an example to show you the syntax of inline embedding custom CUDA operators in
12 torch: The example given architecture is:
13 ```
14 import torch
15 import torch.nn as nn
16 import torch.nn.functional as F
17
18
19 class Model(nn.Module):
20     def __init__(self) -> None:
21         super().__init__()
22
```

```
23        def forward(self, a, b):
24            return a + b
25
26
27    def get_inputs():
28        # randomly generate input tensors based on the model architecture
29        a = torch.randn(1, 128).cuda()
30        b = torch.randn(1, 128).cuda()
31        return [a, b]
32
33
34    def get_init_inputs():
35        # randomly generate tensors required for initialization based on the model
              ↪ architecture
36        return []
37    ```
38
39    The example new arch with custom CUDA kernels looks like this:
40    ```
41    import torch
42    import torch.nn as nn
43    import torch.nn.functional as F
44    from torch.utils.cpp_extension import load_inline
45
46    # Define the custom CUDA kernel for element-wise addition
47    elementwise_add_source = """
48    #include <torch/extension.h>
49    #include <cuda_runtime.h>
50
51    __global__ void elementwise_add_kernel(const float* a, const float* b, float* out, int
         ↪ size) {
52        int idx = blockIdx.x * blockDim.x + threadIdx.x;
53        if (idx < size) {
54            out[idx] = a[idx] + b[idx];
55        }
56    }
57
58    torch::Tensor elementwise_add_cuda(torch::Tensor a, torch::Tensor b) {
59        auto size = a.numel();
60        auto out = torch::zeros_like(a);
61
62        const int block_size = 256;
63        const int num_blocks = (size + block_size - 1) / block_size;
64
65        elementwise_add_kernel<<<num_blocks, block_size>>>(a.data_ptr<float>(), b.data_ptr<
             ↪ float>(), out.data_ptr<float>(), size);
66
67        return out;
68    }
69    """
70
71    elementwise_add_cpp_source = "torch::Tensor elementwise_add_cuda(torch::Tensor a, torch::
         ↪ Tensor b);"
72
73    # Compile the inline CUDA code for element-wise addition
74    elementwise_add = load_inline(
75        name='elementwise_add',
76        cpp_sources=elementwise_add_cpp_source,
77        cuda_sources=elementwise_add_source,
78        functions=['elementwise_add_cuda'],
79        verbose=True,
80        extra_cflags=[''],
81        extra_ldflags=['']
82    )
83
```

```
84  class ModelNew(nn.Module):
85      def __init__(self) -> None:
86          super().__init__()
87          self.elementwise_add = elementwise_add
88
89      def forward(self, a, b):
90          return self.elementwise_add.elementwise_add_cuda(a, b)
91  ```
92
93  You are given the following architecture:
94
95  <PyTorch reference architecture for specific KernelBench Problem>
96
97  Optimize the architecture named Model with custom CUDA operators! Name your optimized
98  output architecture ModelNew. Output the new code in codeblocks. Please generate real
99  code, NOT pseudocode, make sure the code compiles and is fully functional. Just output
100 the new model code, no other text, and NO testing code!
```

### C.2. Repeated Sampling Prompts

For repeated sampling, we use the same prompt that we used for the one-shot baseline in Appendix C.1. We used the same sampling temperature described in (Brown et al., 2024) as they allow sample diversity while ensuring quality. Specifically we use temperature $= 1.6$ for Deepseek-V3 and temperature $= 0.7$ for Llama 3.1-70B.

### C.3. Iterative Refinement Prompts

For iterative refinement, we start with the same initial prompt that we used for the one-shot baseline in Appendix C.1. A limitation of our experiments is that we sample with temperature$= 0$ to focus on the effect of iterating based on feedback rather than introducing variability. On subsequent generations, we prompt the model with the following template depending on the feedback it expects:

```
1   <Initial prompt from one-shot baseline for specific KernelBench problem.>
2
3   Here is your latest generation:
4   <Previously generated kernel G>
5
6   Your generated architecture ModelNew and kernel was evaluated on GPU and checked against
        ↪ the reference architecture Model.
7   Here is your Evaluation Result:
8
9   <Raw Compiler and Execution Feedback from stdout>
10
11  <'if correct:'>
12  Your kernel executed successfully and produced the correct output.
13  Here is your wall clock time: {runtime} milliseconds
14
15  <Profiler information if used and correct.>
16
17  Name your new improved output architecture ModelNew. Output the new code in codeblocks.
        ↪ Please generate real code, NOT pseudocode, make sure the code compiles and is fully
        ↪  functional. Just output the new model code, no other text, and NO testing code!
```

For the compiler and execution feedback, we handle timeouts and deadlocks explicitly with "Your kernel execution timed out", but do not provide any other information.

### C.4. Few-Shot in Context Prompts

For Few-Shot experiments as outlined in Section 5.2.1. We provide more details about the in-context example in Appendix F. We sampled these experiments with temperature $= 0$.

```
1   <Initial Task prompt from one-shot baseline for Instruction>
```

```
 2  <Initial pair of Reference PyTorch and CUDA kernel equivalent for example add kernel from
        ↪ one-shot baseline for Instruction>
 3
 4  Example 
 5  Here is an example architecture
 6  <PyTorch reference architecture for No. i in-context example>
 7
 8  Here is an optimized verison with custom CUDA kernels:
 9  <PyTorch architecture with Custom CUDA Kernel for No. i in-context example>
10
11  .. up to number of in-context sample times
12
13
14  Task:
15  Here is an example architecture:
16
17  <PyTorch reference architecture for specific KernelBench Problem>
18
19  Name your new improved output architecture ModelNew. Output the new code in codeblocks.
        ↪ Please generate real code, NOT pseudocode, make sure the code compiles and is fully
        ↪  functional. Just output the new model code, no other text, and NO testing code!
```

### C.5. Hardware Case Study Prompts

Here we provide hardware information. This is used in Section 4.4 and elaborated more in G, sampled with temperature = 0.

```
 1  <Initial Task prompt from one-shot baseline for Instruction>
 2  <Initial pair of Reference PyTorch and CUDA kernel equivalent for example add kernel from
        ↪ one-shot baseline for Instruction>
 3
 4  Here is some information about the underlying hardware that you should keep in mind.
 5
 6  The GPU that will run the kernel is NVIDIA <GPU NAME>.
 7
 8  - We have <x> GB GDDR6 with ECC of GPU Memory.
 9  - We have <x> GB/s of Memory Bandwidth.
10  - We have <x> of RT Core Performance TFLOPS.
11  - We have <x> of FP32 TFLOPS.
12  - We have <x> of TF32 Tensor Core TFLOPS.
13  - We have <x> of FP16 Tensor Core TFLOPS.
14  - We have <x> of FP8 Tensor Core TFLOPS.
15  - We have <x> of Peak INT8 Tensor TOPS.
16  - We have <x> of Peak INT4 Tensor TOPS.
17  - We have <x> 32-bit registers per SM of Register File Size.
18  - We have <x> of Maximum number of registers per thread.
19  - We have <x> of Maximum number of thread blocks per SM.
20  - We have <x> KB of Shared memory capacity per SM.
21  - We have <x> KB of Maximum shared memory per thread block.
22
23
24
25  Here are some concepts about the GPU architecture that could be helpful:
26
27  - Thread: A thread is a single execution unit that can run a single instruction at a time.
28  - Thread Block: A thread block is a group of threads that can cooperate with each other.
29  - Shared Memory: Shared memory is a memory space that can be accessed by all threads in a
        ↪ thread block.
30  - Register: A register is a small memory space that can be accessed by a single thread.
31  - Memory Hierarchy: Memory hierarchy is a pyramid of memory types with different speeds
        ↪ and sizes.
32  - Memory Bandwidth: Memory bandwidth is the rate at which data can be read from or stored
        ↪ into memory.
33  - Cache: Cache is a small memory space that stores frequently accessed data.
34  - HBM: HBM is a high-bandwidth memory technology that uses 3D-stacked DRAM.
```

```
35
36  Here are some best practices for writing CUDA kernels on GPU
37
38  - Find ways to parallelize sequential code.
39  - Minimize data transfers between the host and the device.
40  - Adjust kernel launch configuration to maximize device utilization.
41  - Ensure that global memory accesses are coalesced.
42  - Minimize redundant accesses to global memory whenever possible.
43  - Avoid long sequences of diverged execution by threads within the same warp.
44    #We added this to reference the specific GPU architecture
45  - Use specialized instructions based on the specific GPU architecture
46
47  You are given the following architecture:
48
49  <PyTorch reference architecture for specific KernelBench Problem>
50
51  Name your new improved output architecture ModelNew. Output the new code in codeblocks.
        ↪ Please generate real code, NOT pseudocode, make sure the code compiles and is fully
        ↪  functional. Just output the new model code, no other text, and NO testing code!
```

# D. Kernels of Interest

In this section we provide examples of interesting or notable kernel generations. We first expand on the discussion in Section 6, where we defined the following categories of optimizations: algorithmic optimizations, operator fusion, and using hardware features.

## D.1. Algorithmic Optimizations

### 13x Speedup on Level 1 Problem 11 by Claude-3.5 Sonnet

The original torch operator is `torch.diag(A) @ B`, multiplying a diagonal matrix formed from the vector A with the matrix B. The model identifies an optimization in the special case of a diagonal matrix multiplication, where the diagonal matrix doesn't need to be explicitly constructed. Instead, each element of the vector A is directly multiplied with the corresponding row in matrix B, significantly improving performance:

```
1   __global__ void diag_matmul_kernel(
2       const float* diag,
3       const float* mat,
4       float* out,
5       const int N,
6       const int M) {
7
8       const int row = blockIdx.y * blockDim.y + threadIdx.y;
9       const int col = blockIdx.x * blockDim.x + threadIdx.x;
10
11      if (row < N && col < M) {
12          out[row * M + col] = diag[row] * mat[row * M + col];
13      }
14  }
```

## D.2. Kernel Fusion

### 2.9x Speedup on Level 1 Problem 87 by DeepSeek-V3

GeLU reference in torch:

```
1   0.5 * x * (1.0 + torch.tanh(math.sqrt(2.0 / math.pi) * (x + 0.044715 * torch.pow(x, 3.0)))
        ↪ )
```

Optimized version fuses in a single kernel. There is also a small constant folding optimization, instead of computing

`math.sqrt(2.0 / math.pi)` repeatedly, the kernel uses the precomputed value `0.7978845608028654f`:

```
1  __global__ void gelu_kernel(const float* x, float* out, int size) {
2      int idx = blockIdx.x * blockDim.x + threadIdx.x;
3      if (idx < size) {
4          float x_val = x[idx];
5          float cdf = 0.5f * (1.0f + tanhf((0.7978845608028654f * (x_val + 0.044715f * x_val
            ↪    * x_val * x_val))));
6          out[idx] = x_val * cdf;
7      }
8  }
```

### *1.3x Speedup* on Level 1 Problem 29 by Claude-3.5 Sonnet

SoftSign reference in torch:

```
1  x / (1 + torch.abs(x))
```

Fused kernel:

```
1  __global__ void softsign_kernel(const float* input, float* output, int size) {
2      int idx = blockIdx.x * blockDim.x + threadIdx.x;
3      if (idx < size) {
4          float x = input[idx];
5          float abs_x = abs(x);
6          output[idx] = x / (1.0f + abs_x);
7      }
8  }
```

### *2.6x Speedup* on Level 2 Problem 13 by Claude-3.5 Sonnet

Sequence of operators in torch:

```
1  x = torch.matmul(x, self.weight.T)  # Gemm
2  x = x / 2  # Divide
3  x = torch.sum(x, dim=1, keepdim=True) # Sum
4  x = x * self.scaling_factor  # Scaling
```

Fused kernel:

```
1   __global__ void fused_ops_kernel(
2       const float* input,
3       const float* weight,
4       float* output,
5       const float scaling_factor,
6       const int batch_size,
7       const int input_size,
8       const int hidden_size
9   ) {
10      // Each thread handles one element in the batch
11      const int batch_idx = blockIdx.x * blockDim.x + threadIdx.x;
12
13      if (batch_idx < batch_size) {
14          float sum = 0.0f;
15
16          // Compute matmul and divide for this batch element
17          for(int h = 0; h < hidden_size; h++) {
18              float elem = 0.0f;
19              for(int i = 0; i < input_size; i++) {
20                  elem += input[batch_idx * input_size + i] *
21                          weight[h * input_size + i];
22              }
23              // Divide by 2 as we go
24              sum += (elem / 2.0f);
25          }
26
27          // Scale and store final result
28          output[batch_idx] = sum * scaling_factor;
29      }
30  }
```

Despite this good example, overall Level 2 generated kernels show insufficient fusion. It is expected that most Level 2 problems can be expressed in a single fused kernel.

### *1.9x Speedup* on Level 3 Problem 49 by OpenAI-o1

We have a hypothetical architecture of the attention mechanism where the softmax is replaced with a ReLU.

```
1       att = (q @ k.transpose(-2, -1)) * (1.0 / math.sqrt(k.size(-1)))
2       att = att.masked_fill(self.bias[:,:,:T,:T] == 0, float('-inf'))
3       att = F.relu(att)
```

The model found an optimization that fuses the scaling, masked fill, and ReLU but not anything else, resulting in a modest improvement of 1.9x.

```
1   __global__ void fused_masked_fill_scale_relu_kernel(
2       const float* __restrict__ att,
3       const float* __restrict__ bias,
4       float* __restrict__ output,
5       int total_elems,
6       float scale,
7       int T,
8       float negative_infinity
9   ) {
10      int idx = blockIdx.x * blockDim.x + threadIdx.x;
11      if (idx < total_elems) {
12          float val = att[idx] * scale;
13          int bias_idx = idx % (T * T);
14          if (bias[bias_idx] == 0.0f) {
15              val = negative_infinity;
16          }
17          if (val < 0.0f) {
18              val = 0.0f;
19          }
20          output[idx] = val;
21      }
22  }
```

### D.3. Hardware Features

### *2.8x Speedup* on Level 1 Problem 96 by OpenAI-o1

Torch reference for Cosine Similarity Loss

```
1   cosine_sim = torch.nn.functional.cosine_similarity(predictions, targets, dim=1)
2   return torch.mean(1 - cosine_sim)
```

The generated kernel uses shared memory for reduce redundant global memory accesses, improving data locality and increasing overall performance. This is a moderately complicated kernel with synchronization points and reductions that would be tricky for humans to get right.

```
1   __global__ void cosine_similarity_loss_kernel(
2       const float* __restrict__ predictions,
3       const float* __restrict__ targets,
4       float* __restrict__ losses,
5       const int batch_size,
6       const int input_size
7   ) {
8       // Each block handles one sample in the batch
9       int sample_idx = blockIdx.x;
10      if (sample_idx >= batch_size) return;
11
12      // Shared memory for reductions
13      extern __shared__ float sdata[];
14
15      // Pointers to data for this sample
16      const float* pred = predictions + sample_idx * input_size;
17      const float* targ = targets + sample_idx * input_size;
18
19      // Intermediate sums for dot product and norms
20      float thread_dot = 0.0f;
21      float thread_pred_norm_sq = 0.0f;
22      float thread_targ_norm_sq = 0.0f;
23
24      for (int idx = threadIdx.x; idx < input_size; idx += blockDim.x) {
25          float p = pred[idx];
```

```
26          float t = targ[idx];
27          thread_dot += p * t;
28          thread_pred_norm_sq += p * p;
29          thread_targ_norm_sq += t * t;
30      }
31
32      // Reduction for dot product
33      sdata[threadIdx.x] = thread_dot;
34      __syncthreads();
35      for (unsigned int s = blockDim.x / 2; s > 0; s >>= 1) {
36          if (threadIdx.x < s) {
37              sdata[threadIdx.x] += sdata[threadIdx.x + s];
38          }
39          __syncthreads();
40      }
41      float dot_product = sdata[0];
42
43      // Reduction for pred_norm_sq
44      sdata[threadIdx.x] = thread_pred_norm_sq;
45      __syncthreads();
46      for (unsigned int s = blockDim.x / 2; s > 0; s >>= 1) {
47          if (threadIdx.x < s) {
48              sdata[threadIdx.x] += sdata[threadIdx.x + s];
49          }
50          __syncthreads();
51      }
52      float norm_pred = sqrtf(sdata[0] + 1e-8f);
53
54      // Reduction for targ_norm_sq
55      sdata[threadIdx.x] = thread_targ_norm_sq;
56      __syncthreads();
57      for (unsigned int s = blockDim.x / 2; s > 0; s >>= 1) {
58          if (threadIdx.x < s) {
59              sdata[threadIdx.x] += sdata[threadIdx.x + s];
60          }
61          __syncthreads();
62      }
63      float norm_targ = sqrtf(sdata[0] + 1e-8f);
64
65      if (threadIdx.x == 0) {
66          float cosine_sim = dot_product / (norm_pred * norm_targ + 1e-8f);
67          losses[sample_idx] = 1.0f - cosine_sim;
68      }
69 }
```

***1.9x Speedup*** on Level 1 Problem 98 by Deepseek-R1
Torch reference for Cosine Similarity Loss

```
1 self.loss_fn = torch.nn.TripletMarginLoss(margin=margin)
2 self.loss_fn(anchor, positive, negative)
```

Another example of a generated kernel using shared memory:

```
1   __global__ void triplet_margin_loss_kernel(
2       const float* anchor,
3       const float* positive,
4       const float* negative,
5       float* losses,
6       float margin,
7       int feature_size)
8   {
9       extern __shared__ float shared_sums[];
10
11      int batch_idx = blockIdx.x;
12      int tid = threadIdx.x;
13
14      int offset = batch_idx * feature_size;
15
16      const float* a = anchor + offset;
17      const float* p = positive + offset;
18      const float* n = negative + offset;
19
20      float a_p_sum = 0.0f;
21      float a_n_sum = 0.0f;
22
23      int stride = blockDim.x;
24      for (int i = tid; i < feature_size; i += stride) {
25          float diff_ap = a[i] - p[i];
26          a_p_sum += diff_ap * diff_ap;
27          float diff_an = a[i] - n[i];
28          a_n_sum += diff_an * diff_an;
29      }
30
31      shared_sums[tid] = a_p_sum;
32      shared_sums[blockDim.x + tid] = a_n_sum;
33
34      __syncthreads();
35
36      for (int s = blockDim.x / 2; s > 0; s >>= 1) {
37          if (tid < s) {
38              shared_sums[tid] += shared_sums[tid + s];
39              shared_sums[blockDim.x + tid] += shared_sums[blockDim.x + tid + s];
40          }
41          __syncthreads();
42      }
43
44      if (tid == 0) {
45          float d_ap = sqrtf(shared_sums[0]);
46          float d_an = sqrtf(shared_sums[blockDim.x]);
47          losses[batch_idx] = fmaxf(d_ap - d_an + margin, 0.0f);
48      }
49  }
```

## D.4. Iterative Refinement Examples

### D.4.1. ITERATIVELY TRYING NEW OPTIMIZATIONS

We provide an example of a kernel that iteratively improves on its existing generation. In the following example, the model attempts new optimizations incorrectly, fixes them, and continue to attempt new optimizations, improving its kernel to faster than the torch.compile baseline (1.34ms) but short of the Torch Eager baseline (0.47ms).

### Level 1, Problem 63: 2D convolution with square input and square kernel. DeepSeek-R1 with Execution and Profile Feedback

In this example, we see a $8\times$ speedup in average kernel runtime from its initial generation, where the model repeatedly

|  | Turn 1 | Turn 2 | Turn 3 | Turn 4 | Turn 5 | Turn 6 | Turn 7 | Turn 8 | Turn 9 | Turn 10 |
|---|---|---|---|---|---|---|---|---|---|---|
| **Compiles?** | ✓ | ✗ | ✓ | ✗ | ✓ | ✓ | ✗ | ✓ | ✗ | ✓ |
| **Correct?** | ✓ | ✗ | ✓ | ✗ | ✓ | ✓ | ✗ | ✓ | ✗ | ✓ |
| **Runtime (ms)** | 9.1 | - | 1.57 | - | 1.83 | 1.43 | - | **1.13** | - | 1.46 |

*Table 5.* Iterative refinement trajectory of DeepSeek-R1 with execution feedback $E$ and profiler feedback $P$ on Problem 63, Level 1. Torch Eager baseline runs in 0.47ms and `torch.compile` runs in 1.34ms.

(incorrectly) refines its kernel, fixes the compiler issues using feedback, then continues to attempt more optimizations. The first big jump in performance (Turn 1 → Turn 3) occurs because the model decides to launch thread blocks along an output channel dimension, when it originally computed these elements sequentially. The model then attempts to use shared memory in Turn 5, and continues using it, along with texture cache memory with the `__ldg` instruction in Turns 7 and 8.

D.4.2. LEVERAGING FEEDBACK TO CORRECT KERNEL CODE

**Level 2, Problem 73: 2D Convolution with a BatchNorm and a scale factor. DeepSeek-R1 with Execution Feedback**
We provide an example of a kernel that the model struggles to generate correctly, and produces a correct kernel after iterative refinement using execution feedback.

|  | Turn 1 | Turn 2 | Turn 3 | Turn 4 | Turn 5 | Turn 6 | Turn 7 | Turn 8 | Turn 9 | Turn 10 |
|---|---|---|---|---|---|---|---|---|---|---|
| **Compiles?** | ✓ | ✓ | ✓ | ✓ | ✓ | ✓ | ✓ | ✓ | ✓ | ✓ |
| **Correct?** | ✗ | ✗ | ✗ | ✗ | ✗ | ✗ | ✗ | ✗ | ✗ | ✓ |
| **Runtime** | - | - | - | - | - | - | - | - | - | **3.16** |

*Table 6.* Iterative refinement trajectory of DeepSeek-R1 with execution feedback $E$ on Problem 73, Level 2. Torch Eager baseline runs in 0.105ms and `torch.compile` runs in 0.156ms.

In the above example, the model continually produces either the wrong output tensor shape or the wrong values and iterates on its kernel using this feedback until the final turn, where it generates a functionally correct, albeit non-performant kernel. We provide another example below that explicitly leverages compiler feedback to fix compiler errors:

**Level 2, Problem 23: 3D Convolution with a GroupNorm and return the mean across all but the batch dimension. DeepSeek-R1 with Execution Feedback**

|  | Turn 1 | Turn 2 | Turn 3 | Turn 4 | Turn 5 | Turn 6 | Turn 7 | Turn 8 | Turn 9 | Turn 10 |
|---|---|---|---|---|---|---|---|---|---|---|
| **Compiles?** | ✓ | ✓ | ✓ | ✓ | ✗ | ✗ | ✓ | ✓ | ✗ | ✓ |
| **Correct?** | ✗ | ✗ | ✓ | ✓ | ✗ | ✗ | ✓ | ✓ | ✗ | ✗ |
| **Runtime** | - | - | 11.4 | 1.36 | - | - | 1.39 | **1.33** | - | - |

*Table 7.* Iterative refinement trajectory of DeepSeek-R1 with execution feedback $E$ on Problem 23, Level 2. Torch Eager baseline runs in 1.29ms and `torch.compile` runs in 0.719ms.

In the above example, the model attempts to use the `CUB` library, but incorrectly invokes function calls. The model is then able to correct these errors and write a slightly faster kernel in Turn 8.

D.4.3. ITERATIVE REFINEMENT NEVER FIXES THE ERROR

**Level 1, Problem 54: 3D Convolution square input and square kernel. DeepSeek-R1 with Execution and Profiler Feedback**

This problem is particularly interesting because no model is able to consistently produce functional code for this kernel, even with different forms of feedback and profiling information. Interestingly, the example before is an arguably more difficult version of this kernel that fuses the 3D convolution with another operator, and the same model is able to generate functional code for this task. In the example above, the model consistently makes the same mistake and continually generates

| | Turn 1 | Turn 2 | Turn 3 | Turn 4 | Turn 5 | Turn 6 | Turn 7 | Turn 8 | Turn 9 | Turn 10 |
|---|---|---|---|---|---|---|---|---|---|---|
| **Compiles?** | ✓ | ✓ | ✓ | ✓ | ✓ | ✓ | ✓ | ✓ | ✓ | ✓ |
| **Correct?** | ✗ | ✗ | ✗ | ✗ | ✗ | ✗ | ✗ | ✗ | ✗ | ✗ |
| **Runtime** | - | - | - | - | - | - | - | - | - | - |

*Table 8.* Iterative refinement trajectory of DeepSeek-R1 with execution feedback $E$ and profiler feedback $P$ on Problem 54, Level 1. Torch Eager baseline runs in 4.47ms and `torch.compile` runs in 4.67ms.

a functionally incorrect kernel with the same value errors.

# E. Iterative Refinement on Correctness

Here we show that $fast_0$ across iterative refinement 5.1.2 configurations at a turn budget of $N = 10$ compared to one-shot baseline 4.1. We find that models self-correct more effectively with execution feedback $E$, fixing issues especially related to execution errors. Notably, DeepSeek-R1 on Level 1 and 2 can generate a functional kernel on $>90\%$ of the tasks given 10 turns of iterative refinement. However, the remaining incorrect kernels almost always fail due to functional incorrectness, likely because correctness feedback is less granular than execution failure messages

| | Level 1 | | | Level 2 | | | Level 3 | | |
|---|---|---|---|---|---|---|---|---|---|
| **Method** | Llama 3.1 70B | DeepSeek V3 | DeepSeek R1 | Llama 3.1 70B | DeepSeek V3 | DeepSeek R1 | Llama 3.1 70B | DeepSeek V3 | DeepSeek R1 |
| Single Attempt (Baseline) | 26% | 43% | 67% | 0% | 6% | 62% | 0% | 30% | 8% |
| Iterative Refinement (w G) | 27% | 48% | 72% | 2% | 7% | 67% | 0% | 36% | 14% |
| Iterative Refinement (w G+E) | **40%** | **53%** | 95% | **7%** | 8% | 85% | 18% | 42% | **50%** |
| Iterative Refinement (w G+E+P) | 36% | 50% | **95%** | **7%** | **9%** | **92%** | 8% | **44%** | 42% |

*Table 9.* **Leveraging execution feedback helps reduce errors:** Here we present the percentage of problems where the LM-generated Kernel is correct for iterative refinement. We note leveraging execution feedback helps the model achieve better correctness $fast_0$, which is the percentage of problems where the model has at least one correct generation up to turn $N = 10$. We note the various iterative refinement configurations, leveraging previous Generation $G$, Execution Result $E$, and Timing Profiles $P$.

# F. Few Shot Experiment

For this experiment, we provide in-context examples of optimization techniques such as fusion, tiling, recompute, and asynchrony to models during kernel generation. As described in Section 5.2.1, we provide three in-context examples: a fused GELU (Hendrycks & Gimpel, 2023), a tiled matrix multiplication (Mills, 2024), and a minimal Flash-Attention (Dao et al., 2022; Kim, 2024) demonstrating effective shared memory I/O management. The prompt used for this experiment is described in Appendix C.4. The speedup of these kernels were computed over PyTorch Eager. We compare the performance of these few-shot kernels over the one-shot baseline below.

| Model | Level | Baseline | | | Few-Shot | | |
|---|---|---|---|---|---|---|---|
| | | $fast_1$ | $fast_0$ | Kernel Length (chars) | $fast_1$ | $fast_0$ | Kernel Length (chars) |
| | 1 | 3% | 27% | 301018 | 6% | 27% | 360212 |
| Llama 3.1-70B | 2 | 0% | 0% | 646403 | 0% | 0% | 566668 |
| | 3 | 0% | 0% | 404596 | 0% | 4% | 485332 |
| | 1 | 10% | 55% | 343995 | 6% | 39% | 437768 |
| OpenAI o1 | 2 | 24% | 56% | 381474 | 16% | 39% | 432800 |
| | 3 | 12% | 56% | 260273 | 8% | 22% | 364551 |

*Table 10.* Comparison of the Section 4.1 baseline and few-shot prompting performance across models. We examine the $fast_0$, $fast_1$, and cumulative character length of generated kernels per level.

77% of matrix multiplication problems in Level 1 achieves a speedup over the one-shot baseline through tiling. The runtime comparison for each GEMM variant is presented below as Table F.

| Problem Name | Baseline (ms) | Few-Shot (ms) | Ref Torch (ms) |
|---|---|---|---|
| 3D Tensor Matrix Multiplication | 20.9 | 7.71 | 1.45 |
| Matmul for Upper-Triangular Matrices | 14 | 5.39 | 2.98 |
| Matrix Scalar Multiplication | 1.19 | 0.811 | 0.822 |
| Standard Matrix Multiplication | 3.39 | 2.46 | 0.397 |
| Matmul with Transposed Both | 3.44 | 2.67 | 0.412 |
| Matmul with Transposed A | 3.61 | 2.99 | 0.384 |
| 4D Tensor Matrix Multiplication | 366 | 338 | 36 |
| Tall Skinny Matrix Multiplication | 3.39 | 3.59 | 1.9 |
| Matmul with Diagonal Matrices | 0.221 | 0.237 | 2.83 |

*Table 11.* Performance comparison of the Section 4.1 baseline and few-shot prompting in level 1 matrix multiplication problems.

Few-shot kernels generated for the following problems in level 2 outperformed PyTorch Eager through aggressive shared memory I/O management.

| Problem Name | Baseline (ms) | Few-Shot (ms) | Ref Torch (ms) |
|---|---|---|---|
| Conv2d InstanceNorm Divide | 0.514 | 0.0823 | 0.0898 |
| Gemm GroupNorm Swish Multiply Swish | 0.124 | 0.0542 | 0.0891 |
| Matmul Min Subtract | 0.0651 | 0.0342 | 0.0397 |
| Matmul GroupNorm LeakyReLU Sum | 0.0935 | 0.0504 | 0.072 |
| ConvTranspose3d Swish GroupNorm HardSwish | 33.3 | 29.6 | 35.2 |
| ConvTranspose2d Mish Add Hardtanh Scaling | 0.235 | 0.209 | 0.243 |
| ConvTranspose3d Add HardSwish | 15.6 | 14.1 | 22.2 |
| ConvTranspose2d Add Min GELU Multiply | 0.365 | 0.349 | 0.4 |
| ConvTranspose2d BiasAdd Clamp Scaling Clamp... | 0.3 | 0.31 | 0.368 |
| Conv2d GroupNorm Tanh HardSwish ResidualAdd... | 0.124 | 0.129 | 0.154 |
| Conv2d ReLU HardSwish | 0.0681 | 0.0711 | 0.0768 |

*Table 12.* Performance comparison of the Section 4.1 baseline and few-shot prompting in level 2 for problems whose few-shot kernels outperform PyTorch Eager.

# G. Cross-Hardware Case Study

## G.1. Evaluation across different hardware

To evaluate how generated kernels fare across different hardware platforms, we utilize a number of different NVIDIA GPUs that span different micro-architectures and capabilities. The specific details for each is provided in Table 13.

| Provider | GPU Type | Memory | Power | Microarchitecture | FP16 TFLOPS | Memory Bandwidth |
|---|---|---|---|---|---|---|
| Baremetal | NVIDIA L40S | 48 GB | 300W | Ada | 362.05 | 864 GB/s |
| Baremetal | NVIDIA H100 | 80 GB | 700W | Hopper | 989.5 | 3350 GB/s |
| Serverless | NVIDIA L40S | 48 GB | 350W | Ada | 362.05 | 864 GB/s |
| Serverless | NVIDIA A100 | 42 GB | 400W | Ampere | 312 | 1935 GB/s |
| Serverless | NVIDIA L4 | 24 GB | 72W | Ada | 121 | 300 GB/s |
| Serverless | NVIDIA T4 | 16 GB | 70W | Turing | 65 | 300 GB/s |
| Serverless | NVIDIA A10G | 24 GB | 300W | Ampere | 125 | 600 GB/s |

*Table 13.* Specifications of different GPUs, including memory, power consumption, micro-architecture, FP16 TFLOPS, memory bandwidth, and their providers.

We ran the same set of kernels generated in Section 4.1 on a variety of hardware (as listed in Table 13). We computed the $\text{fast}_1$ speedup against the PyTorch Eager baseline profiled on that particular hardware platform in Table 14.

| Level | GPUs | Llama-3.1-70b-Inst | DeepSeek-V3 | DeepSeek-R1 |
|---|---|---|---|---|
| 1 | L40S | 3% | 6% | 12% |
| | H100 | 2% | 7% | 16% |
| | A100 | 3% | 7% | 16% |
| | L4 | 2% | 4% | 15% |
| | T4 | 3% | 7% | 22% |
| | A10G | 2% | 7% | 12% |
| 2 | L40S | 0% | 4% | 36% |
| | H100 | 0% | 4% | 42% |
| | A100 | 0% | 4% | 38% |
| | L4 | 0% | 4% | 36% |
| | T4 | 0% | 4% | 46% |
| | A10G | 0% | 4% | 47% |
| 3 | L40S | 0% | 8% | 2% |
| | H100 | 0% | 10% | 2% |
| | A100 | 0% | 8% | 2% |
| | L4 | 0% | 6% | 2% |
| | T4 | 0% | 10% | 2% |
| | A10G | 0% | 10% | 0% |

*Table 14.* KernelBench result across multiple hardware types: Speedup ($\text{fast}_1$) over Torch Eager comparison of GPUs across different models and levels. The kernels used across different GPUs are the same as the ones generated for Single Attempt **without** hardware/platform specific information.

Based on the increased variability in $\text{fast}_1$ score for DeepSeek R1 as described in Section 4.4 and Table 14, we plot the individual speedups for each problem (in Levels 1 and 2) across different GPUs. Speedup is computed against PyTorch Eager and there is a horizontal line at $y = 1.0$ to mark the cutoff for $\text{fast}_1$.

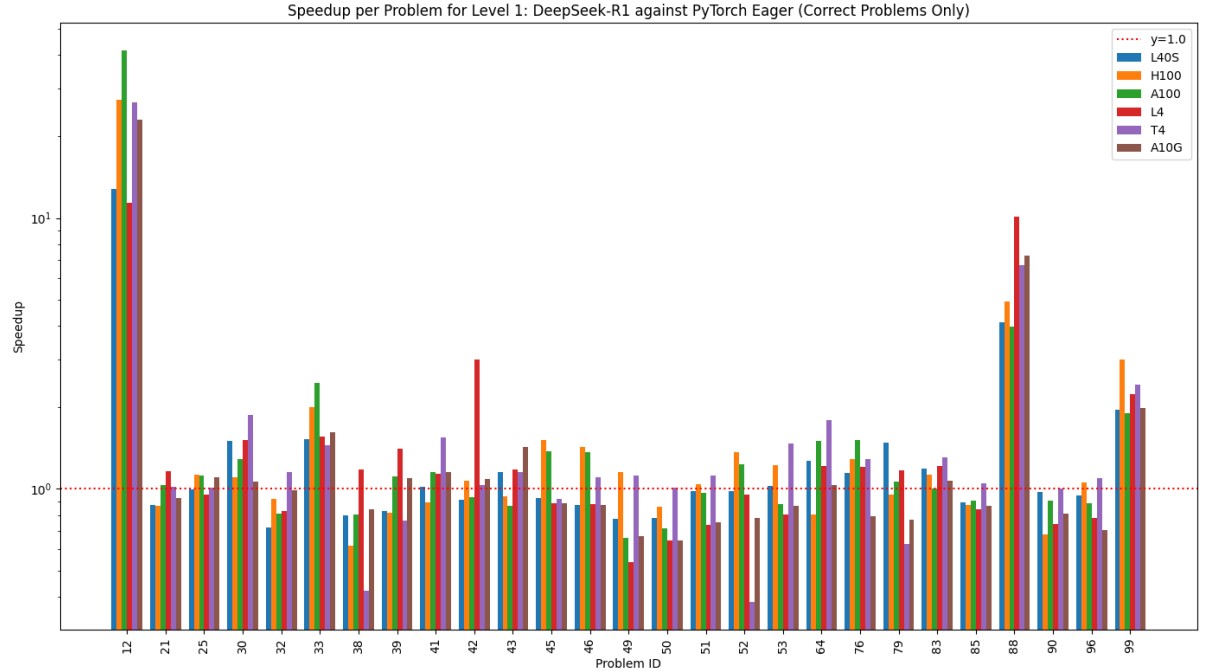

*Figure 8.* Speedup comparison across different GPUs for DeepSeek R1 on Level 1 (log scale).

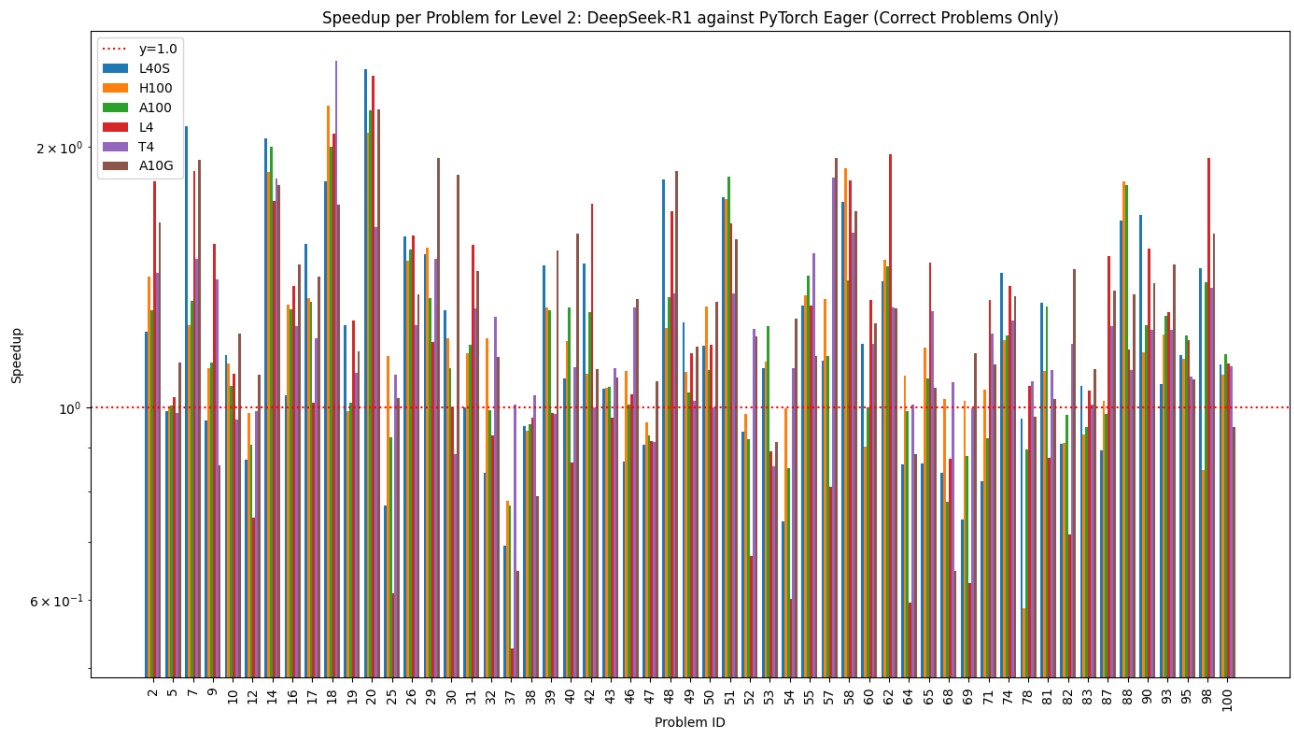

*Figure 9.* Speedup comparison across different GPUs for DeepSeek-R1 on Level 2 (log scale).

## G.2. Effect of Providing Hardware Information

We provided hardware-specific information such as the GPU's micro-architecture, TFLOPS, and memory bandwidth (as shown in Table 13) to models during kernel generation. These generations were then profiled on their respective hardware platforms. The speedup in terms of fast$_1$ over PyTorch Eager is provided below in Table 15.

| Level | GPUs | Llama-3.1-70b-Inst | DeepSeek-V3 | DeepSeek-R1 | OpenAI-o1 |
|---|---|---|---|---|---|
| | **L40S** | 2% | 5% | 14% | 13% |
| **1** | **H100** | 3% | 6% | 14% | 9% |
| | **A100** | 4% | 4% | 16% | 9% |
| | **L40S** | 1% | 4% | 33% | 25% |
| **2** | **H100** | 2% | 2% | 35% | 33% |
| | **A100** | 2% | 1% | 31% | 32% |
| | **L40S** | 0% | 2% | 0% | 10% |
| **3** | **H100** | 2% | 8% | 0% | 12% |
| | **A100** | 0% | 0% | 2% | 8% |

*Table 15.* Providing hardware specifications in context: Speedup (fast$_1$) over Torch Eager comparison of GPUs across different models and levels.

As described in Section 5.2.2, DeepSeek-R1 begins to generate kernels with warp matrix multiply-accumulate (`wmma`) instructions. We show the generated kernel for Problem 1 Level 1 when provided with H100 specifications in context below.

```
#include <torch/extension.h>
#include <cuda_fp16.h>
#include <cuda_runtime.h>
#include <mma.h>

using namespace nvcuda;

__global__ void hgemm_kernel(const half* A, const half* B, half* C, int N) {
    const int WARPS_PER_BLOCK = 8;
    const int WMMA_M = 16;
    const int WMMA_N = 16;
    const int WMMA_K = 16;

    __shared__ half Ashared[WARPS_PER_BLOCK][WMMA_M][WMMA_K];
    __shared__ half Bshared[WARPS_PER_BLOCK][WMMA_K][WMMA_N];

    int warpId = threadIdx.x / 32;
    int laneId = threadIdx.x % 32;

    int tileM = (blockIdx.y * WARPS_PER_BLOCK + warpId) * WMMA_M;
    int tileN = blockIdx.x * WMMA_N;

    wmma::fragment<wmma::matrix_a, WMMA_M, WMMA_N, WMMA_K, half, wmma::row_major> a_frag;
    wmma::fragment<wmma::matrix_b, WMMA_M, WMMA_N, WMMA_K, half, wmma::row_major> b_frag;
    wmma::fragment<wmma::accumulator, WMMA_M, WMMA_N, WMMA_K, half> c_frag;
    wmma::fill_fragment(c_frag, __float2half(0.0f));

    for (int k = 0; k < N; k += WMMA_K) {
        int loadAidx = laneId + (tileM + (laneId / WMMA_K)) * N + k + (laneId % WMMA_K);
        int loadBidx = laneId + (k + (laneId / WMMA_N)) * N + tileN + (laneId % WMMA_N);

        if (tileM + (laneId / WMMA_K) < N && k + (laneId % WMMA_K) < N)
            Ashared[warpId][laneId / WMMA_K][laneId % WMMA_K] = A[loadAidx];
        else
            Ashared[warpId][laneId / WMMA_K][laneId % WMMA_K] = __float2half(0.0f);
```

*Figure 10.* A CUDA kernel generated by DeepSeek-R1 for Level 1 Problem 1 when provided with hardware-specific information on the H100 GPU.

```
37          if (k + (laneId / WMMA_N) < N && tileN + (laneId % WMMA_N) < N)
38              Bshared[warpId][laneId / WMMA_N][laneId % WMMA_N] = B[loadBidx];
39          else
40              Bshared[warpId][laneId / WMMA_N][laneId % WMMA_N] = __float2half(0.0f);
41
42          __syncthreads();
43
44          wmma::load_matrix_sync(a_frag, &Ashared[warpId][0][0], WMMA_K);
45          wmma::load_matrix_sync(b_frag, &Bshared[warpId][0][0], WMMA_N);
46          wmma::mma_sync(c_frag, a_frag, b_frag, c_frag);
47
48          __syncthreads();
49      }
50
51      int storeCidx = (tileM + (laneId / WMMA_N)) * N + tileN + (laneId % WMMA_N);
52      if (tileM + (laneId / WMMA_N) < N && tileN + (laneId % WMMA_N) < N)
53          C[storeCidx] = c_frag.x[laneId];
54  }
55
56  torch::Tensor hgemm_cuda(torch::Tensor A, torch::Tensor B, int N) {
57      auto C = torch::zeros({N, N}, A.options().dtype(torch::kFloat16));
58
59      const int WARPS_PER_BLOCK = 8;
60      dim3 grid((N + 15) / 16, (N + 15) / (16 * WARPS_PER_BLOCK));
61      dim3 block(32 * WARPS_PER_BLOCK);
62
63      hgemm_kernel<<<grid, block>>>(A.data_ptr<half>(), B.data_ptr<half>(), C.data_ptr<half
           ↪ >(), N);
64      return C;
65  }
```

### G.3. In-context examples applying hardware-specific instructions

For this experiment, we combine the approaches of Sections 5.2.1 and 5.2.2 by providing in-context examples using architecture-specific instructions such as (1) wmma and (2) memcpy_async along with the same hardware-specific information in Table 13. We specifically target A100 GPUs since the Ampere architecture supports both Tensor Core operations and Asynchronous Memory Movement. We use DeepSeek-R1 based on its ability to generate kernels with wmma instructions without explicit examples as shown in Section 5.2.2 (though it fails to utilize them correctly). Finally, we target simple KernelBench matrix multiplication problems (17 of the Level 1 problems).

Qualitatively, providing DeepSeek-R1 with an in-context example using WMMA instructions motivated the model to try to apply WMMAs as much as possible (all matrix multiplication problems we were targeting had generated kernels that included WMMAs). However, only 5 out of those 17 kernels were correct. The kernel that achieved a $> 3.5x$ speedup over PyTorch Eager performed a matrix multiplication for two upper triangular matrices and skipped unnecessary computation, an optimization enabled by prior knowledge about input matrix characteristics rather than better Tensor Core utilization. The remaining 4 kernels were slower than PyTorch Eager.

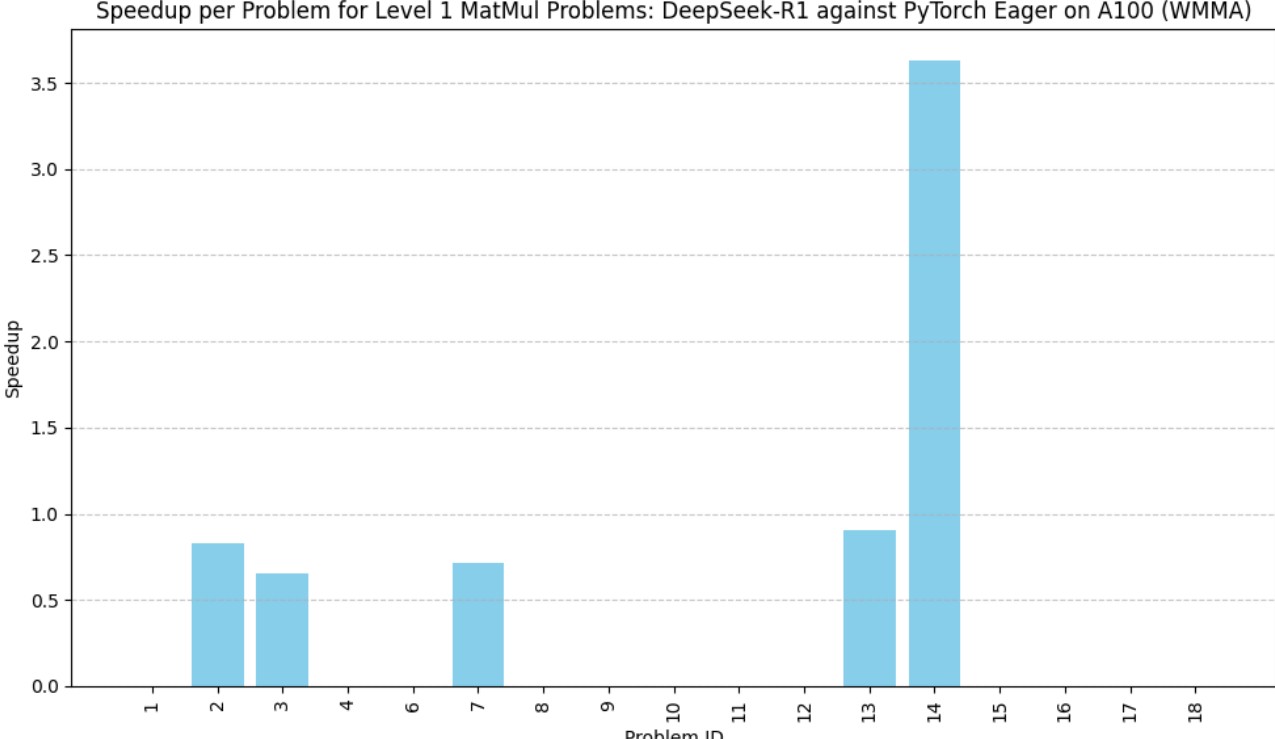

*Figure 11.* Speedup over PyTorch Eager when providing DeepSeek-R1 with an in-context example using WMMA instructions to perform a basic matrix multiplication.

On the other hand, providing DeepSeek-R1 with an in-context example using memcpy_async instructions also motivated it to apply these instructions to all of its generated kernels. However, none of the kernels were correct. The fact that DeepSeek-R1 points to certain hardware capabilities being easier for DeepSeek-R1 to exploit than others.

## H. High-Throughput Evaluation System

### H.1. Single-shot Experiments: Batched Kernel Generation

Given the high volume of GPU kernels to evaluate, we build a fast and highly-parallelized evaluation system, where we separate into the kernel generation and evaluation process into 3 stages, as shown in Figure 12.

- **Inference:** We query LMs in parallel and store the generated kernel.

- **CPU Pre-Compile:** We compile the model-generated kernels with `nvcc` for a specified hardware into a binary, parallelized on CPUs and each kernel binary is saved to their individual specific directory for caching.

- **GPU Evaluation:** With the kernel binary already built on CPU, we focus on evaluating multiple kernels in parallel across multiple GPU devices. However, to ensure accurate kernel timing, we only evaluate one kernel at time on one device.

### H.2. Iterative Refinement Experiments: GPU Orchestrator System

Based on the single-shot system, we also design a platform to handle multiple iterative refinement experiments at once. We treat each iterative refinement experiment as a finite state machine, where the states are LM-based kernel generation, pre-compilation, kernel execution, and profiling. The transitions are based on environment feedback, and can change based on different experiment setups.

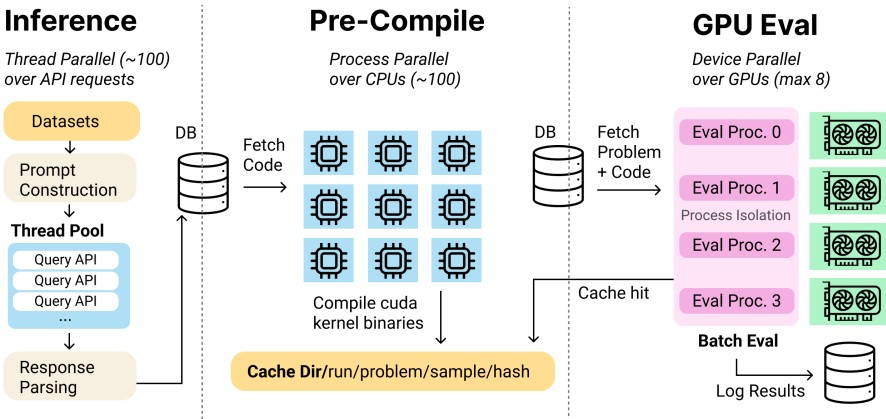

*Figure 12.* **KernelBench provide a high throughput kernel generation and evaluation system**. We parallelized generation, compilation, and evaluation of kernels across CPUs and GPUs.

Our system was run on a node with 8 available GPUs. Unlike the single-shot system, batching each generation and kernel execution is highly inefficient – thus, we design a pipelined, multiprocessing system with a GPU orchestrator with the following characteristics:

- **CPU Parallelism:** The orchestrator spawns multiple independent processes that each handle an independent task in KernelBench. These processes run the multi-turn state machine logic for the iterative refinement experiments – only the kernel execution state requires acquiring a GPU.

- **Acquiring GPUs:** The GPU orchestrator keeps a separate process running that handles which processes can acquire a GPU using semaphores. Processes can request a GPU from this process when it is ready to execute and evaluate kernel code. We try to minimize process control over a GPU to maximize resource throughput, given a system with a limited number of available GPUs.

- **Pre-compiling on the CPU:** To avoid processes hogging GPU time, we pre-compile kernels with `nvcc` on the CPU for a specified hardware into a binary. We also did this same trick for the single-shot system, but for separate reasons.

- **Evaluating Kernels on the GPU:** The only state where the finite state machine uses the GPU is for kernel execution and profiling. We found that waiting on GPUs is the primary bottleneck in the orchestrator, so we designed the orchestrator to maximize device occupancy.

The system generally supports overlapping the generation of kernel code and the execution of already-generated kernel code. There are also several unavoidable errors such as CUDA illegal memory accesses and deadlocks due to faulty kernel generations that the orchestrator solves by releasing and spawning new processes when encountered, and we wrote specifically handlers to ensure these errors are properly captured without crashing the orchestrator itself.

### H.3. UI: Visualizing Kernel Generation Trajectories

To qualitatively observe the generated and compare them across techniques, we design an interface to easily visualize them. We provide this as part of the KernelBench framework.

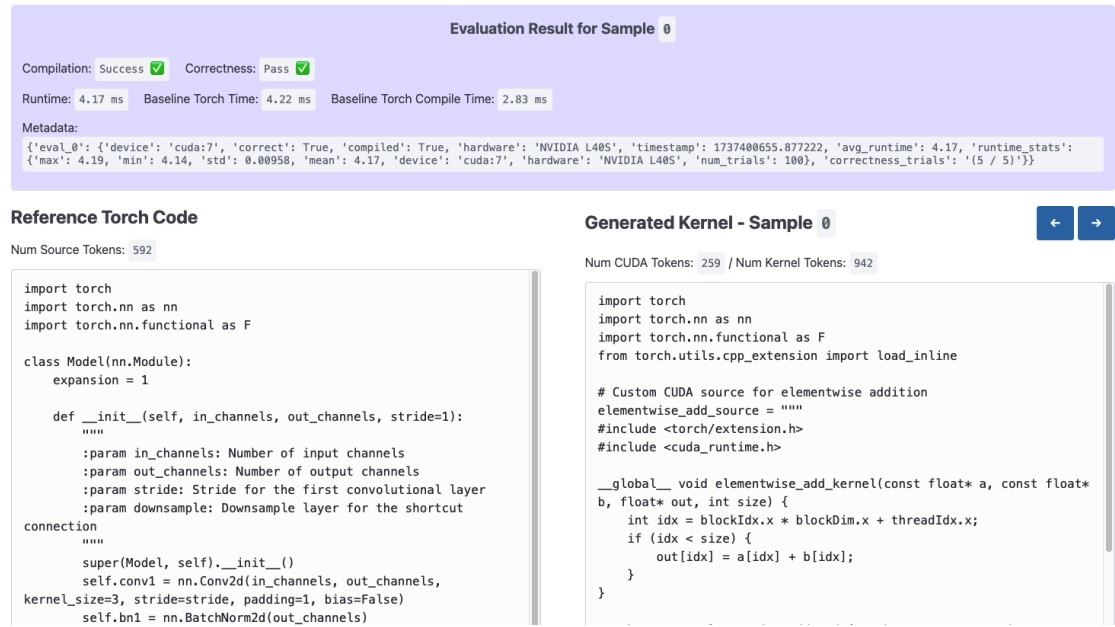

*Figure 13.* **We provide a visual interface for kernel inspection**. This allows us to easily examine kernel content, its performance, and compare across various techniques and configurations.

## I. KernelBench Score Design: Comparing Correctness and Performance

Evaluating kernel generation models requires balancing two crucial aspects: correctness and performance. Correctness ensures that the generated kernel produces the expected outputs, while performance determines the efficiency of execution, typically measured in speedup. Since optimizing one aspect can sometimes degrade the other (e.g., aggressive optimizations may introduce more correctness errors), KernelBench uses the $\text{fast}_p$ metric. In this section, we also include some other metrics we have considered and compare them. Table 16 first shows correctness and speedup separately.

| Model | Level 1 | | Level 2 | | Level 3 | |
|---|---|---|---|---|---|---|
| | % Correct | Speedup | % Correct | Speedup | % Correct | Speedup |
| claude-3.5-sonnet | 53.0 | 0.329074 | 10.0 | 1.026136 | 12.0 | 0.448122 |
| deepseek-V3 | 44.0 | 0.252212 | 5.0 | 1.023760 | 30.0 | 0.778776 |
| deepseek-r1 | 67.0 | 0.483147 | 62.0 | 1.039562 | 8.0 | 0.685164 |
| gpt-4o | 39.0 | 0.330257 | 9.0 | 1.000646 | 14.0 | 0.612829 |
| llama-3.1-405b | 40.0 | 0.228077 | 0.0 | 0.000000 | 4.0 | 0.944302 |
| llama-3.1-70b | 27.0 | 0.178835 | 0.0 | 0.000000 | 0.0 | 0.000000 |
| openai-o1 | 55.0 | 0.325485 | 56.0 | 0.844315 | 56.0 | 0.783615 |

*Table 16.* Correctness and Speedup of Models across different levels. Speedup here is measured as the geometric mean of the correct samples only.

### I.1. Metrics Exploration

The various metrics we explored are defined as follow.

Let $c_i$ be an indicator variable denoting correctness of the $i$-th sample, where:

$$c_i = \begin{cases} 1, & \text{if the solution is correct} \\ 0, & \text{otherwise} \end{cases}$$

Let $T_{b_i}$ be the baseline execution time and $T_{g_i}$ be the actual execution time for the generated kernel. The speedup ratio for a

sample is defined as:

$$S_i = \frac{T_{b_i}}{T_{g_i}}$$

Let $n$ be the total number of samples and $n_{\text{correct}} = \sum_{i=1}^{n} c_i$ be the number of correct samples.

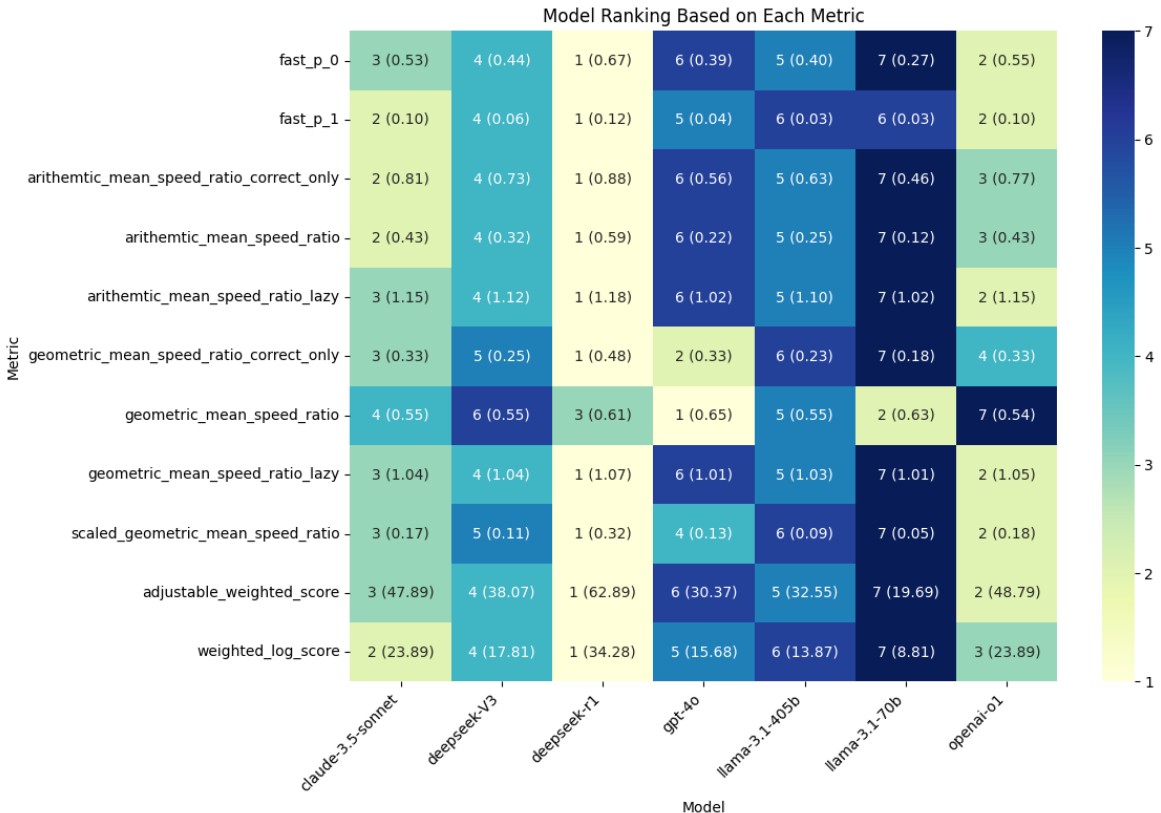

*Figure 14.* Model Performance Ranking Across Various Scoring Metrics

Figure 14 compares model rankings under different performance metrics, capturing variations in correctness and execution speed. Each cell displays the ranking and the corresponding raw score for a given model and metric. Differences in rankings highlight the impact of metric design on performance evaluation.

**Arithmetic Mean Speed Ratio (Correct Only)**

$$\text{AMSR}_{\text{correct}} = \frac{1}{n_{\text{correct}}} \sum_{i=1}^{n} c_i S_i$$

**Arithmetic Mean Speed Ratio (All Samples)**

$$\text{AMSR} = \frac{1}{n} \sum_{i=1}^{n} c_i S_i$$

**Arithmetic Mean Speed Ratio (Lazy)**

$$\text{AMSR}_{\text{lazy}} = \frac{1}{n} \sum_{i=1}^{n} \max(c_i S_i, 1)$$

**Geometric Mean Speed Ratio (Correct Only)**

$$\text{GMSR}_{\text{correct}} = \left( \prod_{i=1}^{n} S_i^{c_i} \right)^{\frac{1}{n_{\text{correct}}}}$$

**Geometric Mean Speed Ratio (All Samples)**

$$\text{GMSR} = \left( \prod_{i=1}^{n} S_i^{c_i} \right)^{\frac{1}{n}}$$

**Geometric Mean Speed Ratio (Lazy)**

$$\text{GMSR}_{\text{lazy}} = \left( \prod_{i=1}^{n} \max(S_i, 1) \right)^{\frac{1}{n}}$$

**Scaled Geometric Mean Speed Ratio**

$$\text{SGMSR} = \frac{n_{\text{correct}}}{n} \times \text{GMSR}_{\text{correct}}$$

**Adjustable Weighted Score**

$$\text{AWS} = \sum_{i=1}^{n} c_i \left( \text{correctness\_weight} + \text{perf\_weight} \times S_i \right)$$

where correctness and performance weights satisfy:

$$\text{correctness\_weight} + \text{perf\_weight} = 1$$

We use correctness_weight=perf_weight=0.5 in our experiments.

**Weighted Log Score**

$$\text{WLS} = \sum_{i=1}^{n} c_i \log(1 + S_i)$$

## J. KernelBench vs. LiveCodeBench Correlation

To assess the relationship between general coding ability and kernel-specific performance, we compare model rankings across LiveCodeBench (Jain et al., 2024) and KernelBench (Levels 1-3).

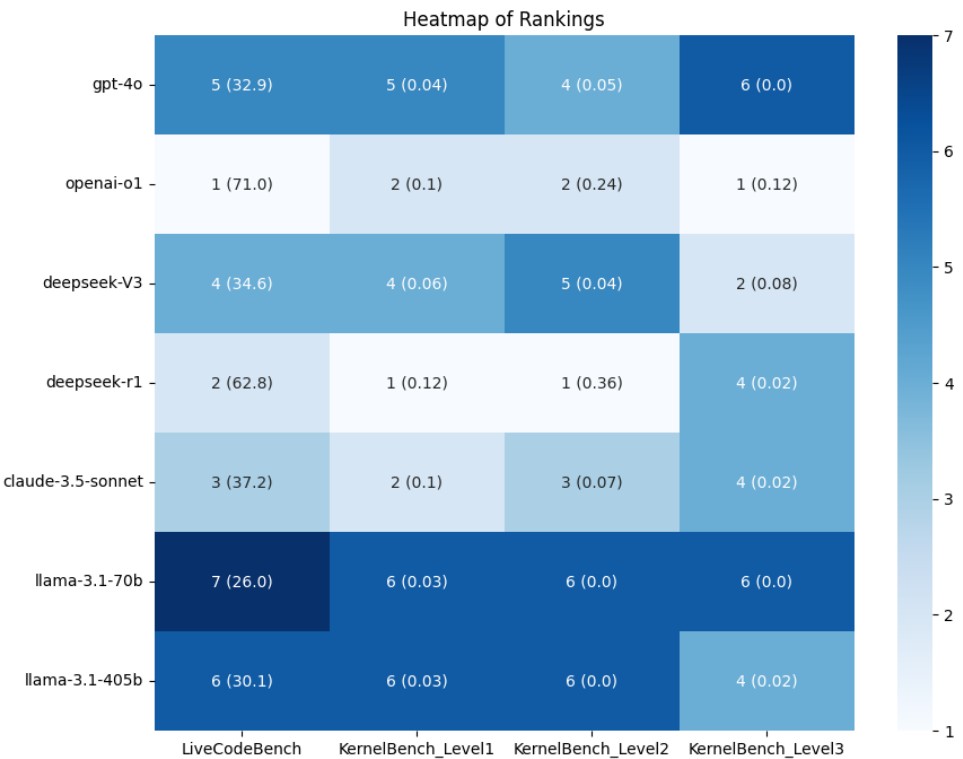

*Figure 15.* Heatmap of model rankings across LiveCodeBench and KernelBench (Levels 1-3).

We observe that **LiveCodeBench performance tend to correlate strongly KernelBench, though the rankings are not perfectly aligned**. llama-3.1-70b and llama-3.1-405b consistently rank the worst across benchmarks, suggesting that weaker general coding ability correlates with poor kernel-specific performance. Openai-o1, which ranks 1st in LiveCodeBench, remains highly competitive in KernelBench (2nd in Level 1 and 2, 1st in Level 3). Deepseek-r1, ranking 2nd in LiveCodeBench, achieves 1st place in KernelBench Level 1 and Level 2.

While models that perform well in LiveCodeBench generally achieve strong results in KernelBench, the **variability in rankings across different levels of KernelBench suggests that additional skills are required for high performance in kernel-specific tasks**. For example, deepseek-r1, which ranks 2nd in LiveCodeBench, drops to 4th in KernelBench level 3, indicating that some aspects of kernel optimization may favor different model strengths.

## K. KernelBench Tasks Breakdown

### K.1. KernelBench Tasks

Reference KernelBench tasks are given in FP32, and given a tolerance threshold ($1e-2$), using lower precision solutions is valid. Furthermore, the problem size of each task is fixed, since our goal is specialized fast kernels not generally fast kernels for any arbitrary shape.

Here we list the names of the KernelBench tasks categorized by level.

# List of Tasks of All Levels

## Level 1 Task Names

1. Square matrix multiplication
2. Standard matrix multiplication
3. Batched matrix multiplication
4. Matrix vector multiplication
5. Matrix scalar multiplication
6. Matmul with large K dimension
7. Matmul with small K dimension
8. Matmul with irregular shapes
9. Tall skinny matrix multiplication
10. 3D tensor matrix multiplication
11. 4D tensor matrix multiplication
12. Matmul with diagonal matrices
13. Matmul for symmetric matrices
14. Matmul for upper triangular matrices
15. Matmul for lower triangular matrices
16. Matmul with transposed A
17. Matmul with transposed B
18. Matmul with transposed both
19. ReLU
20. LeakyReLU
21. Sigmoid
22. Tanh
23. Softmax
24. LogSoftmax
25. Swish
26. GELU
27. SELU
28. HardSigmoid
29. Softplus
30. Softsign
31. ELU
32. HardTanh
33. BatchNorm
34. InstanceNorm
35. GroupNorm
36. RMSNorm
37. FrobeniusNorm
38. L1Norm
39. L2Norm
40. LayerNorm
41. Max Pooling 1D
42. Max Pooling 2D
43. Max Pooling 3D
44. Average Pooling 1D
45. Average Pooling 2D
46. Average Pooling 3D
47. Sum reduction over a dimension
48. Mean reduction over a dimension
49. Max reduction over a dimension
50. Product reduction over a dimension
51. Argmax over a dimension
52. Argmin over a dimension
53. Min reduction over a dimension
54. conv standard 3D square input square kernel
55. conv standard 2D asymmetric input square kernel
56. conv standard 2D asymmetric input asymmetric kernel
57. conv transposed 2D square input square kernel
58. conv transposed 3D asymmetric input asymmetric kernel
59. conv standard 3D asymmetric input square kernel
60. conv standard 3D square input asymmetric kernel
61. conv transposed 3D square input square kernel
62. conv standard 2D square input asymmetric kernel
63. conv standard 2D square input square kernel
64. conv transposed 1D
65. conv transposed 2D square input asymmetric kernel
66. conv standard 3D asymmetric input asymmetric kernel
67. conv standard 1D
68. conv transposed 3D square input asymmetric kernel
69. conv transposed 2D asymmetric input asymmetric kernel
70. conv transposed 3D asymmetric input square kernel

71. conv transposed 2D asymmetric input square kernel

72. conv transposed 3D asymmetric input asymmetric kernel strided padded grouped

73. conv transposed 3D asymmetric input square kernel strided padded grouped

74. conv transposed 1D dilated

75. conv transposed 2D asymmetric input asymmetric kernel strided grouped padded dilated

76. conv standard 1D dilated strided

77. conv transposed 3D square input square kernel padded dilated strided

78. conv transposed 2D asymmetric input asymmetric kernel padded

79. conv transposed 1D asymmetric input square kernel padded strided dilated

80. conv standard 2D square input asymmetric kernel dilated padded

81. conv transposed 2D asymmetric input square kernel dilated padded strided

82. conv depthwise 2D square input square kernel

83. conv depthwise 2D square input asymmetric kernel

84. conv depthwise 2D asymmetric input square kernel

85. conv depthwise 2D asymmetric input asymmetric kernel

86. conv depthwise separable 2D

87. conv pointwise 2D

88. MinGPTNewGelu

89. cumsum

90. cumprod

91. cumsum reverse

92. cumsum exclusive

93. masked cumsum

94. MSELoss

95. CrossEntropyLoss

96. HuberLoss

97. CosineSimilarityLoss

98. KLDivLoss

99. TripletMarginLoss

100. HingeLoss

## Level 2 Task Names

1. Conv2D ReLU BiasAdd

2. ConvTranspose2d BiasAdd Clamp Scaling Clamp Divide

3. ConvTranspose3d Sum LayerNorm AvgPool GELU

4. Conv2d Mish Mish

5. ConvTranspose2d Subtract Tanh

6. Conv3d Softmax MaxPool MaxPool

7. Conv3d ReLU LeakyReLU GELU Sigmoid BiasAdd

8. Conv3d Divide Max GlobalAvgPool BiasAdd Sum

9. Matmul Subtract Multiply ReLU

10. ConvTranspose2d MaxPool Hardtanh Mean Tanh

11. ConvTranspose2d BatchNorm Tanh MaxPool GroupNorm

12. Gemm Multiply LeakyReLU

13. ConvTranspose3d Mean Add Softmax Tanh Scaling

14. Gemm Divide Sum Scaling

15. ConvTranspose3d BatchNorm Subtract

16. ConvTranspose2d Mish Add Hardtanh Scaling

17. Conv2d InstanceNorm Divide

18. Matmul Sum Max AvgPool LogSumExp LogSumExp

19. ConvTranspose2d GELU GroupNorm

20. ConvTranspose3d Sum ResidualAdd Multiply ResidualAdd

21. Conv2d Add Scale Sigmoid GroupNorm

22. Matmul Scale ResidualAdd Clamp LogSumExp Mish

23. Conv3d GroupNorm Mean

24. Conv3d Min Softmax

25. Conv2d Min Tanh Tanh

26. ConvTranspose3d Add HardSwish

27. Conv3d HardSwish ReLU Softmax Mean

28. BMM InstanceNorm Sum ResidualAdd Multiply

29. Matmul Mish Mish

30. Gemm GroupNorm Hardtanh

31. Conv2d Min Add Multiply

32. Conv2d Scaling Min

33. Gemm Scale BatchNorm

34. ConvTranspose3d LayerNorm GELU Scaling

35. Conv2d Subtract HardSwish MaxPool Mish

36. ConvTranspose2d Min Sum GELU Add

37. Matmul Swish Sum GroupNorm

38. ConvTranspose3d AvgPool Clamp Softmax Multiply

39. Gemm Scale BatchNorm

40. Matmul Scaling ResidualAdd

41. Gemm BatchNorm GELU GroupNorm Mean ReLU

42. ConvTranspose2d GlobalAvgPool BiasAdd LogSumExp Sum Multiply

43. Conv3d Max LogSumExp ReLU

44. ConvTranspose2d Multiply GlobalAvgPool GlobalAvgPool Mean

45. Gemm Sigmoid Sum LogSumExp

46. Conv2d Subtract Tanh Subtract AvgPool

47. Conv3d Mish Tanh

48. Conv3d Scaling Tanh Multiply Sigmoid

49. ConvTranspose3d Softmax Sigmoid

50. ConvTranspose3d Scaling AvgPool BiasAdd Scaling

51. Gemm Subtract GlobalAvgPool LogSumExp GELU ResidualAdd

52. Conv2d Activation BatchNorm

53. Gemm Scaling Hardtanh GELU

54. Conv2d Multiply LeakyReLU GELU

55. Matmul MaxPool Sum Scale

56. Matmul Sigmoid Sum

57. Conv2d ReLU HardSwish

58. ConvTranspose3d LogSumExp HardSwish Subtract Clamp Max

59. Matmul Swish Scaling

60. ConvTranspose3d Swish GroupNorm HardSwish

61. ConvTranspose3d ReLU GroupNorm

62. Matmul GroupNorm LeakyReLU Sum

63. Gemm ReLU Divide

64. Gemm LogSumExp LeakyReLU LeakyReLU GELU GELU

65. Conv2d AvgPool Sigmoid Sum

66. Matmul Dropout Mean Softmax

67. Conv2d GELU GlobalAvgPool

68. Matmul Min Subtract

69. Conv2d HardSwish ReLU

70. Gemm Sigmoid Scaling ResidualAdd

71. Conv2d Divide LeakyReLU

72. ConvTranspose3d BatchNorm AvgPool AvgPool

73. Conv2d BatchNorm Scaling

74. ConvTranspose3d LeakyReLU Multiply LeakyReLU Max

75. Gemm GroupNorm Min BiasAdd

76. Gemm Add ReLU

77. ConvTranspose3d Scale BatchNorm GlobalAvgPool

78. ConvTranspose3d Max Max Sum

79. Conv3d Multiply InstanceNorm Clamp Multiply Max

80. Gemm Max Subtract GELU

81. Gemm Swish Divide Clamp Tanh Clamp

82. Conv2d Tanh Scaling BiasAdd Max

83. Conv3d GroupNorm Min Clamp Dropout

84. Gemm BatchNorm Scaling Softmax

85. Conv2d GroupNorm Scale MaxPool Clamp

86. Matmul Divide GELU

87. Conv2d Subtract Subtract Mish

88. Gemm GroupNorm Swish Multiply Swish

89. ConvTranspose3d MaxPool Softmax Subtract Swish Max

90. Conv3d LeakyReLU Sum Clamp GELU

91. ConvTranspose2d Softmax BiasAdd Scaling Sigmoid

92. Conv2d GroupNorm Tanh HardSwish ResidualAdd LogSumExp

93. ConvTranspose2d Add Min GELU Multiply

94. Gemm BiasAdd Hardtanh Mish GroupNorm

95. Matmul Add Swish Tanh GELU Hardtanh

96. ConvTranspose3d Multiply Max GlobalAvgPool Clamp

97. Matmul BatchNorm BiasAdd Divide Swish

98. Matmul AvgPool GELU Scale Max

99. Matmul GELU Softmax

100. ConvTranspose3d Clamp Min Divide

**Level 3 Task Names**

1. MLP
2. ShallowWideMLP
3. DeepNarrowMLP
4. LeNet5
5. AlexNet
6. GoogleNetInceptionModule
7. GoogleNetInceptionV1
8. ResNetBasicBlock
9. ResNet18
10. ResNet101
11. VGG16
12. VGG19
13. DenseNet121TransitionLayer
14. DenseNet121DenseBlock
15. DenseNet121
16. DenseNet201
17. SqueezeNetFireModule
18. SqueezeNet
19. MobileNetV1
20. MobileNetV2
21. EfficientNetMBConv
22. EfficientNetB0
23. EfficientNetB1
24. EfficientNetB2
25. ShuffleNetUnit
26. ShuffleNet
27. RegNet
28. VisionTransformer
29. SwinMLP
30. SwinTransformerV2
31. VisionAttention
32. ConvolutionalVisionTransformer
33. VanillaRNN
34. VanillaRNNHidden
35. LTSM
36. LTSMHn
37. LTSMCn
38. LTSMBidirectional
39. GRU
40. GRUHidden
41. GRUBirectional
42. GRUBidirectionalHidden
43. MinGPTCausalAttention
44. MiniGPTBlock
45. UNetSoftmax
46. NetVladWithGhostClusters
47. NetVladNoGhostClusters
48. Mamba2ReturnY
49. Mamba2ReturnFinalState
50. ReLUSelfAttention

### K.2. Level 2 Synthetic Generation

We want to especially elaborate on the design and construction of Level 2 problems. The construction of level 2 is done by randomly picking one main operator and 2 to 5 epilogue operators, for a total of 3 to 6 operators per task. Figure 16, which highlights the relative frequency of different task sizes.

The mainloop operators include Matmul, BMM, Conv2d, Conv3d, ConvTranspose2d, and ConvTranspose3d, as shown in Figure 17.

The epilogue operators are divided into different classes:

- **activations**: ReLU, Sigmoid, Tanh, LeakyReLU, GELU, Swish, Softmax, Mish, Hardtanh, HardSwish
- **element-wise ops**: Add, Multiply, Subtract, Divide, Clamp, Scale, ResidualAdd
- **normalizations**: BatchNorm, LayerNorm, InstanceNorm, GroupNorm
- **pooling**: MaxPool, AvgPool, GlobalAvgPool
- **bias**: BiasAdd
- **reductions**: Sum, Mean, Max, Min, LogSumExp
- **others**: Dropout, ResidualAdd, Scaling

The distribution of epilogue operators in Level 2 tasks is illustrated in Figure 18.

## L. Kernel Fusion Investigation

Kernel Fusion is an important optimization deployed in optimizing Deep Learning programs. By fusing multiple operations into a single kernel, the optimized program can reduce memory traffic which improves performance especially as data movement is expensive. Just to reiterate, in KernelBench the model has full flexibility to decide what subset of operators in the PyTorch reference to optimize and fuse. We believe this is one of the crucial abilities when a model is given distinct or new architectures in the real-world setting.

KernelBench's 3-level categorization helps disentangle fusion decisions and kernel generation. Level 1 problems (single operators) only test the model's ability to write optimized kernels; Level 2 and 3 problems are designed to additionally evaluate the model's ability to identify and leverage fusion opportunities.

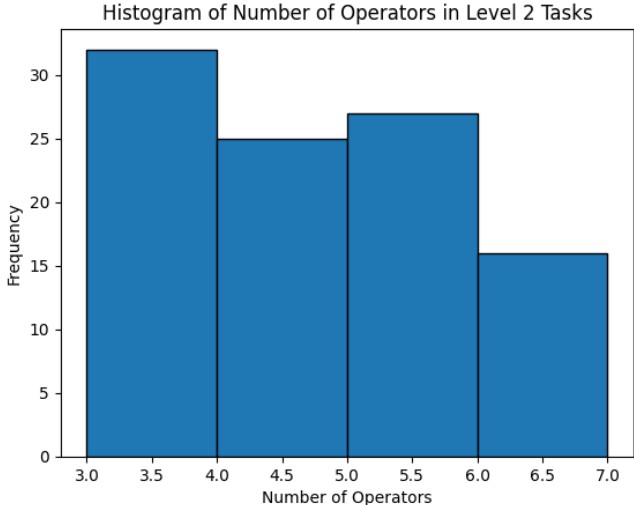

*Figure 16.* Histogram of the number of operators per task in Level 2.

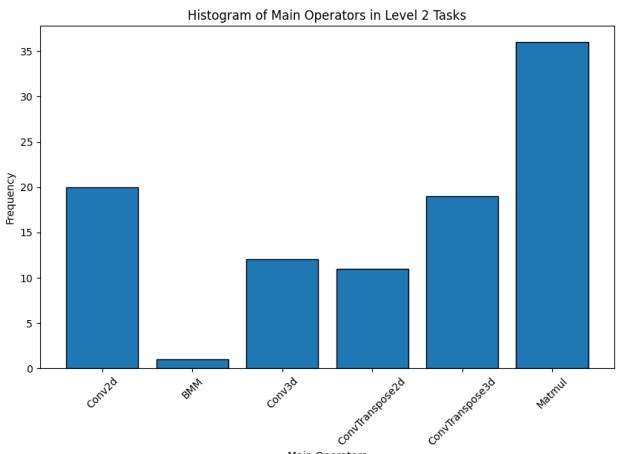

*Figure 17.* Histogram of main operators in Level 2 tasks.

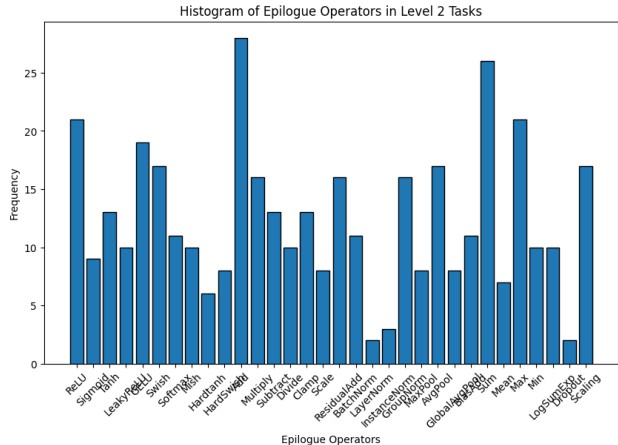

*Figure 18.* Histogram of epilogue operators in Level 2 tasks.

From the baseline results in Section 4.1, we investigate the particular fusion choices made by the language model and how these influence the runtime of these kernels. We focus specifically on Level 2 tasks, where fusions can be applied to all problems. They are composed with one mainloop (e.g. conv, matmul) and 2 - 5 epilogue operations (non-linearities, reductions, etc). Figure 17 and 18 shows distribution of these operators in the 100 Level programs.

We focus our analysis on the behavior of DeepSeek-R1, which has the best performance on Level 2 problems as shown in Table 1.

### L.1. LM-generated Kernel Fusions on Level 2

We manually inspected all 100 generated programs and note down the fusion patterns. Model always attempts to generate 1-2 fused kernels per problem: 88 problems have 1 fused kernels, and 12 problems contain 2 fused kernels.

One interesting point of analysis is how aggressively the LM chooses to fuse operators. In Figure 19, we plot a histogram to inspect the size (in terms of ratio of total operators) of generated fused kernels in Level 2. Specifically, for each kernel we count number of operators fused compared to the total number of operators in the program. For example, suppose a program has 6 operators, and the LM chooses to fuse the first two and last four operators. Then we add $33.3\%$ and $66.6\%$ to our list.

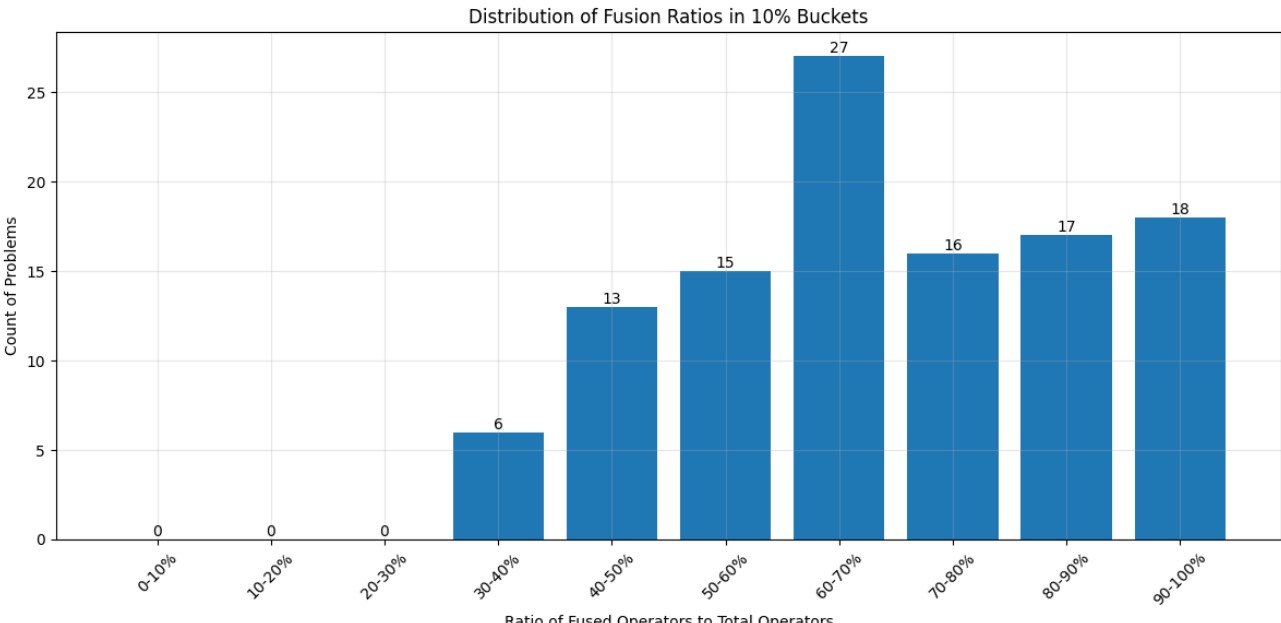

*Figure 19.* **Distribution of Fusions by DeepSeek-R1 on Level 2 Problems**. We bucket (in increments of 10%) all fusions performed by DeepSeek-R1 on Level 2 based on the percentage of operators it fused in the full task. We inspect the fused kernels in terms of ratio of total operators of that program. Specifically, for each kernel we count number of operators fused compared to the total number of operators in the program.

We observe that the models *attempt* to fuse more than half the operators on average. Only 18% of programs fuse all operators in a program into a single kernel.

### L.2. Understanding Fusion in Slower Kernels

To understand the quality of the fusion decision and how poor fusion leads to low performance, we analyzed the DeepSeek-R1 Level 2 kernels that were slower than PyTorch Eager (as shown in Table 17) and drew two observations:

1. Main loop operators (e.g., `Conv`) were not fused with epilogue operators.

2. The model's attempt to fuse main loop operators (e.g., `GEMM` + other ops) was not faster than launching highly optimized `CuBLAS` kernels.

### L.3. Comparison with Auto-Fusion techniques

We focus comparison on widely-adopted `torch.compile` in PyTorch 2 (Ansel et al., 2024) which employs an auto-fusion policy over TorchInductor's define-by-run IR. In Table 17, we can see the difference in fusion decisions of R1 and torch.compile in Table XX. `torch.compile` often creates sophisticated fusion patterns, by breaking Convolutions or GroupNorm into smaller multi-pass kernels that compute parietal results and statistics in parallel, something R1-kernels rarely do with room for improvement. However, 37% of R1-kernels actually are faster than `torch.compile`, as `torch.compile` has to fit pre-existing defined templates, whereas R1 can generate more custom optimizations.

| Problem | Speedup | Sequence of Operators | Total | LLM Generated | `torch.compile` |
|---|---|---|---|---|---|
| 63 | 0.1810 | [gemm, relu, divide] | 3 | [(gemm, relu, divide)] | [gemm, (relu, div)] |
| 59 | 0.1996 | [matmul, swish, scaling] | 3 | [(matmul, swish, scaling)] | [gemm, (sigmoid, mul)] |
| 35 | 0.2331 | [conv2d, subtract, hardswish, maxpool, mish] | 5 | [conv2d, (subtract, hardswish), (maxpool, mish)] | [convolution, convolution, convolution, (convolution,sub,hardswish), (convolution,sub, hardswish,max_pool2d_with_indices,mish)] |
| 27 | 0.5501 | [conv3d, hardswish, relu, softmax, mean] | 5 | [conv, (hardswish, relu), (softmax, mean)] | [convolution, (convolution,hardswish,relu,_softmax), (convolution,hardswish,relu,_softmax,mean)] |
| 80 | 0.6156 | [gemm, max, subtract, gelu] | 4 | [gemm, (max, subtract, gelu)] | [gemm, (max,mean,sub,gelu)] |
| 37 | 0.6922 | [matmul, swish, sum, groupnorm] | 4 | [(matmul, swish, sum, groupnorm)] | [gemm, native_group_norm, native_group_norm] |
| 54 | 0.7394 | [conv2d, multiply, leakyrelu, gelu] | 4 | [conv, (multiply, leakyrelu, gelu)] | [convolution, convolution, convolution, (convolution,mul,leaky_relu,gelu)] |
| 69 | 0.7433 | [conv2d, hardswish, relu] | 3 | [conv, (hardswish, relu)] | [convolution, convolution, convolution, (convolution,hardswish,relu)] |
| 25 | 0.7701 | [conv2d, min, tanh, tanh] | 4 | [conv, (min, tanh, tanh)] | [convolution, convolution, convolution, (convolution,min,tanh)] |
| 36 | 0.7859 | [convtranspose2d, min, sum, gelu, add] | 5 | [convtranspose, (min, sum, gelu, add)] | [convolution, convolution, convolution, (convolution,min), sum, (gelu,add)] |
| 71 | 0.8231 | [conv2d, divide, leakyrelu] | 3 | [conv2d, (divide, leakyrelu)] | [convolution, convolution, convolution, (convolution,div,leaky_relu)] |
| 32 | 0.8399 | [conv2d, scaling, min] | 3 | [conv, (scaling, min)] | [convolution, convolution, convolution, (convolution,mul,min)] |
| 68 | 0.8411 | [matmul, min, subtract] | 3 | [matmul, (min, subtract)] | [gemm, (minimum,sub)] |
| 64 | 0.8595 | [gemm, logsumexp, leakyrelu, leakyrelu, gelu, gelu] | 6 | [gemm, logsumexp, (leakyrelu, leakyrelu, gelu, gelu)] | [gemm, (logsumexp,leaky_relu,gelu)] |
| 65 | 0.8615 | [conv2d, avgpool, sigmoid, sum] | 4 | [conv, (avgpool, sigmoid, sum)] | [convolution, convolution, convolution, convolution, (convolution,avg_pool2d,sigmoid,sum)] |
| 86 | 0.8622 | [matmul, divide, gelu] | 3 | [matmul, (divide, gelu)] | [gemm, (div,gelu)] |
| 46 | 0.8671 | [conv2d, subtract, tanh, subtract, avgpool] | 5 | [(conv2d), (subtract, tanh, subtract), (avgpool)] | [convolution, convolution, convolution, (convolution,sub,tanh), (convolution,sub,tanh,avg_pool2d)] |
| 12 | 0.8715 | [gemm, multiply, leakyrelu] | 3 | [gemm, (multiply, leakyrelu)] | [gemm, (mul,leaky_relu)] |
| 87 | 0.8930 | [conv2d, subtract, subtract, mish] | 4 | [conv, (subtract, subtract, mish)] | [convolution, convolution, convolution, (convolution,sub,mish)] |
| 47 | 0.9063 | [conv3d, mish, tanh] | 3 | [conv, (mish, tanh)] | [convolution, (convolution,mish,tanh)] |
| 82 | 0.9092 | [conv2d, tanh, scaling, biasadd, max] | 5 | [conv2d, (tanh, scaling, biasadd), max] | [convolution, convolution, convolution, (convolution,tanh,mul,add), max_pool2d_with_indices] |
| 52 | 0.9380 | [conv2d, softplus, tanh, multiply, batchnorm] | 5 | [conv2d, (softplus, tanh, multiply), batchnorm] | [convolution, convolution, convolution, (convolution,softplus,tanh,mul, _native_batch_norm_legit_functional), (convolution,softplus,tanh,mul, _native_batch_norm_legit_functional), (convolution,softplus,tanh,mul, _native_batch_norm_legit_functional), (convolution,softplus,tanh,mul, _native_batch_norm_legit_functional), add] |
| 38 | 0.9515 | [convtranspose3d, avgpool, clamp, softmax, multiply] | 5 | [(convtranspose3d), (avgpool), (clamp, softmax, multiply)] | [convolution, convolution, (convolution,avg_pool3d,clamp), _softmax, (_softmax,mul)] |
| 9 | 0.9657 | [matmul, subtract, multiply, relu] | 4 | [matmul, (subtract, multiply, relu)] | [gemm, (sub,mul,relu)] |
| 78 | 0.9701 | [convtranspose3d, max, max, sum] | 4 | [convtranspose, (max, max, sum)] | [convolution, convolution, (convolution,max_pool3d_with_indices), max_pool3d_with_indices, sum] |
| 5 | 0.9909 | [convtranspose2d, subtract, tanh] | 3 | [convtranspose, (subtract, tanh)] | [convolution, convolution, convolution, (convolution,sub,tanh)] |

*Table 17.* **Fusion Decisions of Level 2 Problems where LM-Generated code was slower than PyTorch Eager**. Here we investigate the programs generated by DeepSeek-R1 on Level 2 KernelBench Problems, specifically those that are slower than PyTorch Eager (sorted by lower speedup). We list the sequence of operators in the program, and fusion decisions made by R1, comparing that with `torch.compile` generated kernels.

## M. Alternate Input Representations

We explore alternative input representations of the KernelBench problems to the Language Model other than a PyTorch reference.

For the PyTorch example as shown in Section C.1 that describes the simple operation `a+b` used in our baseline prompt.

### M.1. Natural language representation

Natural Language might be easiest to start for developer from a chat interface, but could suffer from ambiguity, especially for complex operations. We provide a specification in natural language that is as verbose as possible.

```
1  This program should demonstrate element-wise tensor addition.
2  This prorgam takes two input tensors of the same shape and performs element-wise addition
       ↪ between them. Each input tensor has a batch size of 1 and contains 128 features.
3
4  The model has no learnable parameters and simply acts as a direct addition operation
       ↪ between its inputs. Both input tensors are expected to be on the GPU (CUDA device).
5
6  Operation: Element-wise addition
7  Input shapes: (1, 128) for both inputs
8  Output shape: (1, 128)
9  Parameters: None
10 Device: CUDA (GPU)
11 Purpose: Simple demonstration of tensor addition
```

### M.2. Directed-acyclic graph (DAG) representation

A DAG representation explicitly layouts all transforms and operators in the program, which might be more beneficial for the model to conduct kernel fusion (fusing nodes in the graph).

**ONNX Graph Representation**[1]

```
1  Here is a graph representation of the PyTorch program.
2
3  Inputs: ['input', 'b']
4  Outputs: ['add']
5
6  Nodes:
7  Op Type: Add
8    Inputs: ['input', 'b']
9    Outputs: ['add']
10   Attributes: []
11
12
13 To be specific, the input we will provide will be:
14   a: a tensor of shape (1, 128) of FP32
15   b: a tensor of shape (1, 128) of FP32
```

**Torch FX Representation**[2]

```
1  Here is a DAG representation of the program in tabular form
2  opcode          name    target                  args     kwargs
3  -------------   ------  ----------------------  ------   --------
4  placeholder     a       a                       ()       {}
5  placeholder     b       b                       ()       {}
6  call_function   add     <built-in function add> (a, b)   {}
7  output          output  output                  (add,)   {}
```

---

[1]ONNX (Open Neural Network Exchange) is a standard representation of neural network programs, which could exported from PyTorch, https://onnx.ai/onnx/intro/

[2]PyTorch FX is a toolkit that help extract symbolic trace and intermediate representations of PyTorch Programs, https://pytorch.org/docs/stable/fx.html

```
 8
 9
10  To be specific, the input we will provide will be:
11    a: a tensor of shape (1, 128) of FP32
12    b: a tensor of shape (1, 128) of FP32
```

### M.3. Result Testing on Representative Problem

We use the same format for the baseline prompt in C.1, instead of updating the task from PyTorch to PyTorch to CUDA, to Alterative Format (Natural Language, DAG) representation to PyTorch + CUDA. We replace the pair of <PyTorch, CUDA> one-shot example with <Add example in desired representation, CUDA>. Table 18 shows the result when testing this on a representative problem in KernelBench, specifically Level 2 19_ConvTranspose2d_GELU_GroupNorm_. We chose this example because it was correct when using the PyTorch representation, and represents a semantically common pattern in Level 2. We hope test on this example give us insights on the strengths and weaknesses of various representations.

| Input Representation | Execution Result | Analysis |
|---|---|---|
| PyTorch | Correct | Implement Operator in CUDA, No Fusion |
| Natural Language | Compilation Error / Logical Error | Wrong initialization, Did not use weight parameters- |
| DAG (ONNX) | Output Value Mismatch | Implement Operator in CUDA, Wrong Normalization |
| DAG (torch FX) | Output Value Mismatch | Attempted Fusion, incorrect group convolution |

*Table 18.* **Degraded performance with non-PyTorch input specifications on example KernelBench problem.** Evaluation of Level 2 problem 19_ConvTranspose2d_GELU_GroupNorm_ using OpenAI GPT-4o and o1 (medium reasoning effort), across various alternative input representations, best of 5 sampling.

As we observed, when replacing the PyTorch representation with other forms of specifications, the model suffers from ambiguity in its specifications. With natural language especially, despite giving it a verbose description, the model gets confused and creates a program that has wrong initialization or neglects to use weight parameters altogether. DAG representations, especially torch FX, helped model to discover fusion; but going from graph to kernel add additional challenge to implement a functionally correct program. DAG representations might be a beneficial complement to the reference program in PyTorch, though they are not a sufficient replacement.

We chose the PyTorch code reference as a precise, unambiguous, and verifiable input for KernelBench, which is crucial for rigorous benchmarking. Using PyTorch often mirrors the experience of AI researchers (as noted in Section 1, 3) - - who start with PyTorch and then optimize (e.g., popular FlashAttention (Kim, 2024), Mamba (Dao & Gu, 2024b) repositories follow the workflow of PyTorch code with some inline CUDA kernels). The PyTorch reference is also an unambiguous source for verifying generated code for correctness. It is also worth noting that LLMs are fine-tuned on sequence data, such as code, and PyTorch is a natural choice for benchmarking and setup as a code translation task. Future work could consider alternative representations.

# N. Performance Degradation Analysis

Table 19 presents the best speedup achieved by any evaluated model for each problem instance, sorted first by Level and then by ascending speedup (defined as generated code wall clock time over baseline torch reference time). We note the following observations and include some examples of LLM-generated code:

1. Many **foundational operations** in level 1, particularly matrix multiplications (matmuls) and convolutions (convs), **frequently result in significant slowdowns** compared to the highly-optimized proprietary libraries like cuDNN used by PyTorch's baseline that use more advanced hardware features. For instance, even the best model achieved only 0.0105 speedup for 69_conv_transposed_2D_asymmetric_input_asymmetric_kernel and 0.1011 for 11_4D_tensor_matrix_multiplication.

2. While models sometimes correctly identify opportunities for fusion in Level 2 tasks, the resulting kernel performance is often poor. This suggests that **even if the fusion strategy is conceptually sound, the LLM's implementation of the core fused components (like the matmul or convolution within the sequence) is inefficient**, negating any potential benefits from reduced memory access; examples like 63_Gemm_ReLU_Divide (0.1849 speedup) and 59_Matmul_Swish_Scaling (0.1996 speedup) illustrate this issue where complex fusions are much slower than the baseline — both cases successfully fuse all three operators into a single kernel but they're disadvantaged by the matmul implementation.

3. We observe patterns suggesting that LLMs might generate a **custom CUDA kernel for a relatively minor operation within a larger sequence**. While potentially optimizing that single step locally, this can **disrupt PyTorch's native fusion capabilities** (like TorchInductor), preventing more holistic, graph-level optimizations across the sequence that could have yielded greater overall performance gains. One example we found is 4_LeNet5 in level 3. The model generated a custom implementation for ReLU that resulted in 0.6451 speedup.

The generated code for the example problems listed above is included below:

### N.1. Code Examples

Note all speedups are against PyTorch Eager.

**Slow core operator: 0.0105 speedup for 69_conv_transposed_2D_asymmetric_input_asymmetric_kernel**

```
__global__ void conv_transpose2d_kernel(
    const float* input,
    const float* weight,
    float* output,
    const int batch_size,
    const int in_channels,
    const int out_channels,
    const int height_in,
    const int width_in,
    const int height_out,
    const int width_out,
    const int kernel_h,
    const int kernel_w,
    const int stride_h,
    const int stride_w,
    const int padding_h,
    const int padding_w,
    const int out_padding_h,
    const int out_padding_w,
    const int dilation_h,
    const int dilation_w,
    const int groups
) {
    const int h_out = blockIdx.y;
    const int w_out = blockIdx.x;
    const int batch_idx = blockIdx.z / out_channels;
    const int out_ch = blockIdx.z % out_channels;
```

```
28
29      if (h_out >= height_out || w_out >= width_out || batch_idx >= batch_size)
30          return;
31
32      float sum = 0.0f;
33      const int in_ch_per_group = in_channels / groups;
34      const int out_ch_per_group = out_channels / groups;
35      const int group = out_ch / out_ch_per_group;
36
37      for (int kh = 0; kh < kernel_h; kh++) {
38          for (int kw = 0; kw < kernel_w; kw++) {
39              const int h_in = (h_out + padding_h - kh * dilation_h) / stride_h;
40              const int w_in = (w_out + padding_w - kw * dilation_w) / stride_w;
41
42              if (h_in >= 0 && h_in < height_in && w_in >= 0 && w_in < width_in) {
43                  for (int in_ch = group * in_ch_per_group;
44                       in_ch < (group + 1) * in_ch_per_group;
45                       in_ch++) {
46
47                      const float input_val = input[
48                          batch_idx * in_channels * height_in * width_in +
49                          in_ch * height_in * width_in +
50                          h_in * width_in +
51                          w_in
52                      ];
53
54                      const float weight_val = weight[
55                          in_ch * out_ch_per_group * kernel_h * kernel_w +
56                          (out_ch % out_ch_per_group) * kernel_h * kernel_w +
57                          kh * kernel_w +
58                          kw
59                      ];
60
61                      sum += input_val * weight_val;
62                  }
63              }
64          }
65      }
66
67      output[
68          batch_idx * out_channels * height_out * width_out +
69          out_ch * height_out * width_out +
70          h_out * width_out +
71          w_out
72      ] = sum;
```

**Slow core operator: 0.1011 speedup for 11_4D_tensor_matrix_multiplication**

```
1   __global__ void tensor_matmul_kernel(const float* A, const float* B, float* C, int b, int
    ↪ i, int j, int l, int k) {
2       int idx_b = blockIdx.x;
3       int idx_i = blockIdx.y;
4       int idx_j = blockIdx.z;
5       int idx_k = threadIdx.x;
6
7       if (idx_b < b && idx_i < i && idx_j < j && idx_k < k) {
8           float sum = 0.0f;
9           for (int idx_l = 0; idx_l < l; ++idx_l) {
10              sum += A[idx_b * i * j * l + idx_i * j * l + idx_j * l + idx_l] * B[idx_l * k
                  ↪ + idx_k];
11          }
12          C[idx_b * i * j * k + idx_i * j * k + idx_j * k + idx_k] = sum;
13      }
14  }
```

**Good fusion pattern but bad implementation: 0.1849 speedup for 63_Gemm_ReLU_Divide**

```
1   __global__ void linear_relu_div_kernel(
2       const float* __restrict__ x,
3       const float* __restrict__ w,
4       const float* __restrict__ b,
5       float* __restrict__ out,
6       int batch_size,
7       int in_features,
8       int out_features,
9       float divisor
10  ) {
11      int idx = blockIdx.x * blockDim.x + threadIdx.x;
12      int total = batch_size * out_features;
13      if (idx < total) {
14          int i = idx / out_features;
15          int j = idx % out_features;
16          float val = 0.0f;
17          for (int k = 0; k < in_features; k++) {
18              val += x[i * in_features + k] * w[j * in_features + k];
19          }
20          val += b[j];
21          val = val > 0.f ? val : 0.f;
22          val /= divisor;
23          out[idx] = val;
24      }
25  }
```

**Good fusion pattern but bad implementation: 0.1996 speedup for 59_Matmul_Swish_Scaling**

```
1   __global__ void fused_linear_swish_scale_kernel(
2       const float* input, const float* weight, const float* bias,
3       float* output, int B, int IN, int OUT, float scale) {
4
5       int tid = blockIdx.x * blockDim.x + threadIdx.x;
6       if(tid >= B * OUT) return;
7
8       int row = tid / OUT;
9       int col = tid % OUT;
10
11      float sum = 0.0f;
12      for(int i=0; i<IN; ++i) {
13          sum += input[row*IN + i] * weight[col*IN + i];
14      }
15      sum += bias[col];
16
17      float swish = sum * (1.0f / (1.0f + expf(-sum)));
18      output[tid] = swish * scale;
19  }
```

**Minor custom kernel impacting framework's inherent fusion ability: 0.6451 speedup for 4_LeNet5**

```
1   relu_source = r"""
2   #include <torch/extension.h>
3   #include <cuda_runtime.h>
4
5   __global__ void relu_kernel(const float* in, float* out, int size) {
6       int idx = blockIdx.x * blockDim.x + threadIdx.x;
7       if (idx < size) {
8           float val = in[idx];
9           out[idx] = val > 0.0f ? val : 0.0f;
10      }
11  }
12
```

```
13  torch::Tensor relu_cuda(torch::Tensor input) {
14      auto size = input.numel();
15      auto out = torch::zeros_like(input);
16
17      const int block_size = 256;
18      const int grid_size = (size + block_size - 1) / block_size;
19
20      relu_kernel<<<grid_size, block_size>>>(
21          input.data_ptr<float>(),
22          out.data_ptr<float>(),
23          (int)size
24      );
25
26      return out;
27  }
28  """
29
30  relu_cpp_source = r"torch::Tensor relu_cuda(torch::Tensor input);"
31
32  # Compile the inline custom ReLU operator
33  relu_module = load_inline(
34      name="custom_relu",
35      cpp_sources=relu_cpp_source,
36      cuda_sources=relu_source,
37      functions=["relu_cuda"],
38      verbose=False
39  )
40
41  class ModelNew(nn.Module):
42      def __init__(self, num_classes):
43          """
44          Optimized LeNet-5 architecture with a custom CUDA-based ReLU.
45          """
46          super(ModelNew, self).__init__()
47
48          # Convolutional layers
49          self.conv1 = nn.Conv2d(in_channels=1, out_channels=6, kernel_size=5, stride=1)
50          self.conv2 = nn.Conv2d(in_channels=6, out_channels=16, kernel_size=5, stride=1)
51
52          # Fully connected layers
53          self.fc1 = nn.Linear(in_features=16*5*5, out_features=120)
54          self.fc2 = nn.Linear(in_features=120, out_features=84)
55          self.fc3 = nn.Linear(in_features=84, out_features=num_classes)
56
57          # Reference to custom ReLU extension
58          self.custom_relu = relu_module.relu_cuda
59
60      def forward(self, x):
61          # conv1 + custom relu + max pool
62          x = self.conv1(x)
63          x = self.custom_relu(x)
64          x = F.max_pool2d(x, kernel_size=2, stride=2)
65
66          # conv2 + custom relu + max pool
67          x = self.conv2(x)
68          x = self.custom_relu(x)
69          x = F.max_pool2d(x, kernel_size=2, stride=2)
70
71          # Flatten
72          x = x.view(-1, 16*5*5)
73
74          # fc1 + custom relu
75          x = self.fc1(x)
76          x = self.custom_relu(x)
77
```

```
78          # fc2 + custom relu
79          x = self.fc2(x)
80          x = self.custom_relu(x)
81
82          # final fc
83          x = self.fc3(x)
84          return x
```

Table 19: Best achieved speedup per problem instance across evaluated models, categorized by difficulty level and sorted by ascending speedup

| Level | Speedup | Problem | Model |
|---|---|---|---|
| 1 | 0.0105 | 69_conv_transposed_2D__asymmetric_input__a... | claude-3.5-sonnet |
| 1 | 0.0316 | 78_conv_transposed_2D_asymmetric_input_asymm... | claude-3.5-sonnet |
| 1 | 0.0442 | 35_GroupNorm_ | openai-o1 |
| 1 | 0.0597 | 68_conv_transposed_3D__square_input__asymm... | deepseek-r1 |
| 1 | 0.0734 | 40_LayerNorm | llama-3.1-405b |
| 1 | 0.1011 | 11_4D_tensor_matrix_multiplication | deepseek-V3 |
| 1 | 0.1151 | 91_cumsum_reverse | claude-3.5-sonnet |
| 1 | 0.1196 | 16_Matmul_with_transposed_A | claude-3.5-sonnet |
| 1 | 0.1198 | 18_Matmul_with_transposed_both | openai-o1 |
| 1 | 0.1332 | 93_masked_cumsum | openai-o1 |
| 1 | 0.1336 | 77_conv_transposed_3D_square_input_square_k... | deepseek-r1 |
| 1 | 0.1553 | 6_Matmul_with_large_K_dimension_ | claude-3.5-sonnet |
| 1 | 0.1576 | 1_Square_matrix_multiplication_ | claude-3.5-sonnet |
| 1 | 0.1594 | 2_Standard_matrix_multiplication_ | claude-3.5-sonnet |
| 1 | 0.1681 | 97_CosineSimilarityLoss | deepseek-V3 |
| 1 | 0.1871 | 80_conv_standard_2D_square_input_asymmetric:. | deepseek-r1 |
| 1 | 0.1893 | 10_3D_tensor_matrix_multiplication | claude-3.5-sonnet |
| 1 | 0.1895 | 17_Matmul_with_transposed_B | deepseek-r1 |
| 1 | 0.1979 | 13_Matmul_for_symmetric_matrices | claude-3.5-sonnet |
| 1 | 0.2083 | 8_Matmul_with_irregular_shapes_ | claude-3.5-sonnet |
| 1 | 0.2097 | 62_conv_standard_2D__square_input__asymmet... | deepseek-r1 |
| 1 | 0.2164 | 66_conv_standard_3D__asymmetric_input__asy... | deepseek-r1 |
| 1 | 0.2523 | 3_Batched_matrix_multiplication | deepseek-r1 |
| 1 | 0.2584 | 54_conv_standard_3D__square_input__square:. | deepseek-r1 |
| 1 | 0.274 | 86_conv_depthwise_separable_2D | deepseek-r1 |
| 1 | 0.3152 | 24_LogSoftmax | deepseek-r1 |
| 1 | 0.4097 | 89_cumsum | openai-o1 |
| 1 | 0.4179 | 7_Matmul_with_small_K_dimension_ | deepseek-r1 |
| 1 | 0.4479 | 23_Softmax | deepseek-r1 |
| 1 | 0.4807 | 15_Matmul_for_lower_triangular_matrices | deepseek-V3 |
| 1 | 0.5271 | 81_conv_transposed_2D_asymmetric_input_squar... | deepseek-r1 |
| 1 | 0.5497 | 87_conv_pointwise_2D | openai-o1 |
| 1 | 0.5529 | 14_Matmul_for_upper_triangular_matrices | claude-3.5-sonnet |
| 1 | 0.5901 | 9_Tall_skinny_matrix_multiplication_ | claude-3.5-sonnet |

Table 19: Best achieved speedup per problem instance across evaluated models, categorized by difficulty level and sorted by ascending speedup (continued)

| Level | Speedup | Problem | Model |
|-------|---------|---------|-------|
| 1 | 0.6027 | 47_Sum_reduction_over_a_dimension | deepseek-r1 |
| 1 | 0.6105 | 19_ReLU | deepseek-V3 |
| 1 | 0.6908 | 5_Matrix_scalar_multiplication | deepseek-r1 |
| 1 | 0.6915 | 48_Mean_reduction_over_a_dimension | claude-3.5-sonnet |
| 1 | 0.6953 | 44_Average_Pooling_1D | deepseek-r1 |
| 1 | 0.7117 | 27_SELU_ | claude-3.5-sonnet |
| 1 | 0.7125 | 26_GELU_ | deepseek-r1 |
| 1 | 0.7273 | 22_Tanh | deepseek-r1 |
| 1 | 0.7349 | 31_ELU | claude-3.5-sonnet |
| 1 | 0.7375 | 28_HardSigmoid | deepseek-r1 |
| 1 | 0.7756 | 20_LeakyReLU | deepseek-r1 |
| 1 | 0.7809 | 50_Product_reduction_over_a_dimension | deepseek-r1 |
| 1 | 0.8112 | 29_Softplus | deepseek-r1 |
| 1 | 0.8117 | 34_InstanceNorm | gpt-4o |
| 1 | 0.83 | 39_L2Norm_ | deepseek-r1 |
| 1 | 0.8622 | 67_conv_standard_1D | deepseek-r1 |
| 1 | 0.8702 | 21_Sigmoid | deepseek-r1 |
| 1 | 0.872 | 46_Average_Pooling_3D | deepseek-r1 |
| 1 | 0.8808 | 32_HardTanh | claude-3.5-sonnet |
| 1 | 0.8942 | 85_conv_depthwise_2D_asymmetric_input_asymme... | deepseek-r1 |
| 1 | 0.9352 | 94_MSELoss | deepseek-V3 |
| 1 | 0.9439 | 96_HuberLoss | deepseek-r1 |
| 1 | 0.9543 | 4_Matrix_vector_multiplication_ | deepseek-r1 |
| 1 | 0.9671 | 38_L1Norm_ | gpt-4o |
| 1 | 0.9737 | 90_cumprod | deepseek-r1 |
| 1 | 0.9842 | 52_Argmin_over_a_dimension | deepseek-r1 |
| 1 | 0.9859 | 45_Average_Pooling_2D | openai-o1 |
| 1 | 1.0022 | 65_conv_transposed_2D__square_input__asymm... | openai-o1 |
| 1 | 1.0084 | 37_FrobeniusNorm_ | gpt-4o |
| 1 | 1.0115 | 98_KLDivLoss | deepseek-V3 |
| 1 | 1.0131 | 25_Swish | openai-o1 |
| 1 | 1.0259 | 53_Min_reduction_over_a_dimension | deepseek-r1 |
| 1 | 1.0308 | 49_Max_reduction_over_a_dimension | gpt-4o |
| 1 | 1.0697 | 100_HingeLoss | llama-3.1-70b |
| 1 | 1.093 | 51_Argmax_over_a_dimension | openai-o1 |
| 1 | 1.116 | 42_Max_Pooling_2D | openai-o1 |
| 1 | 1.1386 | 41_Max_Pooling_1D | openai-o1 |
| 1 | 1.1417 | 76_conv_standard_1D_dilated_strided__ | deepseek-r1 |

Table 19: Best achieved speedup per problem instance across evaluated models, categorized by difficulty level and sorted by ascending speedup (continued)

| Level | Speedup | Problem | Model |
|---|---|---|---|
| 1 | 1.1545 | 43_Max_Pooling_3D | deepseek-r1 |
| 1 | 1.185 | 83_conv_depthwise_2D_square_input_asymmetric... | deepseek-r1 |
| 1 | 1.2706 | 64_conv_transposed_1D | deepseek-r1 |
| 1 | 1.4855 | 79_conv_transposed_1D_asymmetric_input_squar... | deepseek-r1 |
| 1 | 1.5 | 30_Softsign | deepseek-r1 |
| 1 | 1.5319 | 33_BatchNorm | deepseek-r1 |
| 1 | 1.7083 | 36_RMSNorm_ | openai-o1 |
| 1 | 1.9202 | 95_CrossEntropyLoss | claude-3.5-sonnet |
| 1 | 1.9637 | 99_TripletMarginLoss | deepseek-r1 |
| 1 | 4.1341 | 88_MinGPTNewGelu | deepseek-r1 |
| 1 | 13.2243 | 12_Matmul_with_diagonal_matrices_ | claude-3.5-sonnet |
| 2 | 0.1849 | 63_Gemm_ReLU_Divide | openai-o1 |
| 2 | 0.1996 | 59_Matmul_Swish_Scaling | deepseek-r1 |
| 2 | 0.3008 | 33_Gemm_Scale_BatchNorm | openai-o1 |
| 2 | 0.3564 | 94_Gemm_BiasAdd_Hardtanh_Mish_GroupNorm | openai-o1 |
| 2 | 0.4642 | 66_Matmul_Dropout_Mean_Softmax | openai-o1 |
| 2 | 0.4912 | 44_ConvTranspose2d_Multiply_GlobalAvgPool_Glob... | openai-o1 |
| 2 | 0.5501 | 27_Conv3d_HardSwish_ReLU_Softmax_Mean | deepseek-r1 |
| 2 | 0.5672 | 67_Conv2d_GELU_GlobalAvgPool | openai-o1 |
| 2 | 0.6156 | 80_Gemm_Max_Subtract_GELU | deepseek-r1 |
| 2 | 0.6777 | 4_Conv2d_Mish_Mish | openai-o1 |
| 2 | 0.6922 | 37_Matmul_Swish_Sum_GroupNorm | deepseek-r1 |
| 2 | 0.7181 | 85_Conv2d_GroupNorm_Scale_MaxPool_Clamp | openai-o1 |
| 2 | 0.7433 | 69_Conv2d_HardSwish_ReLU | deepseek-r1 |
| 2 | 0.7451 | 35_Conv2d_Subtract_HardSwish_MaxPool_Mish | openai-o1 |
| 2 | 0.7701 | 25_Conv2d_Min_Tanh_Tanh | deepseek-r1 |
| 2 | 0.7712 | 21_Conv2d_Add_Scale_Sigmoid_GroupNorm | openai-o1 |
| 2 | 0.7859 | 36_ConvTranspose2d_Min_Sum_GELU_Add | deepseek-r1 |
| 2 | 0.7915 | 54_Conv2d_Multiply_LeakyReLU_GELU | openai-o1 |
| 2 | 0.8231 | 71_Conv2d_Divide_LeakyReLU | deepseek-r1 |
| 2 | 0.8377 | 23_Conv3d_GroupNorm_Mean | openai-o1 |
| 2 | 0.8399 | 32_Conv2d_Scaling_Min | deepseek-r1 |
| 2 | 0.8411 | 68_Matmul_Min_Subtract | deepseek-r1 |
| 2 | 0.8595 | 64_Gemm_LogSumExp_LeakyReLU_LeakyReLU_GELU_G... | deepseek-r1 |
| 2 | 0.8615 | 65_Conv2d_AvgPool_Sigmoid_Sum | deepseek-r1 |
| 2 | 0.8622 | 86_Matmul_Divide_GELU | deepseek-r1 |
| 2 | 0.8715 | 12_Gemm_Multiply_LeakyReLU | deepseek-r1 |
| 2 | 0.8801 | 61_ConvTranspose3d_ReLU_GroupNorm | openai-o1 |

Table 19: Best achieved speedup per problem instance across evaluated models, categorized by difficulty level and sorted by ascending speedup (continued)

| Level | Speedup | Problem | Model |
|---|---|---|---|
| 2 | 0.9006 | 46_Conv2d_Subtract_Tanh_Subtract_AvgPool | openai-o1 |
| 2 | 0.9008 | 22_Matmul_Scale_ResidualAdd_Clamp_LogSumExp_... | gpt-4o |
| 2 | 0.9009 | 97_Matmul_BatchNorm_BiasAdd_Divide_Swish | gpt-4o |
| 2 | 0.9062 | 47_Conv3d_Mish_Tanh | deepseek-r1 |
| 2 | 0.9092 | 82_Conv2d_Tanh_Scaling_BiasAdd_Max | deepseek-r1 |
| 2 | 0.9229 | 87_Conv2d_Subtract_Subtract_Mish | openai-o1 |
| 2 | 0.9236 | 3_ConvTranspose3d_Sum_LayerNorm_AvgPool_GELU | openai-o1 |
| 2 | 0.938 | 52_Conv2d_Activation_BatchNorm | openai-o1 |
| 2 | 0.9657 | 9_Matmul_Subtract_Multiply_ReLU | deepseek-r1 |
| 2 | 0.9701 | 78_ConvTranspose3d_Max_Max_Sum | deepseek-r1 |
| 2 | 0.9909 | 5_ConvTranspose2d_Subtract_Tanh | openai-o1 |
| 2 | 0.9922 | 38_ConvTranspose3d_AvgPool_Clamp_Softmax_Mult... | openai-o1 |
| 2 | 1.0014 | 31_Conv2d_Min_Add_Multiply | deepseek-r1 |
| 2 | 1.008 | 13_ConvTranspose3d_Mean_Add_Softmax_Tanh_Sca... | claude-3.5-sonnet |
| 2 | 1.0114 | 24_Conv3d_Min_Softmax | openai-o1 |
| 2 | 1.0118 | 72_ConvTranspose3d_BatchNorm_AvgPool_AvgPool | openai-o1 |
| 2 | 1.025 | 41_Gemm_BatchNorm_GELU_GroupNorm_Mean_ReLU | openai-o1 |
| 2 | 1.0511 | 43_Conv3d_Max_LogSumExp_ReLU | deepseek-r1 |
| 2 | 1.1045 | 16_ConvTranspose2d_Mish_Add_Hardtanh_Scaling | claude-3.5-sonnet |
| 2 | 1.11 | 53_Gemm_Scaling_Hardtanh_GELU | deepseek-r1 |
| 2 | 1.1227 | 100_ConvTranspose3d_Clamp_Min_Divide | deepseek-r1 |
| 2 | 1.1327 | 57_Conv2d_ReLU_HardSwish | deepseek-r1 |
| 2 | 1.1331 | 93_ConvTranspose2d_Add_Min_GELU_Multiply | gpt-4o |
| 2 | 1.1376 | 96_ConvTranspose3d_Multiply_Max_GlobalAvgPool:. | deepseek-V3 |
| 2 | 1.1503 | 95_Matmul_Add_Swish_Tanh_GELU_Hardtanh | deepseek-r1 |
| 2 | 1.1508 | 10_ConvTranspose2d_MaxPool_Hardtanh_Mean_Tanh | deepseek-r1 |
| 2 | 1.1792 | 50_ConvTranspose3d_Scaling_AvgPool_BiasAdd_Sc... | deepseek-r1 |
| 2 | 1.1838 | 83_Conv3d_GroupNorm_Min_Clamp_Dropout | openai-o1 |
| 2 | 1.1852 | 60_ConvTranspose3d_Swish_GroupNorm_HardSwish | deepseek-r1 |
| 2 | 1.1976 | 91_ConvTranspose2d_Softmax_BiasAdd_Scaling_Si... | openai-o1 |
| 2 | 1.2267 | 2_ConvTranspose2d_BiasAdd_Clamp_Scaling_Clamp... | openai-o1 |
| 2 | 1.2419 | 92_Conv2d_GroupNorm_Tanh_HardSwish_ResidualAd... | openai-o1 |
| 2 | 1.2444 | 19_ConvTranspose2d_GELU_GroupNorm | deepseek-r1 |
| 2 | 1.2534 | 49_ConvTranspose3d_Softmax_Sigmoid | deepseek-r1 |
| 2 | 1.2964 | 30_Gemm_GroupNorm_Hardtanh | deepseek-r1 |
| 2 | 1.3112 | 55_Matmul_MaxPool_Sum_Scale | deepseek-r1 |
| 2 | 1.32 | 81_Gemm_Swish_Divide_Clamp_Tanh_Clamp | deepseek-r1 |
| 2 | 1.3333 | 8_Conv3d_Divide_Max_GlobalAvgPool_BiasAdd_Su... | gpt-4o |

Table 19: Best achieved speedup per problem instance across evaluated
models, categorized by difficulty level and sorted by ascending speedup
(continued)

| Level | Speedup | Problem | Model |
|---|---|---|---|
| 2 | 1.3736 | 79_Conv3d_Multiply_InstanceNorm_Clamp_Multipl... | openai-o1 |
| 2 | 1.3981 | 62_Matmul_GroupNorm_LeakyReLU_Sum | deepseek-r1 |
| 2 | 1.4289 | 74_ConvTranspose3d_LeakyReLU_Multiply_LeakyReL... | openai-o1 |
| 2 | 1.4467 | 98_Matmul_AvgPool_GELU_Scale_Max | deepseek-r1 |
| 2 | 1.4604 | 39_Gemm_Scale_BatchNorm | deepseek-r1 |
| 2 | 1.4662 | 42_ConvTranspose2d_GlobalAvgPool_BiasAdd_LogSu... | deepseek-r1 |
| 2 | 1.5042 | 29_Matmul_Mish_Mish | deepseek-r1 |
| 2 | 1.5456 | 17_Conv2d_InstanceNorm_Divide | deepseek-r1 |
| 2 | 1.5745 | 26_ConvTranspose3d_Add_HardSwish | deepseek-r1 |
| 2 | 1.6439 | 88_Gemm_GroupNorm_Swish_Multiply_Swish | deepseek-r1 |
| 2 | 1.7277 | 58_ConvTranspose3d_LogSumExp_HardSwish_Subtrac... | deepseek-r1 |
| 2 | 1.7476 | 51_Gemm_Subtract_GlobalAvgPool_LogSumExp_GELU... | deepseek-r1 |
| 2 | 1.8268 | 18_Matmul_Sum_Max_AvgPool_LogSumExp_LogSumEx... | deepseek-r1 |
| 2 | 1.8839 | 48_Conv3d_Scaling_Tanh_Multiply_Sigmoid | openai-o1 |
| 2 | 2.0192 | 90_Conv3d_LeakyReLU_Sum_Clamp_GELU | claude-3.5-sonnet |
| 2 | 2.3173 | 7_Conv3d_ReLU_LeakyReLU_GELU_Sigmoid_BiasAdd | claude-3.5-sonnet |
| 2 | 2.3347 | 40_Matmul_Scaling_ResidualAdd | openai-o1 |
| 2 | 2.4595 | 20_ConvTranspose3d_Sum_ResidualAdd_Multiply_R... | deepseek-r1 |
| 2 | 2.6044 | 14_Gemm_Divide_Sum_Scaling | claude-3.5-sonnet |
| 3 | 0.0107 | 33_VanillaRNN | claude-3.5-sonnet |
| 3 | 0.18 | 43_MinGPTCausalAttention | openai-o1 |
| 3 | 0.3408 | 34_VanillaRNNHidden | openai-o1 |
| 3 | 0.5782 | 1_MLP | deepseek-r1 |
| 3 | 0.6927 | 24_EfficientNetB2 | openai-o1 |
| 3 | 0.7214 | 22_EfficientNetB0 | openai-o1 |
| 3 | 0.7382 | 9_ResNet18 | openai-o1 |
| 3 | 0.7455 | 20_MobileNetV2 | openai-o1 |
| 3 | 0.767 | 23_EfficientNetB1 | deepseek-V3 |
| 3 | 0.7815 | 10_ResNet101 | openai-o1 |
| 3 | 0.8083 | 46_NetVladWithGhostClusters | deepseek-V3 |
| 3 | 0.8194 | 13_DenseNet121TransitionLayer | claude-3.5-sonnet |
| 3 | 0.8249 | 32_ConvolutionalVisionTransformer | openai-o1 |
| 3 | 0.8284 | 7_GoogleNetInceptionV1 | openai-o1 |
| 3 | 0.855 | 15_DenseNet121 | gpt-4o |
| 3 | 0.8587 | 18_SqueezeNet | gpt-4o |
| 3 | 0.8591 | 16_DenseNet201 | gpt-4o |
| 3 | 0.8699 | 47_NetVladNoGhostClusters | llama-3.1-405b |
| 3 | 0.8743 | 28_VisionTransformer | gpt-4o |

Table 19: Best achieved speedup per problem instance across evaluated models, categorized by difficulty level and sorted by ascending speedup (continued)

| Level | Speedup | Problem | Model |
|---|---|---|---|
| 3 | 0.8846 | 19_MobileNetV1 | deepseek-V3 |
| 3 | 0.9343 | 27_RegNet | openai-o1 |
| 3 | 0.9608 | 26_ShuffleNet | gpt-4o |
| 3 | 0.9811 | 5_AlexNet | openai-o1 |
| 3 | 0.9858 | 25_ShuffleNetUnit | deepseek-V3 |
| 3 | 1.0048 | 14_DenseNet121DenseBlock | openai-o1 |
| 3 | 1.0089 | 6_GoogleNetInceptionModule | openai-o1 |
| 3 | 1.012 | 8_ResNetBasicBlock | openai-o1 |
| 3 | 1.025 | 29_SwinMLP | llama-3.1-405b |
| 3 | 1.027 | 11_VGG16 | deepseek-V3 |
| 3 | 1.0442 | 12_VGG19 | deepseek-V3 |
| 3 | 1.08 | 36_LTSMHn | openai-o1 |
| 3 | 1.0922 | 3_DeepNarrowMLP | deepseek-r1 |
| 3 | 1.2704 | 44_MiniGPTBlock | deepseek-V3 |
| 3 | 1.405 | 4_LeNet5 | claude-3.5-sonnet |
| 3 | 1.9376 | 50_ReLUSelfAttention | openai-o1 |

# O. Alternative Library: Triton

Alternative GPU programming tools to CUDA – such as CUTLASS (NVIDIA, 2017a), Triton (Tillet et al., 2019), ThunderKittens (Spector et al., 2024) – have been developed to make GPU programming easier by exposing a higher level of abstraction. We agree that this is an exciting direction to improve model performance and had noted this as a direction for future work in Section 6.2. We reiterate that the goal of this work is to propose a new benchmark framework and to thoroughly evaluate the baselines, rather than to solve the full kernel generation problem.

### O.1. Triton Task Specification

We extend KernelBench with a Triton evaluation backend and a Triton task specification. Instead of `<PyTorch, PyTorch + CUDA>`, we now define the task as `<PyTorch, PyTorch + Triton>`.

Similar to the PyTorch one-shot example used in C.1 that describes the simple operation `a+b`, we provide the model with an inline Just-In-Time compilation (JIT) kernel example in Triton.

```
1  import torch
2  import torch.nn as nn
3  import torch.nn.functional as F
4  import triton
5  import triton.language as tl
6
7
8  @triton.jit
9  def add_kernel(
10     x_ptr,  # Pointer to first input
11     y_ptr,  # Pointer to second input
12     out_ptr,  # Pointer to output
13     n_elements,  # Total number of elements in input/output
14     BLOCK_SIZE: tl.constexpr,
```

```
15  ):
16          # Each program handles a contiguous block of data of size BLOCK_SIZE
17          block_start = tl.program_id(0) * BLOCK_SIZE
18          # Create a range of offsets [0..BLOCK_SIZE-1]
19          offsets = block_start + tl.arange(0, BLOCK_SIZE)
20          # Mask to ensure we don't go out of bounds
21          mask = offsets < n_elements
22          # Load input values
23          x = tl.load(x_ptr + offsets, mask=mask, other=0.0)
24          y = tl.load(y_ptr + offsets, mask=mask, other=0.0)
25          # Perform the elementwise addition
26          out = x + y
27          # Store the result
28          tl.store(out_ptr + offsets, out, mask=mask)
29
30
31  def triton_add(x: torch.Tensor, y: torch.Tensor):
32          """
33          This function wraps the Triton kernel call. It:
34            1. Ensures the inputs are contiguous on GPU.
35            2. Calculates the grid (blocks) needed.
36            3. Launches the Triton kernel.
37          """
38          assert x.is_cuda and y.is_cuda, "Tensors must be on CUDA."
39          x = x.contiguous()
40          y = y.contiguous()
41
42          # Prepare output tensor
43          out = torch.empty_like(x)
44
45          # Number of elements in the tensor
46          n_elements = x.numel()
47          BLOCK_SIZE = 128  # Tunable parameter for block size
48
49          # Determine the number of blocks needed
50          grid = lambda meta: ((n_elements + meta["BLOCK_SIZE"] - 1) // meta["BLOCK_SIZE"],)
51
52          # Launch the Triton kernel
53          add_kernel[grid](x, y, out, n_elements, BLOCK_SIZE=BLOCK_SIZE)
54          return out
55
56
57  class ModelNew(nn.Module):
58      def __init__(self) -> None:
59          super().__init__()
60
61      def forward(self, a, b):
62          # Instead of "return a + b", call our Triton-based addition
63          return triton_add(a, b)
```

| $\text{fast}_1$ over: | PyTorch Eager | | | torch.compile | | |
|---|---|---|---|---|---|---|
| KernelBench Level | 1 | 2 | 3 | 1 | 2 | 3 |
| GPT-4o | 2% | 7% | 2% | 13% | 2% | 2% |
| OpenAI o1 | 3% | **17%** | **10%** | **22%** | **13%** | **8%** |
| DeepSeek R1 | **6%** | 13% | 2% | 19% | **13%** | 4% |
| Llama 3.1-70B Inst. | 1% | 0% | 0% | 9% | 0% | 0% |

*Table 20.* **KernelBench-Triton Baseline**. Similar to Table 1, we present $\text{fast}_1$ against PyTorch Eager and `torch.compile` on NVIDIA L40S. The LMs target Triton here instead of CUDA.

## O.2. Baseline Evaluation with Triton

We again conduct baseline evaluation on 3 Levels on KernelBench, just like Section 4.1, but with the Triton task and associated evaluation. As shown in Table 20, we find that models perform worse when targeting Triton compared to CUDA (Table 1), both in terms of correctness and performance. For example, DeepSeek R1 $fast_1$ drops from 12%, 36%, 2% to 6%, 13%, 2% for Level 1, 2, and 3 respectively. Analyzing the errors and samples, we found many Triton-related errors, likely due to Triton being a more rare source of training data than CUDA, highlighting challenges for using low-resource domain-specific libraries.

