# OpenReview forum: "KernelBench: Can LLMs Write Efficient GPU Kernels?"
_ICML.cc/2025/Conference — ICML 2025 poster_

### Official Review · Reviewer_qSdx · 2025-03-10

**Overall Recommendation:** 3

**Summary:**

The major contribution of this paper is as follows:
- This paper introduced a benchmark framework to evaluate how good a modern LLM can write efficient GPU kernels. The core of this benchmark framework consisting of 250 tasks with 3 levels of granularity: single primitive, sequence of ops and the overall model.
- Using the benchmark, this paper evaluated modern LLMs. The conclusion is that there's still a large room of improvement as modern LLMs face challenges on both correctness ratio and efficiency. Experiments shows repeated or iterative sampling may help, but still yielding unsatisfactory kernel performance.

**Claims And Evidence:**

The evaluation in this paper well supports the claim that modern LLMs underperform.

**Essential References Not Discussed:**

There's a line of research on automatic kernel generation solving the same problem but using compiler-based methods like AStitch (ASPLOS'22), Welder (OSDI'23), ROLLER (OSDI'22) or older foundation works like TVM (OSDI'18)

**Experimental Designs Or Analyses:**

The evaluation for the correctness and evaluation is well designed.

One concern mentioned above is the lack of evaluation on whether the decision on the subset of ops to be fused is correct.
Moreover, it would be beneficial to compare with some SoTA auto fusion methods.

**Methods And Evaluation Criteria:**

The design of the benchmark successfully cover basic aspects including throughput, correctness while describing the tasks with plain text + pytorch ref code + sample inputs. This design is simple-yet-effective.

In several aspects, the paper should have explored deeper.
- One major concern is letting LLM to generate kernel without hardware information. In real world senarios, the hardware specs like number of registers, shared memory, cuda compatibility, etc would critically affect the performance of a kernel.
- In the task designed by this paper, it's important for the LLM to decide when and how to select a subset of ops to fuse (write kernel) for. E.g. fusing the attention block, but not the entire transformer block or even the entire model. This paper does not show much information on this and it's unclear whether the low performance of LLMs, especially in task 3, are writing low-performance models because of poor CUDA optimizations or simply writing kernels for blocks that should't be fused.

**Other Comments Or Suggestions:**

- It will be good for the paper to provide more details on the designed tasks, especially for level 2&3 for readers to understand the coverage of this tasks.

**Other Strengths And Weaknesses:**

Strengths: A benchmark measuring basics of LLM GPU code generation

Weaknesses: See methods and experiment section

**Questions For Authors:**

- In the benchmark design, how does the LLM make decision on which part of the computation we need to generate kernel for? Do they tend to generate a single kernel for all computation?
- Will it be more beneficial to use a DAG representation instead of a pytorch ref code representation?

**Relation To Broader Scientific Literature:**

This benchmark would facilitate the research on automatic GPU code generation, which is underexplored but attracting increasing attention.

**Theoretical Claims:**

Benchmark paper, no theoretical claims.

---

> ### Author Rebuttal · Authors · 2025-04-01
>
> Updated Paper: https://storage.googleapis.com/anonymous-files/kernelbench.pdf
>
> We thank you for appreciating KernelBench design and suggesting further improvements. As you noted, automatic GPU code generation is an underexplored area with many interesting research questions; KernelBench facilitates research in this direction as the first benchmark and environment for kernel development, with “simple-yet-effective” task definition and “well-designed evaluation”. In fact, we have already seen enthusiasm from the community, with multiple projects tackling KernelBench through agentic optimization and post-training.
>
> **Providing Model with Hardware Information**
> We totally agree that specifying hardware information is important as GPU kernels are inherently platform-dependent. In fact, this was already studied in our original submission. In Section 5.2.2, we provided the model with the exact kind of GPU hardware specifications (see Appendix C.5) that the reviewer described, and found that current models rarely conduct optimization correctly for underlying hardware when provided with such information.
>
> **Clarifying Design Choice to Test Kernel Fusion**
> In KernelBench the model has full flexibility to decide what subset of operators in the PyTorch reference to optimize and fuse. We believe this is one of the crucial abilities when a model is given distinct or new architectures in the real-world setting.
>
> KernelBench’s 3-level categorization helps disentangle fusion decisions and kernel generation. Level 1 problems (single operators) only test the model's ability to write optimized kernels; Level 2 and 3 problems are designed to additionally evaluate the model's ability to identify and leverage fusion opportunities; Appendix K provides a detailed task breakdown.
>
> **Fusion Patterns in Model Generated Code**
> To answer your questions about **fusion patterns** in generated kernels, we manually inspected the kernels generated by the best performing model, DeepSeek R1, and provided new analysis in Appendix L. We focus on level 2 problems, which are composed with one mainloop (e.g. conv, matmul) and 2-5 epilogue operations (non-linearities, reductions, etc). We observe model-generated code always attempts to generate 1-2 fused kernels per problem. As shown in Figure 19, the fused kernels tend to contain more than half the operators in the program. To explicitly answer your question, only 18% of programs fuse all operators into a single kernel.
>
> Regarding your question on the quality of fusion decisions and whether they cause low performance, we analyzed the generated kernels that were slower than PyTorch Eager (as shown in Table 17\) and draw two observations: 1\) main loop operators (eg. Conv) were not fused with epilogue operators 2\) the model’s attempt to fuse main loop operators (e.g. GEMM \+ other ops) was not faster than launching highly-optimized cuBLAS kernels. Also refer to **“Analysis of Performance Degradation Cases”** in response to Reviewer ufH2 for a related study.
>
> **Comparison with SOTA Compiler-Based Approach**
> Thank you for bringing up relevant compiler-based approaches (AStitch, Welder, ROLLER, TVM), which we have added and elaborate on in our updated related works.
>
> To directly address your concern, we compare fusion decisions in model-generated kernels with auto-fusion compilers. Since AStitch, Welder, ROLLER could not be run on KernelBench due to format incompatibilities or outdated support for KernelBench PyTorch 2.5 /CUDA 12.4 (Appendix B), we focus comparison on widely-adopted torch.compile with SoTA performance (better than TVM, see Table 3 in PyTorch 2 \[1\], ASPLOS ‘24) and employs an auto-fusion policy over TorchInductor’s define-by-run IR. We show both fusion decisions of R1 and torch.compile in Table 17\. Torch.compile often creates sophisticated fusion patterns by breaking Convolutions or GroupNorm into smaller multi-pass kernels that compute partial results and statistics in parallel — behavior that R1-kernels rarely exhibit.
>
> **DAG Representation**
> Per your suggestion, we conduct experiments on using a DAG representation (ONNX, torch.fx graph) “instead” of a PyTorch Reference, which might help highlight fusion opportunities. We explored this in Appendix M.2 and found that DAG representations cause output mismatch issues on problems that succeed with PyTorch representations – see response to reviewer ufH2 for details.
>
> We appreciate the comprehensive comments and hope that our additional experiments, analysis of results, and discussion addressed your concerns. We hope you find the paper significantly improved and consider reflecting this in your final score.
>
> \[1\] PyTorch 2: Faster Machine Learning Through Dynamic Python Bytecode Transformation and Graph Compilation. ASPLOS '24. https://doi.org/10.1145/3620665.3640366

---

> > ### Comment · Reviewer_qSdx · 2025-04-03
> >
> > Thanks for the detailed study on fusion and failure patterns, I'll update my score to 3.

---

### Official Review · Reviewer_ufH2 · 2025-03-12

**Overall Recommendation:** 3

**Summary:**

This paper introduces KernelBench, a benchmarking framework designed specifically to evaluate the correctness and performance of GPU CUDA kernels generated by large language models (LLMs). KernelBench compiles a representative set of PyTorch code snippets, categorizing them into three distinct complexity levels based on their granularity. By systematically assessing the correctness and performance gains achieved by various LLMs when converting these PyTorch snippets into CUDA kernels, KernelBench provides a comprehensive evaluation of the kernel generation capabilities across multiple prominent LLMs and hardwares.

**Claims And Evidence:**

While the majority of the claims in this paper are well-supported by strong evidence, there are areas where the arguments and evidence could be further strengthened:

- This paper claims to propose a benchmark for evaluating the performance of LLMs in generating CUDA kernels. While the evaluation and analysis are indeed quite detailed, the results show numerous instances of performance degradation or no change. Providing a more comprehensive analysis of cases where LLM-generated kernels result in performance degradation would offer a more balanced and realistic portrayal of LLMs' capabilities in this domain.

**Essential References Not Discussed:**

N/A

**Experimental Designs Or Analyses:**

I have reviewed the experimental designs and analyses for their soundness and validity, and I found no major issues or concerns to address.

**Methods And Evaluation Criteria:**

**Methods and Potential Issues:**

- This paper introduces KernelBench to evaluate the kernel generation capabilities of LLMs. However, the design of KernelBench primarily focuses on assessing the translation of PyTorch code into CUDA kernels, which essentially evaluates code translation abilities. Including a broader range of code or natural language to CUDA kernel translations would significantly enhance its value.
- KernelBench appears to be more suited for generating raw CUDA C++ code. In reality, LLMs might have the potential to leverage other general-purpose tools (such as Triton, CUTLASS) for kernel generation. Exploring these possibilities in the paper would make the results more comprehensive.

**Evaluation Criteria and Potential Issues:**

- This paper employs the formula (in Line 213) as the standard for evaluating LLM-generated GPU kernels:

  This formula integrates both correctness and performance dimensions to comprehensively assess the capabilities of LLMs. However, the metric does not account for differences in task complexity. For example, Level 1 tasks (single operations) and Level 3 tasks (full architectures) vary significantly in difficulty, but the $fast_p$ metric does not weight or adjust for these differences.

**Other Comments Or Suggestions:**

The optimization prompts for different levels of tasks may need to be differentiated. For instance, for Level 1 tasks (operator-level generation), it might be necessary to guide the LLM to generate lower-level CUDA code, essentially rewriting the entire operator. On the other hand, for Level 3 tasks (graph-level generation), it may not be necessary for the LLM to generate CUDA code for all operators. Instead, it might be more effective to guide the LLM to rewrite part of specific PyTorch functions within the network to achieve optimization.

**Other Strengths And Weaknesses:**

**Strengths:**

- The paper introduces a comprehensive benchmark, KernelBench, designed to evaluate the ability and potential of Language Models (LMs) in generating efficient GPU CUDA kernels.
- The authors propose a novel metric, $fast_p$, which combines both correctness and performance to evaluate the quality of generated kernels. This dual-focus approach offers a more holistic and nuanced assessment compared to traditional metrics that rely solely on correctness or performance. The analysis of mainstream large models using this metric provides valuable insights into their capabilities in generating CUDA kernels.
- The paper conducts a wide range of experiments across multiple dimensions, including prompt content, hardware types, and operator categories.

**Weaknesses:**

- The paper focuses exclusively on evaluating LMs' ability to generate raw CUDA C++ code. However, it does not explore the potential of LMs to leverage high-performance libraries such as CUTLASS or Triton, which could significantly enhance kernel performance. Investigating these tools could reveal the upper bounds of LLM-generated kernel performance and provide a more comprehensive evaluation.
- While the paper examines the impact of different GPU parameters, it does not delve into architecture-specific optimizations. Modern GPUs, such as those based on Ampere or Hopper architectures, offer unique features like asynchronous memory access and warp specialization. Incorporating experiments that utilize these architecture-specific optimizations could unlock further performance gains and provide a more complete picture of LLM capabilities.
- The paper does not address the design of different data types, which are closely related to Tensor Core utilization in CUDA. The benchmark also lacks designation about specific Tensor Core code. Given the importance of Tensor Cores for mixed-precision models (e.g., FP16, FP8) commonly used in SOTA LLMs, incorporating Tensor Core-specific designs would significantly enhance the relevance and value of the benchmark.

**Questions For Authors:**

See Weakness and Comments.

**Relation To Broader Scientific Literature:**

N/A

**Theoretical Claims:**

The paper did not make any theoretical claims or proofs.

---

> ### Author Rebuttal · Authors · 2025-04-01
>
> Updated Paper: https://storage.googleapis.com/anonymous-files/kernelbench.pdf
>
> We sincerely thank you for the detailed and insightful review\! We are truly encouraged by the positive feedback, particularly that the work is seen as "well-supported by strong evidence" and a "comprehensive benchmark." We especially appreciate the recognition of our "novel metric, fast\_p," as offering a "more holistic and nuanced assessment," and that you found value in our "wide range of experiments across multiple dimensions." We were glad to read that you felt our work "provides valuable insights into \[LLMs'\] capabilities in generating CUDA kernels", which are valuable for AI and HPC communities as we explore automating kernel optimization. These strengths highlight core goals we aimed for, and we're glad they resonated.
>
> **Analysis of Performance Degradation Cases**
> Per your suggestion, in addition to our existing error analysis (4.2) and case study of the fastest kernels (Appendix D), we added a new “Case study: Performance Degradation” (Appendix N), which specifically examines instances where generated kernels underperformed compared to the baseline. Here are our findings:
>
> 1. LLM implementations of core ops (matmul, conv) underperform highly-optimized proprietary (e.g. cuDNN) kernels in PyTorch.
> 2. LLM correctly identifies fusion patterns, but fused operations (often matmuls) are not efficiently implemented, outweighing benefits from reduced memory access
> 3. LLM blocks better PyTorch native fusion capabilities by generating a custom kernel for a minor task that prevents optimizing across a larger sequence of operations.
>
> **Alternative Input Specification**
> Per your suggestion, we explored in Appendix M using 1\) Natural Language 2\) DAGs of program operators as input specification. On a representative Level 2 problem that model succeeded with PyTorch representations, it failed with compilation and logical issues on natural language representation due to ambiguity on exact behaviors, even when provided with verbose dimension details. DAG representations capture the program execution much better and hint the model to conduct fusions, but going from DAG to kernel directly can lead to logical errors resulting in output mismatch.
>
> **Using Libraries & DSLs**
> Reviewer raises the point of whether generating code using frameworks like Triton/CUTLASS would be helpful. To address your feedback, we extended KernelBench with a Triton task specification and evaluation backend. As shown in Appendix O Table 20, we found models perform worse when using Triton, both in terms of correctness and performance: fast\_1 for DeepSeek R1 drops from 12%, 36%, 2% to 6%, 13%, 2% across 3 levels respectively. Qualitatively, we found models generate many Triton-related errors, likely due to Triton being a rarer source of training data than CUDA, highlighting potential challenges for using domain-specific libraries.
> We reiterate KernelBench’s goal is to propose a new benchmark with thorough baseline evaluation as a first step, rather than solving kernel generations.
>
> **Level-Specific Prompting and Scoring**
>
> We made a deliberate choice to not explicitly weight fast\_p by task complexity (e.g., Level 1 vs. 3). We report the levels separately, and harder tasks are expected to yield lower scores naturally, reflecting greater challenges. This approach is common in other coding benchmarks (LiveCodeBench) that include easy/medium/hard problems without score normalization by difficulty. Regarding level-specific prompting, our baseline evaluations intentionally used general prompts, rather than the suggested task-specific ones, to evaluate each model's fundamental ability to independently discover and select optimization strategies across task complexities without explicit steering.
>
> **Architecture-Specific Optimizations**
> Based on your suggestions, we added experiments for eliciting architecture-specific optimizations: Tensor Cores and asynchronous memory transfers (Appendix G.3) on Ampere GPUs, in addition to existing experiments in Section 5.2. We provided DeepSeek-R1 with examples using wmma and memcpy\_async instructions, on simple KernelBench matrix multiply problems in FP16 (compatible with Tensor Cores). We observed that the model attempted to apply yet struggled to utilize those advanced instructions. Among the 5/17 correct kernels that use WMMAs, successfully leveraging Tensor Cores did not lead to better performance over PyTorch. No kernels used memcpy\_async correctly. This highlights that successfully utilizing hardware features remains challenging for models and KernelBench provides a playground for the community to develop methods that address this limitation.
>
> Once again, we thank you for your valuable time and feedback. We hope that our additional experiments, analysis of results, and discussion addressed your concerns. We hope you find the paper significantly improved and consider reflecting this in your final score.

---

> > ### Comment · Reviewer_ufH2 · 2025-04-06
> >
> > Thank you for your efforts and clarification. Overall, I am leaning toward acceptance and will keep my score unchanged.

---

### Official Review · Reviewer_AoMw · 2025-03-13

**Overall Recommendation:** 3

**Summary:**

This paper proposes KernelBench, which is a new benchmark for evaluating LLM's performance in writing correct and fast kernels. Specifically, KernelBench gathers three different levels of tasks, including individual operations, sequence of operations, and end-to-end architectures, and introduces a novel fast_p metric to model correctness and efficiency at the same time. KernelBench shows that most frontier models do not perform well in writing kernels, among which the state-of-the-art reasoning models perform the best. KernelBench further shows that leveraging feedback is important for reducing execution errors and discovering faster solutions:

**Claims And Evidence:**

Most claims made in the submission are supported by clear and convincing evidence.

**Essential References Not Discussed:**

N/A

**Experimental Designs Or Analyses:**

There is one weakness regarding the analysis:
- There is a lack of comparison to existing programming benchmarks like HumanEval/MBPP and LiveCodeBench. While it is reasonable that some models are expected to have various ranks across different benchmarks as KernelBench has a different focus, the overall ranking should largely align well with the mainstream benchmarks.

**Methods And Evaluation Criteria:**

The evaluation metric fast_p makes sense for KernelBench. However, apart from fast_p which takes both correctness and speedup into consideration, I would recommend adding two separate metrics for correctness and speedup separately for clearer demonstration.

For the evaluation approach, KernelBench does not handle the issue of cross-platform variation. While the paper claims that it "does not provide ground truth kernels for the tasks since we imagine users benchmarking on a variety of hardware platforms (including new platforms)", it's possible that, in terms of speedup, some model performs the best on one platform while the other model performs the best on the other platform -- it will make the evaluation results (even the model rankings) super hard to reproduce.

**Other Comments Or Suggestions:**

N/A

**Other Strengths And Weaknesses:**

N/A

**Questions For Authors:**

N/A

**Relation To Broader Scientific Literature:**

This paper shows that leveraging feedback is important for reducing execution errors and discovering faster solutions, which aligns well with many existing works that focus on more general code generation tasks [1,2].

[1] Chen, Xinyun, et al. "Teaching large language models to self-debug." arXiv preprint arXiv:2304.05128 (2023).

[2] Xia, Chunqiu Steven, and Lingming Zhang. "Keep the Conversation Going: Fixing 162 out of 337 bugs for $0.42 each using ChatGPT." arXiv preprint arXiv:2304.00385 (2023).

**Theoretical Claims:**

There's no theoretical claim in this paper.

---

> ### Author Rebuttal · Authors · 2025-04-01
>
> Updated Paper: https://storage.googleapis.com/anonymous-files/kernelbench.pdf
>
> We thank reviewer AoMw for your review\! We are glad the reviewer appreciates our "fast\_p evaluation metric for KernelBench" and finds that our "claims are supported by clear and convincing evidence." Below, we address your comments regarding the evaluation metrics, platform dependency, and comparison to other benchmarks.
>
> **Clarity of fast\_p vs. Separate Correctness and Speedup Metrics**
>
> “Adding two separate metrics for correctness and speedup” is a great suggestion for added clarity. In our revision (see link above), we included an extended table presenting correctness (equivalent to fast\_{p=0}) and geo-mean speedup as separate metrics (Appendix I, Table 16), providing a disaggregated view for these specific aspects. The geometric mean of speedups only includes the correct generations, as fast but incorrect code is not helpful.
>
> Thanks for acknowledging that the “evaluation metric fast\_p makes sense for KernelBench\!” We also reiterate that for kernel generation, speedup and correctness are tightly coupled, motivating our choice to design fast\_p. To provide more context on this, we've also added a section discussing our metric design explorations (Appendix I). We hope the combination of the original fast\_p and these new separate metrics offers more clarity.
>
> **Platform Dependency and Reproducibility**
>
> We agree entirely about the platform-specific nature of hardware performance tuning. Thus, it is very important to compare results when controlling for both the input program and the underlying hardware. For instance, our evaluations (as noted in paper Section 4.4 and Appendix G.1) across several hardware platforms, including L40S, A100, and H100 GPUs, revealed reasonable consistency in the kernel generations at Level 1 but more pronounced variation in Level 2\. Other than this hardware evaluation study, most of our experiments in the paper are done on an Nvidia L40S, and we expect the results to be reproducible on this type of GPU.
>
> **Comparison to Existing Programming Benchmarks**
>
> To address your comment, we've added a new experimental section (see Appendix J) where we compare model performance on KernelBench (KB) with LiveCodeBench (LCB). (We chose LCB as HumanEval performance is quite saturated for current models). We present the relative rankings of models across these benchmarks.
>
> Our results show, perhaps unsurprisingly, that models performing well on general coding benchmarks tend to also perform better on KernelBench, but variability in rankings (e.g. o1 ranks 1st in LCB and 2nd in KB Level 1, and R1 ranks 2nd in LCB but 1st in KB Level1) across different levels of KernelBench suggests that additional skills are required for high performance in kernel-specific tasks – intuitively, this aligns with the major differences between GPU programming and standard programming problems found in popular coding benchmarks. We would also like to highlight that KernelBench is not merely another code generation benchmark; it adds the critical dimensions of performance optimization and hardware awareness, testing a model's ability to generate not only correct code, but also efficient code, which presents distinct challenges.
>
> **Relation to Broader Literature of Leveraging Feedback**
>
> We definitely see the connection with existing works on leveraging feedback too (we also added citations of works listed by you here)\! We believe KernelBench takes this concept into a particularly challenging and impactful domain. Optimizing hardware kernels (notoriously difficult even for human experts) offers tangible real-world benefits (cost and energy savings for AI\!), making it a high-stakes environment given the ubiquity and importance of AI systems today.
>
> We intentionally designed KernelBench to facilitate precisely iterative, feedback-driven improvement. By providing rich, actionable feedback signals—clear correctness checks (pass/fail), compilation status, precise runtime measurements, and speedup relative to a baseline—KernelBench creates an environment where AI systems can directly learn from their attempts and refine their solutions. The goal of our baseline results using feedback is to thoroughly characterize the degree to which we can solve KernelBench. We find that despite using these techniques, the best model gets fast\_p=1 of only 18% on level 3, showing there’s a lot more progress to be made on this benchmark. In this sense, we see KernelBench as a stepping stone for pushing forward research in automated kernel engineering, providing a crucial contribution to the community as a standard evaluation environment.
>
> In light of these clarifications as well as new experimental results and modifications to the paper to address your comments, we would really appreciate it if you would re-examine our paper and consider raising your score.

---

### Decision · Program_Chairs · 2025-05-01

**Decision:**

Accept (poster)

**Comment:**

The authors give a benchmark for LLM-generation of GPU code, which is doubly relevant to the machine learning community both because of the target application and the method. Reviewers are unanimous in recommending acceptance. I agree with the reviewers and would urge the authors to incorporate discussion of the related work that was raised by qSdx.